# All-day Multi-scenes Lifelong Vision-and-Language Navigation with Tucker Adaptation

**Xudong Wang**[*,1,3], **Gan Li**[*,1,2], **Zhiyu Liu**[1,3], **Yao Wang**[4], **Lianqing Liu**[1], **Zhi Han**[†,1],

[1] State Key Laboratory of Robotics and Intelligent Systems, Shenyang Institute of Automation, Chinese Academy of Sciences,
[2] North University of China,
[3] University of Chinese Academy of Sciences,
[4] Xi'an Jiaotong University

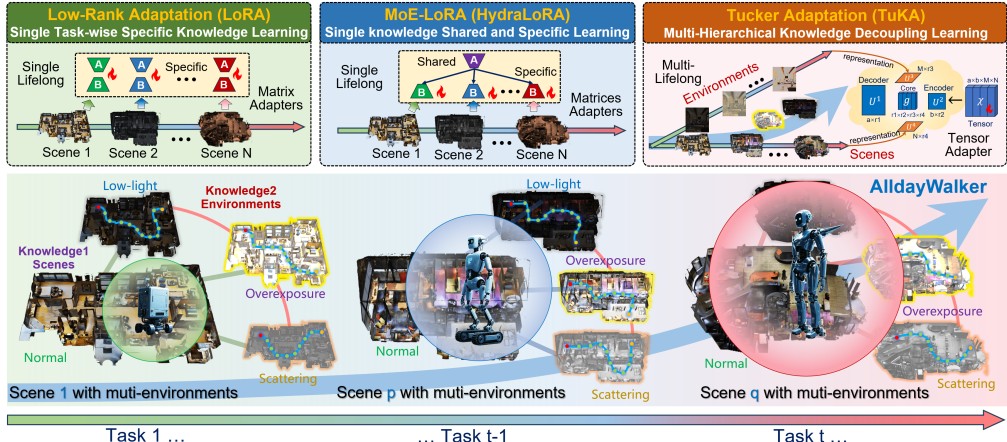

Figure 1: Illustration of the proposed all-day multi-scenes lifelong vision-and-language navigation learning and Tucker Adaptation (TuKA). It requires VLN agents to continually learn across multiple scenes and diverse environments (low-light, overexposure, and scattering), progressively consolidating navigation knowledge to achieve all-day multi-scenes navigation. Different from LoRA and its variants, which only perform continual learning with single-dimensional task knowledge, our proposed TuKA decouples and represents multi-hierarchical task knowledge in a high-order tensor.

## Abstract

Deploying vision-and-language navigation (VLN) agents requires adaptation across diverse scenes and environments, but fine-tuning on a specific scenario often causes catastrophic forgetting in others, which severely limits flexible long-term deployment. We formalize this challenge as the all-day multi-scenes lifelong VLN (AML-VLN) problem. Existing parameter-efficient adapters (e.g., LoRA and its variants) are limited by their two-dimensional matrix form, which fails to capture the multi-hierarchical navigation knowledge spanning multiple scenes and environments. To address this, we propose Tucker Adaptation (TuKA), which represents the multi-hierarchical navigation knowledge as a high-order tensor and leverages Tucker decomposition to decouple the knowledge into shared subspaces and scenario-specific experts. We further introduce a decoupled knowledge incremental learning strategy to consolidate shared subspaces while constraining specific experts for decoupled lifelong learning. Building on TuKA, we also develop a VLN agent named AlldayWalker, which continually learns across multiple navigation scenarios, achieving all-day multi-scenes navigation. Extensive experiments show that AlldayWalker consistently outperforms state-of-the-art baselines. Code and video demos are available at: **https://ganvin-li.github.io/AlldayWalker/**.

## 1 Introduction

Vision-and-Language Navigation (VLN) requires embodied agents to follow user instructions and reach target locations in a navigation scene (Lin et al. (2025); Wei et al. (2025); Zhang et al. (2025c)).

---

[*] Equal contribution. [†] Corresponding author.

Since its introduction (Anderson et al. (2018)), VLN research has rapidly advanced from early studies in discrete, graph-based simulators (Anderson et al. (2018); Ku et al. (2020); Wang et al. (2026a)) to continuous embodied platforms (Krantz et al. (2020)), large-scale persistent navigation benchmarks (Krantz et al. (2023); Song et al. (2025)), and even real deployments (Wei et al. (2025); Zhang et al. (2024a)). These advances make VLN a capability for robots that interact with humans.

However, similar to many other robotics learning tasks (Xiao et al. (2025); Wang et al. (2023d); Liang et al. (2025)), directly deploying VLN agents in the real world often falls short of practical requirements. To achieve reliable performance, agents typically require fine-tuning to adapt to a specific scenario. Yet in real applications, agents often operate in dynamic scenarios that involve both diverse scenes and illumination environments. Adapting to one specific scenario usually comes at the cost of degraded performance in others, leading to catastrophic forgetting (Meng et al. (2025); Yao et al. (2025); Zhu et al. (2025); Ayub et al. (2025)). Inspired by human lifelong learning and continual evolution, we aim to build VLN agents that can learn across multiple scenes and environments, evolving over time to achieve all-day multi-scenes navigation without forgetting, as shown Figure 1.

Parameter-efficient adapters such as LoRA (Hu et al. (2022)) are widely used to adapt pretrained large models with only a few extra parameters. However, applying separate LoRA modules per navigation scenario fails to capture cross-task shared knowledge, preventing the agents from leveraging accumulated knowledge to improve adaptation to new scenarios. To explore task-shared knowledge, mixture-of-expert LoRA variants (e.g., HydraLoRA (Tian et al. (2024)), Dense MoLE (Chen et al. (2024)), BranchLoRA (Zhang et al. (2025a))) employ multi-expert co-activation to represent shared components. As illustrated in the upper part of Figure 1, these LoRA-based approaches still represent knowledge using two hierarchical matrices (one shared factor with several task-specific matrices). Such a two-dimensional, matrix-based representation is inherently limited for learning multi-hierarchical navigation knowledge that spans multiple scenes and diverse environments. Inspired by the powerful capability in high-dimensional space representation learning (Verleysen et al. (2003); Stöckl et al. (2024)), we ask: ***can we represent the multi-hierarchical knowledge in a high-order tensor, thereby enabling stronger multi-hierarchical, decoupled representation learning?***

To realize high-dimensional space representation learning with a high-order tensor, we identify two critical challenges: i) *how to continually decouple representation learning across multi-hierarchical knowledge*, and ii) *how to align the higher-order dimensions with the two-dimensional matrix backbone used in LLM adaptation*. To address the challenges, we propose Tucker Adaptation (TuKA), a new fine-tuning method that employs Tucker decomposition (Kolda & Bader (2009)) to represent multi-hierarchical navigation knowledge. TuKA represents scene knowledge and environmental knowledge in distinct expert factor matrices, and represents shared knowledge across multiple tasks using a shared core tensor and encoder-decoder. To align LLM parameters, TuKA selects each specific expert from a row within its entire expert factor matrices, reducing the expert matrix dimension to a vector. Thus, the higher-order knowledge tensor is reduced to a two-dimensional weight matrix for aligning LLM parameters. We further design a Decoupled Knowledge Incremental Learning (DKIL) strategy that consolidates shared subspaces while constraining the task-specific experts to mitigate catastrophic forgetting, during multi-hierarchical knowledge lifelong learning. Building on TuKA, we also develop AlldayWalker, a lifelong VLN agent that continually adapts across multiple scenes and diverse environments. To support this study, we extend an existing embodied navigation simulator with multiple degraded imaging models (including low-light, scattering, and overexposure) to produce diverse environments for training and evaluation. Extensive experiments are performed to demonstrate that AlldayWalker outperforms the SOTA fine-tuning methods in lifelong VLN performance, enabling all-day multi-scenes VLN. Our contributions are threefold:

- We formalize the all-day multi-scenes lifelong VLN learning problem, and propose a novel parameter-efficient adaptation method named TuKA to decouple and represent the multi-hierarchical knowledge in a high-order tensor, for more powerful representation learning.

- We develop AlldayWalker, an all-day multi-scenes lifelong VLN agent that continually evolves using a decoupled knowledge incremental learning strategy across multiple navigation scenes and diverse environments, thus achieving all-day multi-scenes navigation.

- We extend the existing embodied navigation simulators with imaging models to construct an all-day multi-scenes lifelong VLN benchmark for evaluation with diverse environments. And additional real-world deployments also validate the superiority of our AlldayWalker.

## 2 PRELIMINARY AND PROBLEM FORMULATION

**Preliminary:** In the vision-and-language navigation (VLN) task, an embodied agent is required to understand user language instruction $\mathcal{I}$, e.g., "*walk forward to the white wooden table, turn right into the bedroom, turn left to the wardrobe*", and follow the instruction to navigate in a scene $S$. Following prior work on VLN agents (Wei et al. (2025); Zheng et al. (2024); Zhang et al. (2025b); Gao et al. (2025a)), we adopt a pre-trained large language model (LLM), Qwen2-7B (Team (2024)), with a tokenizer $e$, as the backbone agent $\mathcal{F}$. For encoding the agent's video stream observations $\mathcal{O}$, we use the CLIP vision encoder $\mathcal{V}(\mathcal{O})$, consistent with StreamVLN (Wei et al. (2025)). The overall architecture is similar to multimodal large language models such as LLaVA-Video (Zhang et al. (2024c)). At each navigation step $i$, the agent $\mathcal{F}$ reasons over the user instruction $\mathcal{I}$ and the current observation $\mathcal{O}^i$ to generate the next action: $R^i = \mathcal{F}(\mathcal{V}(\mathcal{O}^i), e(\mathcal{I}^i)) \in \mathcal{A}$, where the action space $\mathcal{A}$ consists of four low-level navigation actions: $\mathcal{A} = \{$FORWARD (0.25m), TURN LEFT (15°), TURN RIGHT (15°), STOP$\}$, supporting continuous navigation in embodied environments as in VLN-CE (Krantz et al. (2020)). In real-world dynamic deployments, however, agents inevitably face diverse scenes and environments (e.g., low-light, overexposure, scattering), which severely degrade performance (Yang et al. (2025a); Solmaz et al. (2024)). As illustrated in Figure 2, adapting to a new navigation scenario $S_t$ (defined by a specific scene $S_e$ and environment $E_e$) often causes catastrophic forgetting of previously learned scenarios $\{S_1, S_2, ..., S_{t-1}\}$, thus limiting flexible practical deployment.

**Problem Definition:** To tackle the above challenge, we introduce a new problem setting, all-day multi-scenes lifelong vision-and-language navigation (AML-VLN). In this setting, the agent is required to continually learn a sequence of navigation scenarios while alleviating the forgetting of old navigation scenarios, thereby forming an all-day, multi-scenes universal VLN agent. The VLN *Multi-hierarchical Knowledge* of each scenario $T_i$ includes a specific scene $S_e$ under a specific environment $E_e$. Formally, let $\mathcal{T} = \{T_1, T_2, ..., T_t\}$ denote a sequence of navigation scenarios. Each scenario $T_i$ is defined by a pair $(S, E)$, where the scene set $\mathcal{S} = \{S_1, S_2, ..., S_M\}$ includes $M$ scenes and the environment set $\mathcal{E} = \{E_1, E_2, ..., E_N\}$ includes $N$ environments. The VLN agent $\mathcal{F}$ must learn all tasks $\mathcal{T}$ sequentially, and evaluation is conducted across all scenarios after train-

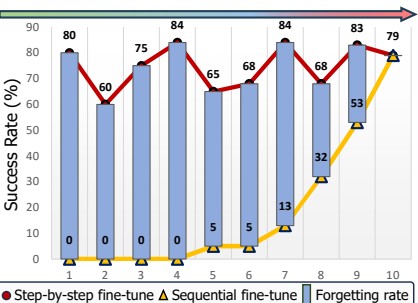

Figure 2: Illustration of catastrophic forgetting in lifelong navigation learning. The new scenario adaptation leads to catastrophic forgetting of old scenarios.

ing. Importantly, in AML-VLN the task-id $t$ is seen during agent training but is agnostic during the testing phase, and each new scenario $\{S_t, E_t\}$ does not overlap with any previous scenario: $\{S_t, E_t\} \bigcap (\bigcup_{j=1}^{t-1} \{S_j, E_j\}) = \emptyset$. A trivial solution is to store all adaptation weights for all past tasks and load them during inference. However, navigation tasks inherently share common and task-specific knowledge: for example, the same scene under different environments (day vs. night), or different scenes under the same environment. Thus, the crucial challenge of AML-VLN is to explore and exploit shared and specific knowledge across multiple scenarios for efficient lifelong learning.

## 3 METHOD

In this section, we first analyze the limitations of the existing low-rank adapters, including vanilla LoRA and MoE LoRA family models (§3.1). Then, we introduce Tucker-Adaption (TuKA) architecture (§3.2), describe how to perform continual learning with TuKA (§3.3) and inference (§3.4).

### 3.1 EXISTING LOW-RANK ADAPTION

Low-Rank Adaptation (LoRA) (Hu et al. (2022)) enables efficient fine-tuning of large language models by injecting low-rank adaptation weights into each transformer layer. Specifically, for the $l$-th layer with backbone weights $\boldsymbol{W}_0^l$, LoRA introduces an update $\Delta \boldsymbol{W}^l = \boldsymbol{B}^l \boldsymbol{A}^l$, where $\boldsymbol{B}^l \in \mathbb{R}^{b_l \times r}$ is a low-rank dimension-raising matrix and $\boldsymbol{A}^l \in \mathbb{R}^{r \times a_l}$ is a low-rank dimension-reducing matrix. As shown in Figure 3(a), the layer output is computed as $y^l = \boldsymbol{W}_0^l x^l + \boldsymbol{B}_t^l \boldsymbol{A}_t^l x^l$. This method learns each task-specific knowledge, making it suitable for specific single-task continual learning. However, its task-specific independent structure limits the ability to explore and reuse shared knowledge across

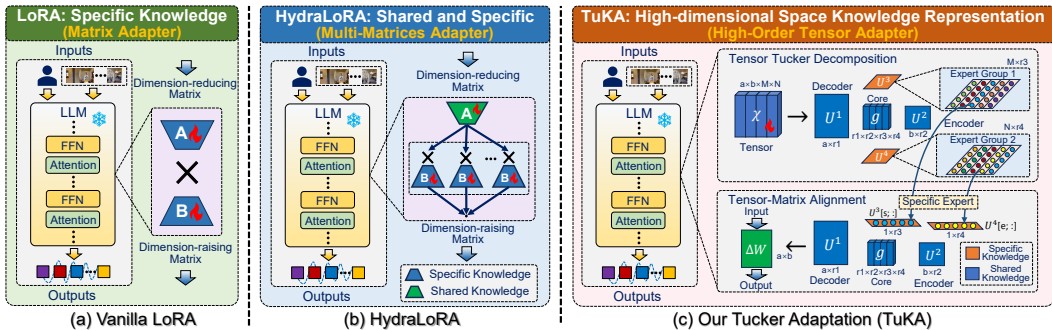

Figure 3: Illustration for comparison of existing LoRAs and our TuKA architecture. Different from the LoRA or MoE-LoRA variants, which represent simple knowledge within a two-hierarchical matrix, TuKA decoupling represents the multi-hierarchical knowledge within a high-order tensor.

multiple tasks. To address this, Mixture-of-Experts (MoE) based methods (Zhang et al. (2025a); Tian et al. (2024); Gao et al. (2024); Chen et al. (2024)) extend LoRA by introducing MoE structures. For example, as Figure 3 (b), HydraLoRA (Tian et al. (2024)) proposes multiple tasks share a single dimension-reducing matrix $\boldsymbol{A}$ with multiple specific dimension-raising matrices $\{\boldsymbol{B}_1, ..., \boldsymbol{B}_K\}$:

$$y^l = \boldsymbol{W}_0^l \cdot x^l + \Delta \boldsymbol{W} \cdot x^l = \boldsymbol{W}_0^l \cdot x + \sum_{n=1}^{K} (\boldsymbol{B}_n^l \cdot \boldsymbol{A}^l \cdot x^l), \tag{1}$$

This design implicitly separates task-shared and task-specific two-hierarchical navigation knowledge. But in our AML-VLN setting, knowledge spans multiple hierarchical levels: core navigation skills, scene-specific knowledge, and environment-specific knowledge. These methods represent all knowledge within two hierarchical matrices: one shared matrix and several task-specific matrices. It restricts them to representing only two hierarchical knowledge structures. This limitation motivates us to explore higher-order representations that can explicitly decouple multi-hierarchical knowledge.

## 3.2 TUCKER-ADAPTION ARCHITECTURE

To achieve high-dimensional space representation with a high-order tensor $\boldsymbol{\mathcal{X}} \in \mathbb{R}^{d_1 \times d_2, ..., \times d_N}$, one of the critical challenges is how to align the higher-order dimensions with the two-dimensional matrix backbone of LLM. A few existing explorations have treated LLM backbones as tensors for learning (Jahromi & Orús (2024)), yet these methods only consider specific architectures, such as splicing multi-attention matrices into third-order tensors within multi-head attention (Zhang et al. (2025d)), or treating matrices as second-order tensors (Bershatsky et al. (2024)). These methods fundamentally fail to resolve the above dimensional alignment problem, and thus do not actually perform representation learning within high-order tensors. To address this, we propose Tucker Adaptation (TuKA), a new fine-tuning method that lifts adaptation into a high-dimensional tensor space. Formally, in order to learn the $t$-th navigation scenario task (with the $s$-th scene and the $e$-th environment) $T_t = \{S_s, E_e\}$, we finetune the StreamVLN agent $\mathcal{F}_{\boldsymbol{\theta}_0}$ (Wei et al. (2025)), in a high-dimensional space, on the task-specific data $S_t = \{\mathcal{O}_t, \mathcal{I}_t\}$, and then obtain an updated $\mathcal{F}_{\boldsymbol{\theta}'_t}$, where $\boldsymbol{\theta}'_t = \boldsymbol{\theta}_0 + \Delta \boldsymbol{\theta}_t$, $\Delta \boldsymbol{\theta}_t = \{\Delta \boldsymbol{W}_t^l\}_{l=1}^{L}$, and $\Delta \boldsymbol{W}_t^l \in \mathbb{R}^{a_l \times b_l}$ is the updated weight in $l$-th layer of a total of $L$ transformer layers. Specifically, in TuKA, we follow the tensor Tucker decomposition (Kolda & Bader (2009)) to decouple a high-order tensor for multi-hierarchical knowledge decoupling representation and dimension alignment. Specifically, a tensor $\boldsymbol{\mathcal{X}}^l \in \mathbb{R}^{a_l \times b_l \times M \times N}$ can be decomposed:

$$\boldsymbol{\mathcal{X}}^l = \boldsymbol{\mathcal{G}} \times_1 \boldsymbol{U}^1 \times_2 \boldsymbol{U}^2 \times_3 \boldsymbol{U}^3 \times_4 \boldsymbol{U}^4, \tag{2}$$

where $\times_n, n = 1, 2, 3, 4$ denotes the $n$-th modal product of the tensor and matrix (Kolda & Bader (2009)). $\boldsymbol{\mathcal{G}} \in \mathbb{R}^{r_1 \times r_2 \times r_3 \times r_4}$ is a core tensor, which contains interaction information between all patterns, and is used to learn the shared core navigation skills. Factor matrix $\boldsymbol{U}^1 \in \mathbb{R}^{a_l \times r_1}$ represents the transformation pattern of the feature from $r_1$ dimension to $a_l$, which can be regarded as a shared decoder; $\boldsymbol{U}^2 \in \mathbb{R}^{b_l \times r_2}$ represents the transformation from $b_l$ dimension to $r_2$, which can be regarded as a shared encoder. Factor matrix $\boldsymbol{U}^3 \in \mathbb{R}^{M \times r_3}$ is the $M$ group of scene experts, with each scene expert $\boldsymbol{U}^3[i, :]$ is used to represent the $i$-th specific scene knowledge; $\boldsymbol{U}^4 \in \mathbb{R}^{N \times r_4}$ is $N$ group of environment experts, with each expert $\boldsymbol{U}^4[j, :]$ is used to represent the $j$-th specific environment knowledge. Thus, for the $t$-th scenario with $s$-th scene and $e$-th environment adaptation, we can

Figure 4: Illustration of the proposed decoupled knowledge incremental learning. Our TuKA performs decoupled incremental learning for multi-hierarchical knowledge in a high-dimensional space.

extract the task-specific $\boldsymbol{U}^3[s,:]$ and $\boldsymbol{U}^4[e,:]$ from tensor $\boldsymbol{\mathcal{X}}$ to constitute adaptation weight $\Delta \boldsymbol{W}_t$:

$$\Delta \boldsymbol{W}_t = \boldsymbol{U}^1 \cdot (\boldsymbol{\mathcal{G}} \times_3 \boldsymbol{U}^3[s,:] \times_4 \boldsymbol{U}^4[e,:]) \cdot (\boldsymbol{U}^2)^T. \tag{3}$$

The TuKA represents a decoupled shared-specific architecture for the multi-hierarchical knowledge (scene $\mathcal{S}$, environment $\mathcal{E}$), in a high-dimensional space $\boldsymbol{\mathcal{X}}$ for effective adaptation, as Figure 3 (c).

## 3.3 DECOUPLED KNOWLEDGE INCREMENTAL LEARNING

To realize a decoupled representation learning for the multi-hierarchical knowledge within a higher-dimensional space $\boldsymbol{\mathcal{X}}$, we propose a Decoupled Knowledge Incremental Learning strategy (DKIL), as illustrated in Figure 4. Specifically, for the initial learning, we use the Kaiming initialization (He et al. (2015)) to initialize the factor matrices and core tensor $\{\boldsymbol{\mathcal{G}}, \{\boldsymbol{U}^i\}_{i=0}^{i=2}\}$ and $\boldsymbol{U}^3, \boldsymbol{U}^4$ with zero-initialization, and then we train only $\{\boldsymbol{\mathcal{G}}, \{\boldsymbol{U}^i\}_{i=1}^{i=2}, \boldsymbol{U}^3[1,:], \boldsymbol{U}^4[1,:]\}$ to adapt scenario $T_1$.

**Inheritance Scenario-Shared Knowledge.** When learning subsequent new scenario task (with $s$-th scene and $e$-th environment) denoted as $T_t = \{S_s, E_e\}, t > 1$ continually, we perform expert knowledge inheritance on the current core tensor $\boldsymbol{\mathcal{G}}$ with the learned $\boldsymbol{\mathcal{G}}'$, and the current encoder decoder $\{\boldsymbol{U}^{1'}, \boldsymbol{U}^{2'}\}$ with the learned $\{\boldsymbol{U}^{1'}, \boldsymbol{U}^{2'}\}$. For previous knowledge inheritance, we also initialize the current scene expert $\boldsymbol{U}^3[s,:]$ or environment expert $\boldsymbol{U}^4[e,:]$ with $\boldsymbol{U}^{3'}[s,:]$ or $\boldsymbol{U}^{4'}[e,:]$ if previous scenario $\{T_i\}_{i=1}^{t-1}$ has learned the same experts. This inheritance mechanism maintains the shared knowledge. In addition, to progressively refine the shared knowledge and avoid old knowledge catastrophic forgetting, we also perform elastic weight consolidation on these shared subspaces:

$$\mathcal{L}_{ewc,t} = \lambda_1 (||F_{\boldsymbol{\mathcal{G}},t-1} \odot (\boldsymbol{\mathcal{G}} - \boldsymbol{\mathcal{G}}')||_F^2 + ||F_{\boldsymbol{U}^1,t-1} \odot (\boldsymbol{U}^1 - \boldsymbol{U}^{1'})||_F^2 + ||F_{\boldsymbol{U}^2,t-1} \odot (\boldsymbol{U}^2 - \boldsymbol{U}^{2'})||_F^2), \tag{4}$$

where $\lambda_1$ is the balance hyper-parameter, $\odot$ denoted as the Hadamard product, $F_{\boldsymbol{\mathcal{G}},t-1} \in \mathbb{R}^{r_1 \times r_2 \times r_3 \times r_4}$, $F_{\boldsymbol{U}^1,t-1} \in \mathbb{R}^{a \times r_1}$, and $F_{\boldsymbol{U}^2,t-1} \in \mathbb{R}^{b \times r_2}$ are Fisher information weights (Kirkpatrick et al. (2017)) measuring the importance of each learnable parameter in $T_{t-1}$, and can be calculated:

$$F_{\boldsymbol{\theta},t-1} = \mathbb{E}_{(S_{t-1},\mathcal{Y}) \sim T_{t-1}} \left[ \left( \partial_{\boldsymbol{\theta}_{t-1}} \log p(\mathcal{Y} \mid S_{t-1}; \boldsymbol{\theta}^{t-1}) \right)^2 \right], \tag{5}$$

where $\{\boldsymbol{\mathcal{G}}, \boldsymbol{U}^1, \boldsymbol{U}^2\} \subseteq \boldsymbol{\theta}$, and $S_{t-1} = \{\mathcal{O}_{t-1}, \mathcal{I}_{t-1}\}$ are the input data and $\mathcal{Y}$ is the output navigation actions. It measures the sensitivity of each parameter $\boldsymbol{\theta}$ to the model's output probability, with a higher value indicating greater importance. In addition, we also perform incremental updates to the $t$-th Fisher $F_t$ during the continual navigation learning to gradually learn the shared knowledge:

$$F_{\boldsymbol{\theta},t} = \omega \cdot F_{\boldsymbol{\theta},t-1} + (1 - \omega) \cdot F_{\boldsymbol{\theta},t}, \tag{6}$$

where $\omega$ is the exponential moving average coefficient to control the smooth update of Fisher $F_{\boldsymbol{\theta},t}$.

In addition to the shared knowledge learning, including the core tensor and the encoder decoder $\{\boldsymbol{\mathcal{G}}, \boldsymbol{U}^1, \boldsymbol{U}^2\}$, to avoid catastrophic forgetting of scene expert knowledge $\boldsymbol{U}^3[s,:]$ and environment expert knowledge $\boldsymbol{U}^4[e,:]$, we also perform expert consistency constraints, and the consistency loss:

$$\mathcal{L}_{co} = \lambda_2 (\alpha \cdot ||\boldsymbol{U}^3[s,:] - \boldsymbol{U}^{3'}[s,:]||_F^2 + \beta \cdot ||\boldsymbol{U}^4[e,:] - \boldsymbol{U}^{4'}[e,:]||_F^2), \tag{7}$$

where $\alpha = 1$ if the $s$-th scene has been learned in the previous scenario $\{T_i\}_{i=1}^{t-1}$; and $\beta = 1$ if the $e$-th environment also has been learned before; and $\lambda_2$ is the consistency balance hyper-parameter.

**Exploration Scenario-Specific Knowledge.** As illustrated in Figure 4, when learning subsequent scenario task $T_t = \{S_s, E_e\}, t > 1$ continually, we freeze the experts $\{\{\boldsymbol{U}^3[i,:]\}_{i \neq s}, \{\boldsymbol{U}^4[j,:]\}_{j \neq e}\}$ to keep previous expert knowledge intact, and only train the specific expert $\{\boldsymbol{U}^3[s,:], \boldsymbol{U}^4[e,:]\}$ in-

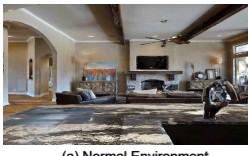 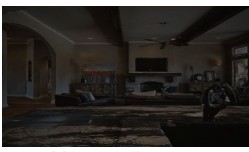 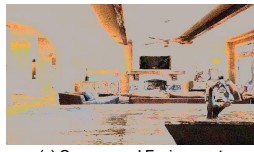 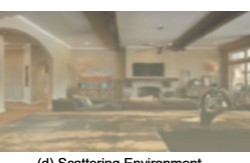

(a) Normal Environment     (b) Low-light Environment     (c) Overexposed Environment     (d) Scattering Environment

Figure 5: Illustration of navigation scenario examples of the Allday-Habitat simulation platform. It includes four common environments: (a) normal, (b) low-light, (c) overexposure, and (d) scattering.

crementally to learn the task-specific decoupled knowledge. To learn new task-knowledge more effectively and independently, we perform the orthogonal optimization on the scene expert $\boldsymbol{U}^3[s,:]$ or environment expert $\boldsymbol{U}^4[e,:]$. Specifically, during adapting the $t$-th scenario task, we prefer that the scene expert $\boldsymbol{U}^3[s,:]$ or the environment expert $\boldsymbol{U}^4[e,:]$ be orthogonal to the previous scene experts $\sum_{i=1}^{i\neq s}(\boldsymbol{U}^3[i,:]\cdot\boldsymbol{U}^3[s,:])=0$ or environment experts $\sum_{j=1}^{j\neq e}(\boldsymbol{U}^4[j,:]\cdot\boldsymbol{U}^4[e,:])=0$, to learn specific knowledge more thoroughly. Thus, the task-specific expert subspace orthogonal constraint is:

$$\mathcal{L}_{es}=\lambda_3((1-\alpha)\cdot||\hat{\boldsymbol{U}}^3(\hat{\boldsymbol{U}}^3)^T-I||_F^2+(1-\beta)\cdot||\hat{\boldsymbol{U}}^4(\hat{\boldsymbol{U}}^4)^T-I||_F^2),\hat{\boldsymbol{U}}^3=Norm(\boldsymbol{U}^3),\hat{\boldsymbol{U}}^4=Norm(\boldsymbol{U}^4), \quad (8)$$

where $Norm(\boldsymbol{U})=(\boldsymbol{U}[i,:])/||\boldsymbol{U}[i,:]||_F^2,(i=1,...,m)$ denotes the normalization of each row for $\boldsymbol{U}$ to have unit Euclidean norm, $\lambda_3$ is the balance hyper-parameter for orthogonal constraint $\mathcal{L}_{es}$.

In summary, during the lifelong navigation learning process for task $T_t$, the adaptation loss for the LLM-based navigation agent $\mathcal{F}$ performing auto-regressive action generation training is as follows:

$$\mathcal{L}_t = -\lambda \sum_{n=1}^{N} \log p_t(\mathcal{A}_n, \hat{\mathcal{P}}_{n|\mathcal{I},\mathcal{O}_t}) + \mathcal{L}_{sk} + \mathcal{L}_{co} + \mathcal{L}_{es}, \quad (9)$$

where $p_t(\mathcal{A}_n, \hat{\mathcal{P}}_{n|\mathcal{I},\mathcal{O}})$ denotes the predicted probability with $n$-th annotation action under agent's current observation $\mathcal{IO}_t = \{\mathcal{I}_t, \mathcal{O}_t\}$, and the balance hyper-parameter is $\lambda = 1 - (\lambda_1 + \lambda_2 + \lambda_3)$.

### 3.4 TASK-SPECIFIC EXPERTS SEARCH

To accurately invoke scene expert and environment expert during inference, we store retrieval features based on the CLIP vision encoder $\mathcal{V}(\mathcal{O})$ for each scene $\mathcal{S}$ and each environment $\mathcal{E}$ during the continual training phase. Specifically, during training, we store the vision features $Fe_s = \mathcal{V}(\mathcal{O}_s)$ for each scene to form a scene feature set $\{Fe_{s1}, Fe_{s2}, ..., Fe_{sM}\}$, and the vision features $Fe_e = \mathcal{V}(\mathcal{O}_e)$ for each environment to form a environment feature set $\{Fe_{e1}, Fe_{e2}, ..., Fe_{eN}\}$. During inference in an unknown navigation scenario $S_q$, we perform a two-step matching to determine the selection of the specific scene expert and the specific environment expert. Specifically, we extract the vision features $Fe_q = \mathcal{V}(\mathcal{O}_q)$ of the agent's observation $\mathcal{O}_q$ in the unknown scenario $S_q$. Then we match the scene expert $\boldsymbol{U}^3[s,:]$, where $s = \arg\max Sim(Fe_q, \{Fe_{s1}, ..., Fe_{sM}\})$, and we also match the environment expert $\boldsymbol{U}^4[e,:]$, where $e = \arg\max Sim(Fe_q, \{Fe_{e1}, ..., Fe_{eM}\})$, and the $Sim(\cdot, \cdot)$ denotes the cosine similarity between the input element and each element of the input set.

## 4   ALLDAY-HABITAT SIMULATION PLATFORM

To train and evaluate the proposed AML-VLN task, we extend the embodied AI simulation platform Habitat (Savva et al. (2019)) by expanding its simulation scenarios from a single normal environment to diverse degraded environments, including normal, scattering, low-light, and overexposure conditions. We synthesize the degraded environment from the normal environment based on three imaging models. Specifically, **i)** to synthesize scattering environments, we perform degradation synthesis based on the atmospheric scattering model (Narasimhan & Nayar (2000); Wang et al. (2024c)). The model is used to describe the imaging process in a scattering environment, which can be expressed:

$$I(x_i) = J(x_i)e^{-\beta d(x_i)} + A(1 - e^{-\beta d(x_i)}), \quad (10)$$

where $I(x_i)$ denotes the pixel value of pixel point $x_i$ in a degraded scattering image, and $J(x_i)$ denotes the clear image in normal environment. $t(x_i) = e^{-\beta d(x_i)}$ denotes the medium transmission map, where $d(x_i)$ is the scene depth and $\beta$ is the scattering density coefficient. $A$ is the global atmospheric light. **ii)** Moreover, to synthesize the low-light environments, referring to the abnormal

light imaging models (e.g., Healey & Kondepudy (2002); Wang et al. (2024d); Cao et al. (2023); Wang et al. (2023a)), our formation model for a low-light degradation can be expressed as follows:

$$I(x_i) = \text{CRF}(S(x_i) + N(x_i)), \ S(x_i) = G \cdot T \cdot L(x_i), \ N(x_i) = N_{\text{shot}}(x_i) + N_{\text{read}}(x_i), \quad (11)$$

where $I(x_i)$ denotes the pixel value at location $x_i$, and $\text{CRF}(i) = i^{\gamma}$ is the camera response function, which introduces a nonlinear Gamma mapping from the sensor irradiance to the digital output. And the signal term $S(x)$ consists of the system gain $G$ that converts photoelectrons into digital units, scene irradiance $L(x)$, and exposure time $T$. And the noise term $N(x)$ consists of the photon shot noise $N_{\text{shot}}(x)$ (Poisson distributed, with variance proportional to the signal intensity), the read-out noise $N_{\text{read}}(x)$ (Gaussian distributed, signal-independent). **iii)** To synthesize the overexposure environments, our formation model for an overexposure degradation can be expressed as follows:

$$I(x_i) = \text{CRF}\left(\text{clip}\left(G \cdot T \cdot L(x_i) + N_{\text{shot}}(x_i) + N_{\text{read}}(x_i), \ 0, \ S_{\text{Sat}}\right)\right), \quad (12)$$

where $L(x_i)$ is the scene irradiance, Sat denotes the sensor saturation level, and $\text{clip}(\cdot, 0, S_{\text{Sat}})$ restricts the signal within the valid dynamic range. Based on the aforementioned three imaging models, we synthesize three degraded scenarios, with examples shown in the Figure 5. And the specific parameters and implementation details of the degradation models can be found in our Appendix §E.

## 5 EXPERIMENTS

### 5.1 IMPLEMENTATION DETAILS

**The Proposed All-day Multi-scenes Lifelong VLN Benchmark Settings:** As described in the Allday-Habitat (§ 4) and our Problem Definition (§ 2), we construct a multi-hierarchical navigation task settings to evaluate our proposed AlldayWalker. Specifically, the proposed benchmark as shown in Figure 6, consists of 24 hierarchical navigation task scenarios. These scenarios include five distinct simulation scenes, each containing four environments: normal, low light, overexposure, and scattering. And the scenarios also include two real-world scenes,

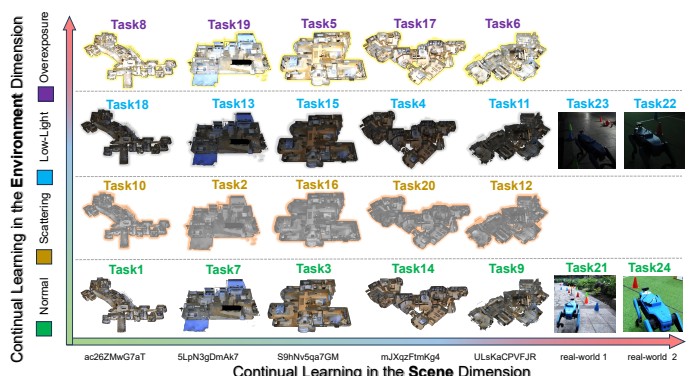

Figure 6: Illustration of our all-day multi-scenes lifelong VLN Benchmark. Agents are required to perform continual learning across two dimensions: scene and environment. The order of tasks is randomized. Further details please refer to Appendix §E.

each containing four environments: normal, low light. All the scenarios are trained sequentially before proceeding to navigation inference. Please note that for more tailored practical real-world navigation applications, the task-id $t$ is seen during training but is agnostic during the testing phase.

**The Used Training and Evaluation Settings:** For fair comparisons, both our AlldayWalker and all comparison methods are implemented on the StreamVLN agent (Wei et al. (2025)). Training and evaluation are conducted on eight NVIDIA RTX 6000 Ada GPUs using PyTorch 2.1.2. We use the Adam optimizer with an initial learning rate of $1.0 \times 10^{-4}$. The low-rank settings of our TuKA are $r_1 = r_2 = 8, r_3 = 64, r_4 = 64$, with the number of experts set to $M = 7, N = 4$. $\lambda_1 = 0.2, \lambda_2 = 0.2, \lambda_3 = 0.1$ and $\omega = 0.95$. All other hyperparameters of VLN agent follow the StreamVLN settings. All hyperparameters are summarized in our Appendix C. Following (Zheng et al. (2024); Wei et al. (2025); Anderson et al. (2018)), we report three standard evaluation metrics: success rate (SR), success rate weighted by path length (SPL), and oracle success rate (OSR). To further assess resistance to forgetting in lifelong learning, we introduce three additional measures: SR forgetting rate ($F$-$SR$), SPL forgetting rate ($F$-$SPL$), and OSR forgetting rate ($F$-$OSR$), and they are defined:

$$F\text{-}SR_t = \frac{M\text{-}SR_t - SR_t}{M\text{-}SR_t}, F\text{-}SPL_t = \frac{M\text{-}SPL_t - SPL_t}{M\text{-}SPL_t}, F\text{-}OSR_t = \frac{M\text{-}OSR_t - OSR_t}{M\text{-}OSR_t}, \quad (13)$$

Table 1: Test Results (**SR** ↑ in %) of Comparison Experiments under the AML-VLN Settings.

| Comparisons | T1 | T2 | T3 | T4 | T5 | T6 | T7 | T8 | T9 | T10 | T11 | T12 | T13 | T14 | T15 | T16 | T17 | T18 | T19 | T20 | T21 | T22 | T23 | T24 | Avg. |
|---|---|---|---|---|---|---|---|---|---|---|---|---|---|---|---|---|---|---|---|---|---|---|---|---|---|
| Seq-FT | 0 | 0 | 0 | 0 | 0 | 0 | 5 | 0 | 10 | 5 | 4 | 10 | 7 | 6 | 7 | 14 | 7 | 8 | 4 | 24 | 28 | 24 | 37 | 64 | 11 |
| LwF-LoRA | 5 | 0 | 5 | 0 | 0 | 4 | 3 | 13 | 5 | 2 | 14 | 8 | 4 | 8 | 14 | 10 | 9 | 5 |  | 25 | 31 | 28 | 40 | 65 | 12 |
| EWC-LoRA | 10 | 7 | 6 | 8 | 3 | 4 | 10 | 4 | 15 | 14 | 5 | 14 | 10 | 6 | 12 | 19 | 16 | 14 | 8 | 24 | 28 | 29 | 38 | 67 | 15 |
| Dense MoLE | 14 | 8 | 14 | 21 | 13 | 12 | 18 | 7 | 19 | 14 | 17 | 19 | 13 | 10 | 18 | 20 | 23 | 17 | 10 | 29 | 32 | 33 | 41 | 66 | 20 |
| Sparse MoLE | 29 | 10 | 24 | 29 | 17 | 28 | 29 | 11 | 37 | 24 | 23 | 30 | 18 | 24 | 25 | 34 | 31 | 10 | 14 | 38 | 36 | 36 | 45 | 69 | 28 |
| MoLA | 34 | 13 | 32 | 34 | 22 | 21 | 37 | 13 | 48 | 28 | 29 | 33 | 24 | 32 | 29 | 39 | 37 | 22 | 14 | 43 | 41 | 39 | 49 | 68 | 33 |
| HydraLoRA | 45 | 12 | 39 | 43 | 24 | 33 | 34 | 29 | 57 | 38 | 33 | 34 | 29 | 37 | 32 | 42 | 38 | 28 | 19 | 52 | 46 | 42 | 48 | 70 | 38 |
| BranchLoRA | 52 | 16 | 43 | 51 | 26 | 39 | 57 | 21 | 65 | 39 | 46 | 43 | 33 | 53 | 37 | 57 | 43 | 33 | 24 | 62 | 52 | 46 | 51 | 69 | 44 |
| O-LoRA | 67 | 19 | 47 | 58 | 31 | 42 | 67 | 27 | 67 | 62 | 58 | 49 | 38 | 68 | 52 | 62 | 46 | 52 | 34 | 71 | 59 | 49 | 53 | 71 | 52 |
| SD-LoRA | 68 | 22 | 52 | 63 | 32 | 48 | 71 | 28 | 71 | 74 | 63 | 62 | 42 | 75 | 56 | 72 | 50 | 49 | 36 | 69 | 64 | 52 | 55 | 69 | 56 |
| FSTTA | 52 | 18 | 46 | 55 | 24 | 35 | 58 | 26 | 51 | 48 | 51 | 43 | 34 | 56 | 46 | 50 | 44 | 41 | 29 | 52 | 45 | 38 | 41 | 47 | 44 |
| FeedTTA | 58 | 19 | 53 | 62 | 27 | 41 | 65 | 30 | 59 | 56 | 59 | 50 | 39 | 64 | 54 | 58 | 51 | 48 | 34 | 61 | 52 | 45 | 48 | 55 | 50 |
| **AlldayWalker** | 79 | 23 | 71 | 81 | 33 | 50 | 87 | 38 | 79 | 75 | 79 | 67 | 50 | 86 | 71 | 76 | 67 | 63 | 43 | 81 | 68 | 58 | 62 | 72 | 65 |

Table 2: Test Results (**F-SR** ↓ in %) of Comparison Experiments under the AML-VLN Settings.

| Comparisons | T1 | T2 | T3 | T4 | T5 | T6 | T7 | T8 | T9 | T10 | T11 | T12 | T13 | T14 | T15 | T16 | T17 | T18 | T19 | T20 | T21 | T22 | T23 | T24 | Avg. |
|---|---|---|---|---|---|---|---|---|---|---|---|---|---|---|---|---|---|---|---|---|---|---|---|---|---|
| Seq-FT | 100 | 100 | 100 | 100 | 100 | 100 | 93 | 100 | 88 | 93 | 94 | 83 | 91 | 92 | 92 | 83 | 87 | 89 | 94 | 73 | 84 | 81 | 78 | 0 | 87 |
| LwF-LoRA | 92 | 100 | 91 | 100 | 100 | 100 | 92 | 89 | 81 | 93 | 95 | 78 | 88 | 93 | 89 | 83 | 85 | 84 | 93 | 71 | 82 | 76 | 73 | 0 | 84 |
| EWC-LoRA | 85 | 92 | 89 | 90 | 94 | 91 | 86 | 87 | 82 | 82 | 92 | 78 | 84 | 78 | 86 | 77 | 75 | 75 | 88 | 72 | 77 | 68 | 66 | 0 | 79 |
| Dense MoLE | 80 | 91 | 82 | 73 | 78 | 78 | 81 | 86 | 77 | 81 | 78 | 72 | 73 | 65 | 79 | 72 | 63 | 69 | 85 | 63 | 74 | 64 | 61 | 0 | 72 |
| Sparse MoLE | 68 | 89 | 69 | 64 | 72 | 49 | 58 | 82 | 56 | 68 | 64 | 56 | 68 | 51 | 70 | 58 | 51 | 82 | 82 | 54 | 63 | 59 | 56 | 0 | 62 |
| MoLA | 62 | 79 | 59 | 58 | 63 | 62 | 46 | 78 | 43 | 63 | 55 | 42 | 65 | 48 | 68 | 54 | 46 | 60 | 79 | 44 | 55 | 51 | 43 | 0 | 55 |
| HydraLoRA | 45 | 64 | 51 | 46 | 60 | 40 | 26 | 67 | 31 | 49 | 49 | 39 | 61 | 44 | 63 | 50 | 44 | 49 | 72 | 29 | 46 | 49 | 38 | 0 | 46 |
| BranchLoRA | 34 | 59 | 46 | 36 | 53 | 29 | 13 | 52 | 29 | 48 | 29 | 23 | 58 | 18 | 54 | 32 | 36 | 42 | 56 | 12 | 36 | 36 | 29 | 0 | 36 |
| O-LoRA | 19 | 41 | 41 | 28 | 38 | 24 | 4 | 38 | 15 | 17 | 11 | 13 | 43 | 2 | 40 | 24 | 32 | 5 | 39 | 2 | 34 | 28 | 20 | 0 | 23 |
| SD-LoRA | 12 | 38 | 34 | 13 | 34 | 13 | 4 | 36 | 12 | 1 | 3 | 8 | 37 | -2 | 36 | 12 | 26 | 13 | 37 | 3 | 28 | 21 | 17 | 0 | 18 |
| **AlldayWalker** | 2 | 27 | 23 | 8 | 21 | 1 | 0 | 28 | 9 | 0 | 2 | 6 | 30 | -3 | 24 | 10 | 1 | 4 | 24 | -4 | 18 | 24 | 12 | 0 | 11 |

where $M\text{-}SR_t$, $M\text{-}SPL_t$, and $M\text{-}OSR_t$ denote the performance (SR, SPL, OSR) obtained when training solely on navigation tasks 1 through $t$, i.e., training on $\{T_1, T_2, ..., T_t\}, t \leq 20$. Thus larger values of $F\text{-}SR_t$, $F\text{-}SPL_t$, and $F\text{-}OSR_t$ indicate a higher degree of forgetting in the $t$-th task.

## 5.2 COMPARISON EXPERIMENT RESULTS

This experiment evaluates the navigation capability of our proposed AlldayWalker. We compare it with a range of state-of-the-art LoRA-based continual learning methods: Seq-FT denotes sequential fine-tuning over all tasks; LwF-LoRA (Li & Hoiem (2017)) applies knowledge distillation to retain prior knowledge; EWC-LoRA (Xiang et al. (2023)) regularizes parameters critical to past tasks to mitigate forgetting; Dense MoLE (Chen et al. (2024)) adopts dense expert routing, while Sparse MoLE (Dou et al. (2024)) introduces sparse expert routing within MoE-LoRA; MoLA (Gao et al. (2024)) enhances Sparse MoLE by incorporating deeper expert hierarchies; O-LoRA (Wang et al. (2023c)) leverages orthogonal loss to disentangle task-specific features; HydraLoRA (Tian et al. (2024)) shares a global module **A** for common knowledge while employing multiple **B** modules for task-specialization; BranchLoRA (Zhang et al. (2025a)) further strengthens the sparse routing mechanism; and SD-LoRA (Wu et al. (2025)) adaptively composes LoRA modules from previously learned skills. To keep the number of trainable parameters comparable across comparison methods, the task-specific LoRA uses a rank of $r = 6$, and MoE-LoRA applies $r = 16$ with $K = 8$ experts, whereas MoE-LoRA shared **A** applies $r = 32$ with $K = 8$ experts. The implementation details and methods parameter comparison are provided in Appendix C. We also compare with FSTTA (Gao et al. (2023)) and FeedTTA (Kim et al.), which aim to perform small, temporary adaptation during test time to adapt the agent to distribution shifts in a single new scene. The results on SR and F-SR are presented in Table 1, 2. And results on SPL, F-SPL, OSR, and F-OSR are presented in Figure 7. Based on the results, our AlldayWalker achieves consistent superiority across various metrics.

## 5.3 ABLATION ANALYSIS

We provide ablation analysis to validate the effectiveness of TuKA. Unless otherwise specified, all ablations are trained on the 20 simulation tasks and evaluated on the same ID-unseen 20 tasks.

**Does the third-order tensor can well represent the multi-hierarchical knowledge?** Our Allday-Walker uses a fourth-order tensor based TuKA to represent multi-hierarchical navigation knowledge (i.e., scenes and environments). In this section, we explore whether a third-order tensor are

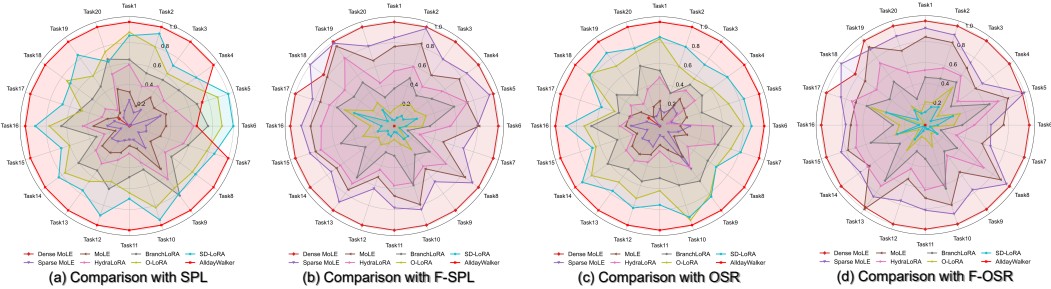

Figure 7: Test results ((a) **SPL** ↑, (b) **F-SPL** ↓, (c) **OSR** ↑, (d) **F-OSR** ↓) of comparison experiment under the AML-VLN settings. For detailed quantitative results, please refer to our Appendix § K.

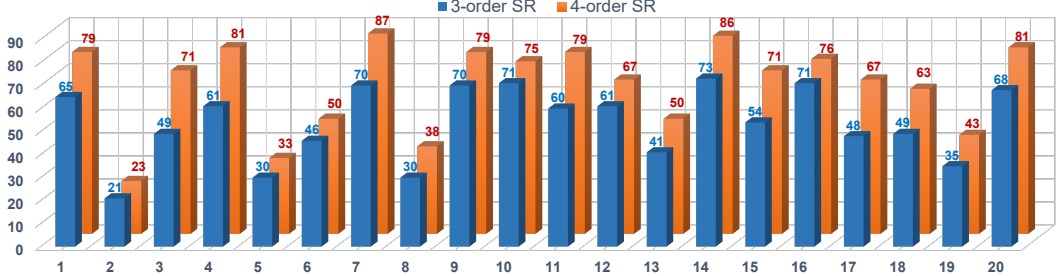

Figure 8: The comparisons of third-order tensors for representation multi-hierarchical knowledge.

sufficient to represent multi-hierarchical knowledge. Specifically, we construct third-order tensors $\boldsymbol{\mathcal{X}}^l \in \mathbb{R}^{a_l \times b_l \times (M \times N)}$ and perform a Tucker decomposition on it. We employ $\boldsymbol{U}^3 \in \mathbb{R}^{(M \times N) \times r_3}$ as the navigation scenario expert, following the DKIL strategy to learn task-specific knowledge with each row. For a more detailed description, please refer to our Appendix § H. We perform continual learning across the 20 simulation tasks using the same training method as for fourth-order tensors ( a total of 20 rows of experts, $\boldsymbol{U}^3 \in \mathbb{R}^{20 \times 128}$). The ablation results are summarized in Figure H. Based on the results, fourth-order tensors achieve superior performance across all 20 tasks. This suggests that, compared to third-order tensors which represent multi-hierarchical knowledge through a coupled expert set structure, fourth-order tensors employ a decoupled representation of multi-hierarchical knowledge. This natural structure facilitates both shared and task-specific knowledge learning. We also explore more hierarchical knowledge representations with fifth-order tensors in Appendix §J.

**Why share the core tensor, encoder and decoder?** We explore the effects of all tasks with shared core tensor $\boldsymbol{\mathcal{G}}$, encoder $\boldsymbol{U}^2$, and decoder $\boldsymbol{U}^1$, the ablation results are summarized in Table 3. Specifically, the "Sd-$\boldsymbol{\mathcal{G}}$" denotes the stored TuKA has only a core tensor $\boldsymbol{\mathcal{G}}$; If without "Sd-$\boldsymbol{\mathcal{G}}$", denotes store specific "Sd-$\boldsymbol{\mathcal{G}}_t$" for each task $T_t$. Similarly, "Sd-$\boldsymbol{U}^2$" denotes the

Table 3: Ablation Results (%) for Shared Components.

| Sd-$\mathcal{G}$ | Sd-$U^1$ | Sd-$U^2$ | SR | F-SR | SPL | F-SPL | OSR | F-OSR |
|---|---|---|---|---|---|---|---|---|
| ✗ | ✗ | ✗ | 53 | **10** | 47 | **17** | 56 | **8** |
| ✗ | ✓ | ✓ | 55 | **10** | 49 | **17** | 57 | 9 |
| ✓ | ✗ | ✓ | **65** | 11 | **58** | 18 | **69** | 9 |
| ✓ | ✓ | ✗ | 62 | 11 | 54 | 18 | 66 | 9 |
| ✓ | ✗ | ✗ | 63 | 11 | 55 | 18 | 67 | 9 |
| ✓ | ✓ | ✓ | **65** | 11 | **58** | 18 | 68 | 9 |

stored TuKA has only a shared encoder $\boldsymbol{U}^2$, "Sd-$\boldsymbol{U}^1$" denotes the stored TuKA has only a shared decoder $\boldsymbol{U}^1$. Based on the results (w/o Sd-$\boldsymbol{U}^2$, w/o Sd-$\boldsymbol{\mathcal{G}}$), both the shared core tensor and encoder between tasks contribute to representing shared knowledge across tasks, thereby improving lifelong navigation performance. Although shared the decoder do not provide a noticeable performance improvement (w/o Sd-$\boldsymbol{U}^1$), it is shared to contribute to the integrity of tensor representation, and significantly reduces storage consumption (no multi-decoders stored), thus our TuKA shared $\boldsymbol{U}^1$.

**Does continual learning for more tasks lead to catastrophic forgetting?** Our 24 tasks already cover diverse scenes and four distinct imaging conditions across both simulation and real-world environments, forming a sufficiently challenging lifelong learning benchmark. To further validate the continual learning performance of AlldayWalker when dealing with more tasks. We also conduct additional continual learning experiments by adding two new real-world tasks and four simulation tasks. For additional task scenes and environments, refer to Appendix Table 22. The results are

Table 4: Comparison of SR (%) under 24-task and 30-task learning.

| Tasks | T1 | T2 | T3 | T4 | T5 | T6 | T7 | T8 | T9 | T10 | T11 | T12 | T13 | T14 | T15 |
|---|---|---|---|---|---|---|---|---|---|---|---|---|---|---|---|
| SR (24 tasks) | 79 | 23 | 71 | 81 | 33 | 50 | 87 | 38 | 79 | 75 | 79 | 67 | 50 | 86 | 71 |
| SR (30 tasks) | 77 | 23 | 71 | 70 | 33 | 50 | 86 | 38 | 79 | 75 | 80 | 66 | 50 | 85 | 70 |

| Tasks | T16 | T17 | T18 | T19 | T20 | T21 | T22 | T23 | T24 | T25 | T26 | T27 | T28 | T29 | T30 |
|---|---|---|---|---|---|---|---|---|---|---|---|---|---|---|---|
| SR (24 tasks) | 76 | 67 | 63 | 43 | 81 | 68 | 58 | 62 | 72 | – | – | – | – | – | – |
| SR (30 tasks) | 77 | 68 | 63 | 44 | 81 | 68 | 58 | 62 | 66 | 72 | 72 | 76 | 35 | 67 | 61 |

Table 5: Generalization performance SR (%) across unseen environments (G1–G6).

| Task | Scene | Environment | Test Number | StreamVLN SR | BranchLoRA SR | SD-LoRA SR | AlldayWalker SR |
|---|---|---|---|---|---|---|---|
| G1 | JeFG25nYj2p | Normal | 105 | 45 | 52 | 53 | **65** |
| G2 | ur6pFq6Qu1A | Low-light | 120 | 41 | 44 | 45 | **63** |
| G3 | r47D5H71a5s | Scattering | 105 | 36 | 40 | 41 | **54** |
| G4 | Vvot9Ly1tCj | Overexposed | 108 | 31 | 42 | 39 | **51** |
| G5 | Real-World 4 | Normal | 100 | 36 | 38 | 37 | **55** |
| G6 | Real-World 5 | Low-light | 100 | 18 | 21 | 21 | **43** |
| Avg.SR | – | – | – | 35 | 40 | 39 | **55** |

summarized in Table 4. The results show that incorporating more tasks does not lead to noticeable performance degradation, demonstrating that AlldayWalker remains stable under lifelong learning with more tasks. The visualization of these navigation tasks is shown in Appendix Figures 13-14.

**Can our AlldayWalker achieve generalization to unseen scenarios?** We also explore the proposed AlldayWalker's generalization performance on unseen scenarios compared to other methods. Specifically, we perform six completely unseen tasks for generalization testing. Select the expert with the highest similarity during testing. These six tasks include four simulation scenarios (four distinct scenes with four distinct environments) and two real-world scenarios (two distinct real-world scenes in low-light and normal environments). The details of the unseen task scenarios and results are summarized in Table 5. The results show that our AlldayWalker has superior generalization performance, achieving an average SR of 55%, surpassing SD-LoRA (39%) by 16% and BranchLoRA (40%) by 15%. The visualization of these tasks is shown in Appendix Figures 15-16

We provide more ablation analyses for TuKA scaling, extension, and effect in Appendix § G, I, J.

## 6 CONCLUSION

We formalize the all-day multi-scenes lifelong vision-and-language navigation (AML-VLN) learning problem to study VLN agent lifelong adaptation across multiple scenes and diverse environments. To address AML-VLN, we propose Tucker Adaptation (TuKA), a new parameter-efficient method that represents the multi-hierarchical knowledge in a high-order tensor and uses Tucker decomposition to decouple task-shared and task-specific knowledge. We further propose a decoupled knowledge incremental learning strategy to support multi-hierarchical knowledge continual learning. Based on the proposed TuKA, we also develop a lifelong VLN agent named AlldayWalker, which achieves superior navigation performance compared to the SOTA baselines on the AML-VLN problem, enabling all-day multi-scenes navigation. Our research demonstrates the value of high-order tensor adaptation for continual multi-hierarchical knowledge representation learning.

## 7 ACKNOWLEDGMENTS

This work was supported by the National Natural Science Foundation of China under Grant T2596040, T2596045 and U23A20343, CAS Project for Young Scientists in Basic Research, Grant YSBR-041, Liaoning Provincial "Selecting the Best Candidates by Opening Competition Mechanism" Science and Technology Program under Grant 2023JH1/10400045, Joint Innovation Fund of DICP & SIA under Grant UN202401, Fundamental Research Project of SIA under Grant 2024JC3K01, Natural Science Foundation of Liaoning Province under Grant 2025-BS-0193.

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

APPENDIX

## A    RELATED WORKS

**Vision-and-Language Navigation (VLN).** VLN is an important embodied AI task where agents follow natural language instructions to navigate environments (Wang et al. (2025); Qiao et al. (2025); Zhao et al. (2025); Gao et al. (2025b); Dong et al. (2026); Wei et al. (2025); Zhang et al. (2025c); Han et al. (2026)). Since the introduction of the simulator and Room-to-Room benchmark dataset (Anderson et al. (2018)), research has evolved rapidly. Early works relied on discrete navigation graphs (Ku et al. (2020); Qi et al. (2020)), which was able to reduce the action solution space of the agent but limited transfer to real-world continuous spaces. To address this, the VLN-CE benchmark (Krantz et al. (2020)) extended VLN into continuous environments, enabling fine-grained actions (Wang et al. (2023b)) and waypoint-based planning (Hong et al. (2022); Krantz et al. (2021); Krantz & Lee (2022)). With the emergence of large-scale embodied datasets (Chang et al. (2017); Ramakrishnan et al. (2021)) and advanced simulators (Savva et al. (2019); Kolve et al. (2017); Makoviychuk et al. (2021)), VLN research has shifted toward more realistic, long-horizon navigation settings, and more complex navigation environments. Recent works integrate large pre-trained vision–language models to enhance reasoning capability and generalization capability (Zhang et al. (2024b); Zheng et al. (2024); Zhang et al. (2025b); Gao et al. (2025a); Wei et al. (2025)), while others exploit history modeling (Qiao et al. (2023); Chen et al. (2021); Anwar et al. (2024)) or spatial memory maps (Krantz et al. (2023); Wang et al. (2023e); Hong et al. (2023); An et al. (2023)). And the recent methods NavQ (Xu et al. (2025)), MAGIC (Wang et al. (2024a)), and YouTube-VLN (Lin et al. (2023)) improve VLN policy learning through foresight modeling, meta-ability guidance, or video-driven pretraining. Despite significant progress, existing VLN agents still face challenges in practical deployment: when adapting to new environments (e.g., day/night, scattering conditions), they often suffer from catastrophic forgetting of previously learned scenarios, severely limiting their robustness in dynamic real-world settings. Test-Time Adaptation (TTA) VLN methods (Qiao et al. (2023); Kim et al.; Gao et al. (2023)) aim to perform small, temporary adaptation during test time to adapt the agent to distribution shifts in a single new scene. FeedTTA (Kim et al.) improves plasticity to new scenarios while selectively correcting certain gradients, thereby suppressing drastic parameter drift caused by non-stationarity to reduce forgetting of previous tasks. The strength of these methods is that agents can quickly adapt to new scenarios. However, they do not accumulate knowledge over continual tasks. They do not maintain any explicit scene or environment knowledge after a single episode ends. They are primarily designed for rapid adaptation in short-term scenarios rather than long-term, multi-scenario persistent operations.

**Lifelong and Continual Learning.** Lifelong learning (Li et al. (2025); Yang et al. (2025b); Wang et al. (2024b; 2026b); Dong et al. (2025); Liu et al. (2025); Liang et al. (2024)) seeks to enable models to continually acquire new capabilities without erasing past knowledge. Existing approaches can be grouped into three categories: (i) *regularization-based methods*, which constrain weight updates to protect important past-task parameters (Derakhshani et al. (2021); Douillard et al. (2020); Li & Hoiem (2017)); (ii) *architecture-based methods*, which expand the network or allocate new modules for different tasks (Jung et al. (2020); Toldo & Ozay (2022); Wang et al. (2022); Wu et al. (2021)); and (iii) *replay-based methods*, which rehearse stored or generated past samples (Bang et al. (2021); Li et al. (2022); Rebuffi et al. (2017); Sun et al. (2024; 2022); Wan et al. (2024); Xiang et al. (2019)). While effective in standard benchmarks, these methods face difficulties in embodied tasks where memory replay is costly and task boundaries are often unclear. Recently, parameter-efficient fine-tuning methods such as LoRA (Hu et al. (2022)) have been adapted for continual learning due to their modularity and separability. Several variants, including HydraLoRA (Tian et al. (2024)), BranchLoRA (Zhang et al. (2025a)), MoLA (Gao et al. (2024)), and Sparse/OLoRA (Dou et al. (2024); Wang et al. (2023c)), explore expert-based sharing mechanisms to balance efficiency and adaptability. However, these methods fundamentally rely on two-dimensional low-rank matrices, which struggle to capture the multi-level navigation knowledge (shared action semantics, scene-specific, and environment-specific factors) required in AML-VLN.

**Tensor Decomposition for Representation Learning.** Beyond matrix-based LoRA, tensor methods provide a new framework for modeling multi-hierarchical structures. Classical tensor decomposition methods, such as CP and Tucker decomposition (Kolda & Bader (2009)), have been widely studied in signal processing and multi-way data analysis (Liu et al. (2026)). Recent explorations

in neural networks investigate using tensors to compress parameters (Jahromi & Orús (2024); Bershatsky et al. (2024)) or to model structured attention mechanisms (Zhang et al. (2025d)). FiPS (Üyük et al. (2024)) combines sparse tensor decomposition and parameter sharing to compress and share parameters across layers in ViTs and LLMs, demonstrating that parameter sharing via high-order factorization can maintain performance with heavy compression. MetaTT (Lopez-Piqueres et al. (2025)) uses Tensor-Train decompositions globally across multiple linear sub-modules of transformers, compressing adapters with Tensor-Train structure rather than separate per-layer matrix adapters. However, these methods typically treat specific architectures (e.g., multi-head attention) as tensors rather than aligning high-order tensors with the full LLM backbone. Consequently, they do not fully realize representation learning in higher-order spaces. In contrast, our work leverages Tucker decomposition to explicitly factorize navigation knowledge into shared components and task-specific experts (scene and environment), offering a principled way with multi-hierarchical knowledge for robotic lifelong VLN.

## B  NOTATIONS AND DEFINITIONS

To make it easier for readers, we summarize some important notations and definitions used in this paper in Table 6.

## C  COMPARISONS IMPLEMENTATION

We provide a detailed comparison experimental implementation. For a fair comparison, both our AlldayWalker and all comparison methods are implemented on the StreamVLN baseline (Wei et al. (2025)). Training and evaluation are conducted on eight NVIDIA RTX 6000 Ada GPUs using PyTorch 2.1.2 (cu121). We adopt the Adam optimizer with an initial learning rate of $1.0 \times 10^{-4}$. The low-rank settings of our TuKA are $r_1 = r_2 = 8, r_3 = 64, r_4 = 64$, with the number of experts set to $M = 7, N = 4$. To keep the number of trainable parameters comparable across comparison methods, the task-specific LoRA configuration uses a rank of $r = 6$ with a total of 24 sets of LoRA, the single-LoRA configuration uses a rank of $r = 128$, and MoE-LoRA applies $r = 6$ with $K = 24$ experts, whereas MoE-LoRA shared **A** applies $r = 12$ with $K = 24$ experts. Specifically, for our proposed TuKA, its parameters are calculated as follows:

$$
\begin{aligned}
Param_{TuKA} &= Param_{\mathcal{G}} + Param_{U^1} + Param_{U^2} + Param_{U^3} + Param_{U^4} \\
&= (r_1 \times r_2 \times r_3 \times r_4) + (a \times r_1) + (b \times r_2) + (M \times r_3) + (N \times r_4) \\
&= (8 \times 8 \times 64 \times 64) + (1024 \times 8) + (1024 \times 8) + (7 \times 64) + (4 \times 64) \\
&= 279,232 \approx 0.3M.
\end{aligned} \tag{14}
$$

And for the task-specific LoRA comparison methods (i.e., SD-LoRA, O-LoRA), their parameters are calculated as follows:

$$
\begin{aligned}
Param_{LoRA} &= T \times (Param_{\mathbf{A}} + Param_{\mathbf{B}}) \\
&= T \times ((a \times r) + (b \times r)) \\
&= 24 \times ((1024 \times 6) + (1024 \times 6)) \\
&= 294,912 \approx 0.3M.
\end{aligned} \tag{15}
$$

And for the single-LoRA comparison methods (i.e., LwF-LoRA, EWC-LoRA), their parameters are calculated as follows:

$$
\begin{aligned}
Param_{LoRA} &= Param_{\mathbf{A}} + Param_{\mathbf{B}} \\
&= (a \times r) + (b \times r) \\
&= (1024 \times 128) + (1024 \times 128) \\
&= 262,144 \approx 0.3M.
\end{aligned} \tag{16}
$$

And for the MoE-LoRA comparison methods (i.e., Dense MoLE, Sparse MoLE, MoLA), their parameters are calculated as follows:

$$
\begin{aligned}
Param_{MOE-LoRA} &= K \times (Param_{\mathbf{A}} + Param_{\mathbf{B}}) \\
&= K \times ((a \times r) + (b \times r)) \\
&= 24 \times ((1024 \times 6) + (1024 \times 6)) \\
&= 294,912 \approx 0.3M.
\end{aligned} \tag{17}
$$

Table 6: Some Important Notations and Definitions Used in This Paper.

| Notations | Definitions | Notations | Definitions |
|---|---|---|---|
| $\mathcal{I}$ | The user language instruction | $\odot$ | The Hadamard product |
| $\mathcal{S}$ | Navigation Scene | $F_{\boldsymbol{U}^1,t-1}$ | The Fisher information weights for $\boldsymbol{U}^1$ |
| $\mathcal{E}$ | Scene Environment | $F_{\boldsymbol{U}^2,t-1}$ | The Fisher information weights for $\boldsymbol{U}^2$ |
| $\mathcal{F}$ | The backbone agent | $\mathcal{V}(\mathcal{O}_s)$ | The vision features for each scene |
| $O$ | The video stream observation | $\mathcal{V}(\mathcal{O}_e)$ | The vision features for each environment |
| $\mathcal{V}(\mathcal{O})$ | The CLIP vision encoder | $\mathcal{Y}$ | The output navigation actions annotations |
| $i$ | The navigation step | $\omega$ | The exponential moving coefficient |
| $\mathcal{F}(\mathcal{V}(\mathcal{O}^i))$ | The next action | $F_{\theta,t}$ | The smooth update of Fisher |
| $\mathcal{T}$ | A sequence of navigation taks | $\boldsymbol{W}_0^l$ | The $l$-th layer with backbone weights |
| $\boldsymbol{A}^l$ | A low-rank dimension-reducing matrix | $S_q$ | An unknown navigation scenario |
| $\boldsymbol{B}^l$ | A low-rank dimension-raising matrix | $\mathcal{O}_q$ | The agent's observation |
| $\boldsymbol{\mathcal{X}}$ | A high-order tensor | $J(x_i)$ | The clear image in normal environment |
| $T_t = \{S_s, E_e\}$ | The $t$-th navigation scenario task | $d(x_i)$ | The scene depth |
| $\mathcal{F}_{\boldsymbol{\theta}_0}$ | The base navigation agent | $\beta$ | The scattering density coefficient |
| $\Delta \boldsymbol{W}_t^l$ | The updated weight in $l$-th layer for $T_t$ | $A$ | The global atmospheric light |
| $\boldsymbol{\mathcal{X}}^l$ | A four-order tensor | $\mathrm{CRF}(i) = i^\gamma$ | The camera response function |
| $\mathcal{G}$ | A core tensor | $S(x)$ | The signal term |
| $\boldsymbol{U}^1$ | The factor matrix represents the transformation pattern of the feature from $r_1$ dimension to $a$ | $L(x)$ | The scene irradiance |
| $\boldsymbol{U}^2$ | The factor matrix represents the transformation pattern of the feature from $b$ dimension to $r_2$ | $T$ | The exposure time |
| $\boldsymbol{U}^3$ | The factor matrix represents the $M$ group of scene experts | $N(x)$ | The noise term |
| $U^4$ | The factor matrix represents $N$ group of environment experts | $N_{\mathrm{shot}}(x)$ | The photon shot noise |

And for the MoE-LoRA (i.e., BranchLoRA, HydraLoRA) shared **A** comparison methods, their parameters are calculated as follows:

$$
\begin{aligned}
Param_{MOE-LoRA} &= Param_{\mathbf{A}} + K \times Param_{\mathbf{B}} \\
&= (a \times r) + K \times (b \times r) \\
&= (1024 \times 12) + 24 \times (1024 \times 12) \\
&= 307,200 \approx 0.3M.
\end{aligned}
\tag{18}
$$

Therefore, all parameters across comparison methods are maintained at a consistent order of magnitude and remain comparable, ensuring uniformity in learnable parameters and guaranteeing experimental fairness. All other hyperparameters of the VLN agent follow the original StreamVLN settings, and we summarize some important parameter settings as shown in Table 7. During lifelong learning, to obtain the Fisher Matrix, we compute it using the first 10% of the data before adaptation to each task, using Eq. 5. In line with prior work (Zheng et al. (2024); Wei et al. (2025); Anderson et al. (2018)), we mainly report three standard evaluation metrics: SR, SPL, OSR. To further assess resistance to catastrophic forgetting in lifelong learning, we also introduce three additional

measures: SR Forgetting Rate ($F$-$SR$), SPL Forgetting Rate ($F$-$SPL$), and OSR Forgetting Rate ($F$-$OSR$). The specific calculation is outlined below.

**Navigation Success Rate (SR).** The navigation success rate is a widely adopted metric for evaluating whether an embodied agent successfully reaches its target location. It is defined based on the Euclidean distance between the predicted final position of the agent and the ground-truth goal position. A navigation attempt is considered successful if this distance is smaller than or equal to a predefined threshold $\epsilon$. Formally,

$$SR = \begin{cases} 1, & \text{if } d(T_n^{pred}, T_n^{ref}) \leq \epsilon, \\ 0, & \text{otherwise,} \end{cases} \tag{19}$$

where $T_n^{pred}$ is the predicted endpoint, $T_n^{ref}$ is the reference endpoint, and $d(\cdot, \cdot)$ denotes the geodesic distance. SR provides a binary indicator of task completion quality and reflects the agent's reliability in precisely reaching the goal.

**Oracle Success Rate (OS).** While SR only evaluates the final stopping point, the Oracle Success Rate captures whether the agent's trajectory passes sufficiently close to the goal, regardless of the final decision to stop. This metric therefore measures the feasibility of correcting the navigation path with an ideal stopping policy. It is defined as:

$$OS = \begin{cases} 1, & \text{if } \min_{i \in [1,n]} d(T_i, T_t) \leq \epsilon, \\ 0, & \text{otherwise,} \end{cases} \tag{20}$$

where $T_t$ is the ground-truth target location, and $T_i$ denotes the $i$-th point along the trajectory. Compared to SR, OS relaxes the evaluation by rewarding trajectories that approach the goal, even if the final stopping action is inaccurate.

**Success Rate weighted by Path Length (SPL).** The SPL metric provides a more comprehensive evaluation by jointly considering task success and path efficiency. Specifically, it penalizes trajectories that are unnecessarily long compared to the reference path, thereby encouraging agents to follow efficient routes. Formally,

$$SPL = \frac{\text{Success Rate} \times TL}{TL_{ref}}, \tag{21}$$

where $TL$ is the actual trajectory length executed by the agent and $TL_{ref}$ is the ground-truth shortest trajectory length. A high SPL score requires not only reaching the goal but also doing so along a near-optimal path, making it a stricter and more informative metric than SR alone.

During testing, we perform 100 episode and calculate the average value of the matrices. To evaluate the navigation capability of our proposed AlldayWalker, we compare it with a broad range of state-of-the-art LoRA-based continual learning baselines. Below we summarize their main implementation details:

- **Seq-FT**: Sequential fine-tuning of the base model with a vanilla LoRA on each task without any forgetting prevention mechanism. After training is complete, use the stored vanilla LoRA for inference.

- **LwF-LoRA** (Li & Hoiem (2017)): Extends Learning without Forgetting to vanilla LoRA by applying knowledge distillation to retain prior task knowledge during training on new tasks, and we set the parameters in accordance with the literature (Li & Hoiem (2017)). After training is complete, use the stored vanilla LoRA for inference.

- **EWC-LoRA** (Xiang et al. (2023)): Incorporates Elastic Weight Consolidation into vanilla LoRA, constraining updates of parameters that are estimated to be important for previous tasks, and we set the parameters in accordance with the literature (Xiang et al. (2023)). After training is complete, use the stored vanilla LoRA for inference.

- **Dense MoLE** (Chen et al. (2024)): Uses a dense mixture-of-LoRA-experts where all experts are activated for every task, capturing more cross-task knowledge at the cost of higher computation. After training is complete, use the stored MOE-LoRA for inference.

- **Sparse MoLE** (Dou et al. (2024)): Improves efficiency by sparsely routing inputs to only a subset of LoRA experts for each task, by selecting only the top-$k$ experts per instance ($k = 2$). After training is complete, use the stored MOE-LoRA for inference.

---

**Algorithm 1** Training pipeline of AlldayWalker (TuKA + DKIL)

---

1: Prepare task stream $\mathcal{T} = \{T_1, \ldots, T_T\}$, where $T_t = \{S_t, E_t\}$.
2: Learnable factors: core $\boldsymbol{\mathcal{G}}$, encoder/decoder $\boldsymbol{U}^1, \boldsymbol{U}^2$, scene experts $\boldsymbol{U}^3$, environment experts $U^4$.
3: **for** $t = 1$ **to** $T$ **do**
4:     Get current scenario $(s, e)$ from $T_t$.
5:     **if** $t = 1$ **then**
6:         Initialize $\boldsymbol{\mathcal{G}}, \boldsymbol{U}^1, \boldsymbol{U}^2$ by Kaiming initialization.
7:         Initialize $\boldsymbol{U}^3[s, :], \boldsymbol{U}^4[e, :]$ with zero-initialization.
8:     **else**
9:         Inherit the current shared parameters $\boldsymbol{\mathcal{G}}, \boldsymbol{U}^1, \boldsymbol{U}^2$ from previous tasks learned $\boldsymbol{\mathcal{G}}', \boldsymbol{U}^{1'}, \boldsymbol{U}^{2'}$.
10:         If $s$-th scene or $e$-th environment appeared before, inherit the current $\boldsymbol{U}^3[s, :], \boldsymbol{U}^4[e, :]$ from
        previous $\boldsymbol{U}^{3'}[s, :], \boldsymbol{U}^{4'}[e, :]$.
11:     **end if**
12:     Freeze all non-current experts $\{\boldsymbol{U}^3[i, :]\}_{i \neq s}, \{\boldsymbol{U}^4[j, :]\}_{j \neq e}$.
13:     Compute Fisher Matrix $F_t$ with Eq.5.
14:     Update and save Fisher Matrix $F_t$ with Eq.6.
15:     **for** each minibatch $(\mathcal{O}_t^i, \mathcal{I}_t^i)$ **do**
16:         Reconstruct adapter $\Delta \boldsymbol{W}$ by Tucker decomposition with $(\boldsymbol{\mathcal{G}}, \boldsymbol{U}_1, \boldsymbol{U}_2, \boldsymbol{U}_3[s, :], \boldsymbol{U}_4[e, :])$.
17:         Compute $L_{ewc}$ (shared knowledge consolidation) using Fisher matrices with Eq.4.
18:         Compute $L_{co}$ (expert consistency) with Eq.7 and $L_{es}$ (orthogonality) with Eq.8.
19:         Compute auto-regressive loss $\mathcal{L}_{AR}$ from predicted actions with Eq.9.
20:         Total loss: $L = \mathcal{L}_{AR} + L_{sk} + L_{co} + L_{es}$.
21:         Update trainable parameters $\{\boldsymbol{\mathcal{G}}, \boldsymbol{U}_1, \boldsymbol{U}_2, \boldsymbol{U}_3[s, :], \boldsymbol{U}_4[e, :]\}$ with Adam.
22:     **end for**
23:     Save CLIP features for $(s, e)$ as retrieval keys for inference (§ 3.4).
24: **end for**

---

- **MoLA** (Gao et al. (2024)): Builds on Sparse MoLE by adding deeper hierarchical expert layers, allowing richer task-specific adaptations. We increase the number of deep experts to 16, whilst reducing the number of shallow experts to 4. After training is complete, use the stored MOE-LoRA for inference.

- **O-LoRA** (Wang et al. (2023c)): Stores one task-specific vanilla LoRA module per task and employs an orthogonal loss to encourage disentangled task representations. This method requires maintaining $T$ LoRA modules for $T$ tasks. After training is complete, it employs selection inference based on the proposed expert selection methodology (§ 3.4).

- **HydraLoRA** (Tian et al. (2024)): Shares a global module $\mathbf{A}$ across all tasks to encode common knowledge, while employing multiple task-specific $\mathbf{B}$ modules for specialization. After training is complete, use the stored HydraLoRA for inference.

- **BranchLoRA** (Zhang et al. (2025a)): Enhances sparse expert routing by introducing a branching mechanism, enabling more flexible expert selection across diverse tasks. After training is complete, it employs task-specific $\mathbf{B}$ selection inference based on the proposed expert selection methodology (§ 3.4).

- **SD-LoRA** (Wu et al. (2025)): Stores $T$ task-specific LoRA modules and dynamically composes them during inference, transferring knowledge from previously learned tasks, in accordance with the literature (Wu et al. (2025)). After training is complete, it employs selection inference based on the proposed expert selection methodology (§ 3.4).

## D   ALGORITHM SUMMARY

For ease of readers understanding, a summary of the proposed AlldayWalker learning algorithm is provided in the Algorithm 1, and a summary of inference algorithm is provided in the Algorithm 2.

---

**Algorithm 2** Inference pipeline of AlldayWalker (TuKA)

---

1: Input: instruction $\mathcal{I}_q$, current observation $\mathcal{O}_q$.
2: Encode $\mathcal{O}_q$ with CLIP to get feature $f_q = En(\mathcal{O}_q)$.
3: Match scene expert index $s = \arg\max_i \text{Sim}(f_q, \{Fe_{s1}, Fe_{s2}, ..., Fe_{sM}\})$.
4: Match environment expert index $e = \arg\max_j \text{Sim}(f_q, \{Fe_{e1}, Fe_{e2}, ..., Fe_{eN}\})$.
5: Reconstruct adapter $\Delta\boldsymbol{W}$ using $\mathcal{G}, \boldsymbol{U}_1, \boldsymbol{U}_2, \boldsymbol{U}_3[s,:], \boldsymbol{U}_4[e,:]$.
6: Adapt base model: $\boldsymbol{\theta} = \boldsymbol{\theta}_0 + \Delta\boldsymbol{W}$.
7: Run auto-regressive decoding with $(\mathcal{I}, \mathcal{O})$ to generate navigation actions.
8: Execute actions step by step until the goal is reached.

---

Table 7: Some Important Hyper-parameters in AlldayWalker.

| Hyperparameters Name | Value |
|---|---|
| Number of scene-specific experts $S$ | 7 |
| Number of environment-specific experts $E$ | 4 |
| 4D Tucker decomposition ranks $\mathbf{r} = (r_1, r_2, r_3, r_4)$ | (8,8,64,64) |
| Elastic Weight Consolidation regularization weight $\lambda_1$ | 0.2 |
| Stability constraint weight for previously trained experts $\lambda_2$ | 0.2 |
| Orthogonal regularization weight for $\boldsymbol{U}3/\boldsymbol{U}4 \ \lambda_3$ | 0.1 |
| Number of samples for Fisher matrix computation | 20 |
| Fisher matrix exponential moving coefficient $\omega$ | 0.9 |
| Number of training epochs per task $N_{epochs}$ | 10 |
| Training batch size per device $B_{train}$ | 2 |
| Base learning rate $\eta$ | 1e-4 |
| Number of historical frames $N_{hist}$ | 8 |
| Number of future steps to predict $N_{fut}$ | 4 |
| Number of frames per video $N_{frames}$ | 32 |

## E    ALLDAY-HABITAT SIMULATION PLATFORM

To validate the proposed AML-VLN task, we construct a multi-hierarchical navigation task settings to evaluate our proposed AlldayWalker. Specifically, the proposed benchmark as shown in Figure 6, consists of 20 simulation navigation task scenarios. These simulation scenarios include five distinct scenes, each containing four environments: normal, low light, overexposure, and scattering. All the scenarios are trained sequentially before proceeding to navigation inference. We provide the specific parameters and implementation details of the degradation models (§ 4). We summarize the used parameters in Table 8. And the statistics for the constructed dataset are shown in Table 9.

## F    ROBOTIC NAVIGATION PLATFORM

The robotic navigation platform used in this work is illustrated in Figure 9. The platform consists of a DeepRobotDog Lite2 quadruped robot, a Hikvision DS-E12 camera, a portable WiFi communication module, and a remote computation server equipped with an NVIDIA A6000 GPU. During deployment, the Hikvision DS-E12 mounted on the robot captures RGB visual streams from the real environment, providing essential perception signals for navigation and scene understanding. These visual data are transmitted in real time to the remote server through the portable WiFi module. The server performs inference using the proposed AlldayWalker system running on the A6000 GPU, which processes both user instructions and visual observations to generate navigation actions. The resulting control signals are then sent back to the DeepRobotDog Lite2 via the wireless communication channel. The robot executes these actions in the physical environment, enabling closed-loop interaction between perception, language reasoning, and embodied control. This platform supports flexible and robust experimentation of all-day multi-scenes lifelong navigation, bridging simulation and real-world deployment.

Table 8: Some Important Degradation Imaging Model Parameters.

| Pattern | Hyperparameters Name | Value |
|---|---|---|
| Low-night | Brightness factor $B$ | 0.15 |
| | Exposure time $T$ | 0.15 |
| | Sensor gain / ISO $G$ | 8.0 |
| | Shot noise factor $\alpha_{\text{shot}}$ | 0.4 |
| | Read noise standard deviation $\sigma_{\text{read}}$ | 3.0 |
| | Gamma correction value $\gamma$ | 2.2 |
| | Denoise strength $D_s$ | 0.75 |
| | Detail preservation factor $P_d$ | 0.7 |
| Scattering | Atmospheric scattering coefficient $\beta$ | 0.01 |
| | Atmospheric light intensity (RGB) $A$ | [0.95, 0.95, 1.0] |
| | Maximum effective distance $d_{\max}$ | 200.0 |
| | Particle size $r_p$ | 0.1 |
| Overexposure | Exposure multiplier $T_e$ | 2.5 |
| | Sensor gain $G$ | 1.5 |
| | Sensor saturation value (full-well capacity) $S_{\text{sat}}$ | 0.9 |
| | Read noise standard deviation $\sigma_{\text{read}}$ | 0.015 |
| | Gamma correction value $\gamma$ | 2.0 |
| | Bloom strength $B_s$ | 0.3 |
| | Color shift vector (RGB) $C_{\text{shift}}$ | [1.0, 0.96, 0.92] |

Table 9: Statistics of the 24 Sequential Tasks in Allday-Habitat Benchmark. Each Task Corresponds to One Scene under One Environment Type.

| Task ID | Scene Index Number | Scene ID | Environment ID | Train Number | Test Number |
|---|---|---|---|---|---|
| 1 | ac26ZMwG7aT | Scene-2 | Normal-1 | 455 | 120 |
| 2 | 5LpN3gDmAk7 | Scene-1 | Scattering-2 | 405 | 150 |
| 3 | S9hNv5qa7GM | Scene-4 | Normal-1 | 410 | 105 |
| 4 | mJXqzFtmKg4 | Scene-3 | Low-light-3 | 445 | 105 |
| 5 | mJXqzFtmKg4 | Scene-4 | Overexposure-4 | 410 | 105 |
| 6 | ULsKaCPVFJR | Scene-5 | Overexposure-4 | 390 | 120 |
| 7 | 5LpN3gDmAk7 | Scene-1 | Normal-1 | 405 | 150 |
| 8 | ac26ZMwG7aT | Scene-2 | Overexposure-4 | 455 | 120 |
| 9 | ULsKaCPVFJR | Scene-5 | Normal-1 | 390 | 120 |
| 10 | ac26ZMwG7aT | Scene-2 | Scattering-2 | 455 | 120 |
| 11 | ULsKaCPVFJR | Scene-5 | Low-light-3 | 390 | 120 |
| 12 | ULsKaCPVFJR | Scene-5 | Scattering-2 | 390 | 120 |
| 13 | 5LpN3gDmAk7 | Scene-1 | Low-light-3 | 405 | 150 |
| 14 | mJXqzFtmKg4 | Scene-3 | Normal-1 | 445 | 105 |
| 15 | S9hNv5qa7GM | Scene-4 | Low-light-3 | 410 | 105 |
| 16 | S9hNv5qa7GM | Scene-4 | Scattering-2 | 410 | 105 |
| 17 | mJXqzFtmKg4 | Scene-3 | Overexposure-4 | 445 | 105 |
| 18 | ac26ZMwG7aT | Scene-2 | Low-light-3 | 445 | 120 |
| 19 | 5LpN3gDmAk7 | Scene-1 | Overexposure-4 | 405 | 150 |
| 20 | mJXqzFtmKg4 | Scene-3 | Scattering-2 | 445 | 105 |
| 21 | real-world 1 | Scene-6 | Normal-1 | 400 | 100 |
| 22 | real-world 2 | Scene-7 | Low-light-3 | 400 | 100 |
| 23 | real-world 1 | Scene-6 | Low-light-3 | 400 | 100 |
| 24 | real-world 2 | Scene-7 | Normal-1 | 400 | 100 |

## G DOES A LARGER DIMENSION YIELD BETTER RESULTS (MODEL SCALING)?

To study the scalability of Tucker Adaptation (TuKA) and the capacity trade-offs between shared and task-specific factors, we performed a rank-scaling ablation. The default TuKA setting used in the main experiments is:

$$r_1 = r_2 = 8, \quad r_3 = r_4 = 64,$$

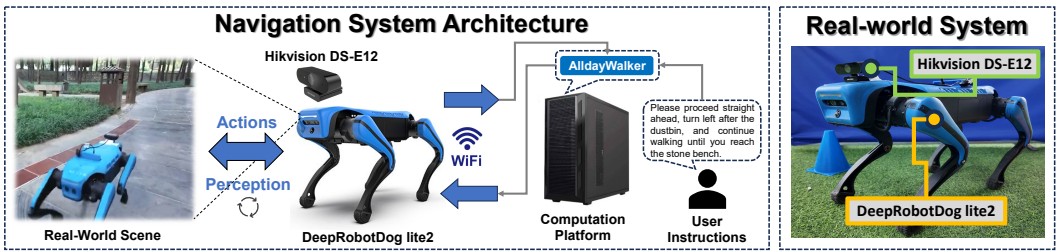

Figure 9: Illustration of the proposed robotic navigation system.

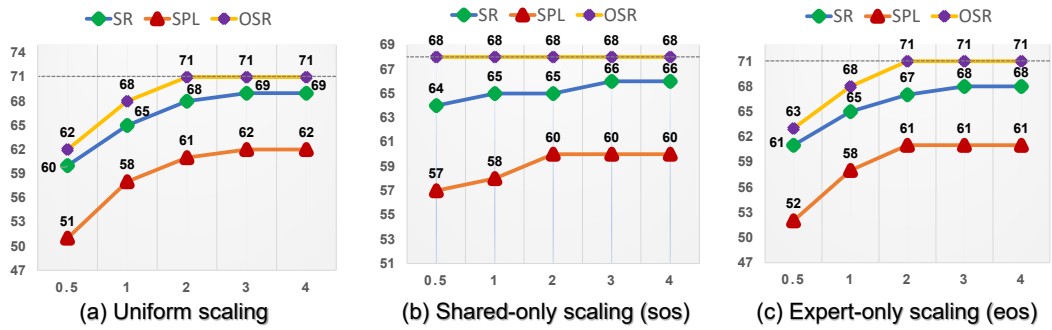

Figure 10: The analysis results for TuKA parameter growth for scaling of ranks.

with the number of scene experts $M = 5$ and environment experts $N = 4$. These default settings follow the implementation details in Appendix § C.

We consider multiplicative scale factors $s \in \{0.5, 2, 3, 4\}$ and evaluate the following scaling schemes:

- Uniform scaling (all): multiply $(r_1, r_2, r_3, r_4)$ by $s$, i.e. $(s \cdot 8, s \cdot 8, s \cdot 64, s \cdot 64)$.

- Shared-only scaling (sos): multiply only $(s \cdot r_1, s \cdot r_2)$ by $s$, keep $(r_3, r_4)$ at baseline.

- Expert-only scaling (eos): multiply only $(s \cdot r_3, s \cdot r_4)$ by $s$, keep $(r_1, r_2)$ at baseline.

For each configuration we train AlldayWalker under the same DKIL pipeline and evaluation protocol as in the main paper (same optimizer, learning rate, task ordering and evaluation metrics). We report averaged metrics across the 20 AML-VLN tasks: Success Rate (SR), SR forgetting (F-SR), SPL, F-SPL, OSR and F-OSR. As shown in Figure 10, our proposed TuKA is similar to vanilla LoRA in that increasing the rank in low-rank scaling experiments typically yields better results, yet this improvement exhibits an upper bound. Detailed experimental results are presented in Table 10, Table 11, Table 12, Table 13. Based on experimental results, we observe that specific components exhibit higher saturation values at upper limits compared to shared components. Consequently, we recommend prioritising expert dimensions over shared dimensions. Furthermore, owing to the higher-order properties of tensors, even linear scaling can induce substantial changes in adapter parameters. We therefore advise favouring adaptation with lower ranks.

Table 10: The Analysis Results for TuKA Parameter Growth for Scaling of Ranks
(**scale factors** $s = 0.5$).

| Rank $r_1, r_2$ | Rank $r_3, r_4$ | Scale-Type | SR↑ | F-SR↓ | SPL↑ | F-SPL↓ | OSR↑ | F-OSR↓ |
|---|---|---|---|---|---|---|---|---|
| 4,4 | 32,32 | all | 60 | 11 | 51 | 18 | 62 | 9 |
| 4,4 | 64,64 | sos | 64 | 11 | 57 | 18 | 68 | 9 |
| 8,8 | 32,32 | eos | 61 | 11 | 52 | 18 | 63 | 9 |
| 8,8 | 64,64 | equal | **65** | **11** | **58** | **18** | **68** | **9** |

Table 11: The Analysis Results for TuKA Parameter Growth for Scaling of Ranks (**scale factors** $s = 2$).

| Rank $r_1, r_2$ | Rank $r_3, r_4$ | Scale-Type | SR↑ | F-SR↓ | SPL↑ | F-SPL↓ | OSR↑ | F-OSR↓ |
|---|---|---|---|---|---|---|---|---|
| 16,16 | 128,128 | all | 68 | 11 | 61 | 18 | 71 | 9 |
| 16,16 | 64,64 | sos | 65 | 11 | 60 | 18 | 68 | 9 |
| 8,8 | 128,128 | eos | 67 | 11 | 61 | 18 | 71 | 9 |
| 8,8 | 64,64 | equal | 65 | 11 | 58 | 18 | 68 | 9 |

Table 12: The Analysis Results for TuKA Parameter Growth for Scaling of Ranks (**scale factors** $s = 3$).

| Rank $r_1, r_2$ | Rank $r_3, r_4$ | Scale-Type | SR↑ | F-SR↓ | SPL↑ | F-SPL↓ | OSR↑ | F-OSR↓ |
|---|---|---|---|---|---|---|---|---|
| 24,24 | 192,192 | all | 69 | 11 | 62 | 18 | 71 | 9 |
| 24,24 | 64,64 | sos | 66 | 11 | 60 | 18 | 68 | 9 |
| 8,8 | 192,192 | eos | 68 | 11 | 61 | 18 | 71 | 9 |
| 8,8 | 64,64 | equal | 65 | 11 | 58 | 18 | 68 | 9 |

Table 13: The Analysis Results for TuKA Parameter Growth for Scaling of Ranks (**scale factors** $s = 4$).

| Rank $r_1, r_2$ | Rank $r_3, r_4$ | Scale-Type | SR↑ | F-SR↓ | SPL↑ | F-SPL↓ | OSR↑ | F-OSR↓ |
|---|---|---|---|---|---|---|---|---|
| 32,32 | 256,256 | all | 69 | 11 | 62 | 18 | 71 | 9 |
| 32,32 | 64,64 | sos | 66 | 11 | 60 | 18 | 68 | 9 |
| 8,8 | 256,256 | eos | 68 | 11 | 61 | 18 | 71 | 9 |
| 8,8 | 64,64 | equal | 65 | 11 | 58 | 18 | 68 | 9 |

## H DOES THE THIRD-ORDER TENSOR CAN WELL REPRESENT THE MULTI-HIERARCHICAL KNOWLEDGE?

Our TuKA uses a fourth-order tensor to represent multi-hierarchical navigation knowledge (i.e., scenes and environments). In this section, we explore whether third-order tensors are sufficient to represent multi-hierarchical knowledge. Specifically, a three-order tensor $\boldsymbol{\mathcal{X}}^l \in \mathbb{R}^{a_l \times b_l \times (M \times N)}$ can be decomposed:

$$\boldsymbol{\mathcal{X}}^l = \boldsymbol{\mathcal{G}} \times_1 \boldsymbol{U}^1 \times_2 \boldsymbol{U}^2 \times_3 \boldsymbol{U}^3, \tag{22}$$

where $\times_n, n = 1, 2, 3, 4$ denotes the $n$-th modal product of the tensor and matrix. $\boldsymbol{\mathcal{G}} \in \mathbb{R}^{r_1 \times r_2 \times r_3}$ is a core tensor, which contains interaction information between all patterns, and is used to learn the navigation-shared knowledge. The factor matrix $\boldsymbol{U}^1 \in \mathbb{R}^{a_l \times r_1}$ represents the transformation pattern of the feature from $r_1$ dimension to $a$, which can be regarded as a decoder; $\boldsymbol{U}^2 \in \mathbb{R}^{b_l \times r_2}$ represents the transformation pattern of the feature from $b$ dimension to $r_2$, which can be regarded as a encoder. The factor matrix $\boldsymbol{U}^3 \in \mathbb{R}^{(M \times N) \times r_3}$ represents the $(M \times N)$ group of experts, with each scene expert $\boldsymbol{U}^3[t, :]$ is used to learn the $t$-th specific navigation task knowledge. Thus, for the $t$-th scenario with $s$-th scene and $e$-th environment adaptation, we extract the task-specific $\boldsymbol{U}^3[t, :]$ from the tensor $\boldsymbol{\mathcal{X}}$ to constitute adaptation weight $\Delta \boldsymbol{W}_t$:

$$\Delta \boldsymbol{W}_t = \boldsymbol{U}^1 \cdot (\boldsymbol{\mathcal{G}} \times_3 \boldsymbol{U}^3[t, :]) \cdot (\boldsymbol{U}^2)^T. \tag{23}$$

The results of learning multi-hierarchical knowledge using third-order tensors are presented in Table 14. Based on the experiment results, compared to the multi-hierarchical knowledge decoupled representation of fourth-order tensors, third-order tensors do not decouple the multi-hierarchical knowledge. Instead, they employ a single expert matrix to learn all knowledge sequentially, consequently yielding suboptimal performance. In summary, the proposed multi-hierarchical representation learning using higher-order tensors is significant.

Table 14: The Analysis Results (SR/F-SR/SPL/F-SPL/OSR/F-OSR in %) of Third-order Tensors for Representation multi-hierarchical Knowledge.

| Comparisons | T1 | T2 | T3 | T4 | T5 | T6 | T7 | T8 | T9 | T10 | T11 | T12 | T13 | T14 | T15 | T16 | T17 | T18 | T19 | T20 | Avg. |
|---|---|---|---|---|---|---|---|---|---|---|---|---|---|---|---|---|---|---|---|---|---|
| **3-order SR ↑** | 65 | 21 | 49 | 61 | 30 | 46 | 70 | 30 | 70 | 71 | 60 | 61 | 41 | 73 | 54 | 71 | 48 | 49 | 35 | 68 | 54 |
| **4-order SR ↑** | 79 | 23 | 71 | 81 | 33 | 50 | 87 | 38 | 79 | 75 | 79 | 67 | 50 | 86 | 71 | 76 | 67 | 63 | 43 | 81 | **65** |
| **3-order F-SR ↓** | 13 | 36 | 33 | 12 | 35 | 15 | 6 | 35 | 11 | 2 | 5 | 10 | 36 | 2 | 33 | 16 | 28 | 16 | 36 | 0 | 19 |
| **4-order F-SR ↓** | 2 | 27 | 23 | 8 | 21 | 1 | 0 | 28 | 9 | 0 | 2 | 6 | 30 | 3 | 24 | 10 | 1 | 4 | 21 | 0 | **11** |
| **3-order SPL ↑** | 60 | 16 | 51 | 60 | 28 | 43 | 70 | 26 | 60 | 65 | 50 | 51 | 38 | 61 | 48 | 62 | 48 | 42 | 35 | 51 | 48 |
| **4-order SPL ↑** | 70 | 20 | 67 | 73 | 25 | 32 | 80 | 34 | 73 | 71 | 68 | 56 | 48 | 75 | 65 | 70 | 62 | 60 | 41 | 67 | **58** |
| **3-order F-SPL ↓** | 18 | 36 | 38 | 21 | 36 | 18 | 18 | 45 | 26 | 15 | 18 | 22 | 43 | 11 | 39 | 24 | 35 | 16 | 39 | 0 | 26 |
| **4-order F-SPL ↓** | 13 | 31 | 28 | 12 | 25 | 8 | 12 | 34 | 15 | 6 | 12 | 14 | 38 | 5 | 29 | 16 | 14 | 8 | 27 | 0 | **17** |
| **3-order OSR ↑** | 71 | 22 | 55 | 66 | 36 | 51 | 72 | 26 | 74 | 71 | 66 | 61 | 42 | 80 | 51 | 70 | 51 | 53 | 41 | 74 | 57 |
| **4-order OSR ↑** | 81 | 26 | 73 | 81 | 41 | 57 | 88 | 38 | 83 | 79 | 83 | 71 | 52 | 87 | 74 | 78 | 71 | 63 | 48 | 84 | **68** |
| **3-order F-OSR ↓** | 12 | 38 | 35 | 14 | 35 | 12 | 5 | 39 | 15 | 2 | 5 | 10 | 35 | 0 | 38 | 14 | 26 | 15 | 36 | 0 | 19 |
| **4-order F-OSR ↓** | 1 | 24 | 19 | 8 | 18 | 1 | 1 | 28 | 8 | 2 | 1 | 4 | 27 | 5 | 19 | 8 | 1 | 3 | 20 | 0 | **9** |

# I  DOES THE HIERARCHICAL ARCHITECTURE PLAY A ROLE, OR IS IT THE HIGHER-ORDER TENSORS?

Our proposed TuKA represents task adaptation in a higher-order tensor space, which naturally decouples scene and environment knowledge. A natural question is whether such gains stem merely from having a hierarchical architecture (scene branch + environment branch), or from the expressive power of high-order tensor representations. To disentangle these factors, we design a hierarchical baseline using three-layer LoRA modules. Specifically, we construct a three-level LoRA adaptation $\Delta \boldsymbol{W} = C_e \cdot B_s \cdot A$.

- A shared matrix $\mathbf{A}$ that captures common knowledge across all tasks.
- Multiple scene-specific matrices $\mathbf{B}_s$, each dedicated to a scene $s$.
- Multiple environment-specific matrices $\mathbf{C}_e$, each dedicated to an environment $e$.

During training, the model activates one $\mathbf{B}_s$ and one $\mathbf{C}_e$ depending on the current task, while $\mathbf{A}$ remains shared. The $\mathbf{A}$ is constrained to employ a loss function similar to elastic weight consolidation (Eq.4-Eq.6) and $\mathbf{B}_s$, $\mathbf{C}_e$ is constrained to employ a loss function similar to expert consistency constraints (Eq.7) and task-specific expert subspace orthogonal constraint (Eq.8). It employs selection inference based on the proposed expert selection methodology (§ 3.4). This design mimics the scene–environment hierarchy but still operates entirely within second-order (matrix) structures, without using higher-order tensor decomposition. We align the parameter budget of this hierarchical LoRA with TuKA's default configuration for fair comparison (keeping the rank of $\mathbf{A} \in \mathbb{R}^{a \times 48}$, $\mathbf{B} \in \mathbb{R}^{48 \times 48}$, and $\mathbf{C} \in \mathbb{R}^{48 \times b}$ so that the total trainable parameters are comparable). Specifically, we use one $\mathbf{A}$, five $\mathbf{B}_s$, and four $\mathbf{C}_e$, thus

$$Param_{H-LoRA} = 1024 \times 48 + 5 \times 48 \times 48 + 4 \times 48 \times 1024 = 257,280 \approx 0.3M.$$

The training follows the AML-VLN setting (20 sequential tasks, same task order, optimizer, and hyperparameters). We report averaged SR, F-SR, SPL, F-SPL, OSR, and F-OSR across the 20 tasks. The results are summarized in Figure 11 and Table 15. Based on results, the hierarchical ABC-LoRA improves over flat single-LoRA baselines by better capturing scene–environment structure, showing modest gains in SR and reduced forgetting compared to O-LoRA. However, it consistently underperforms TuKA. While hierarchical LoRA introduces structural disentanglement, it cannot fully capture the *multi-hierarchical interactions* (scene–environment couplings) that higher-order tensors naturally model. Specifically, TuKA's Tucker core enables parameter sharing not only across scene or environment dimensions individually, but also across their joint combinations, which hierarchical LoRA lacks.

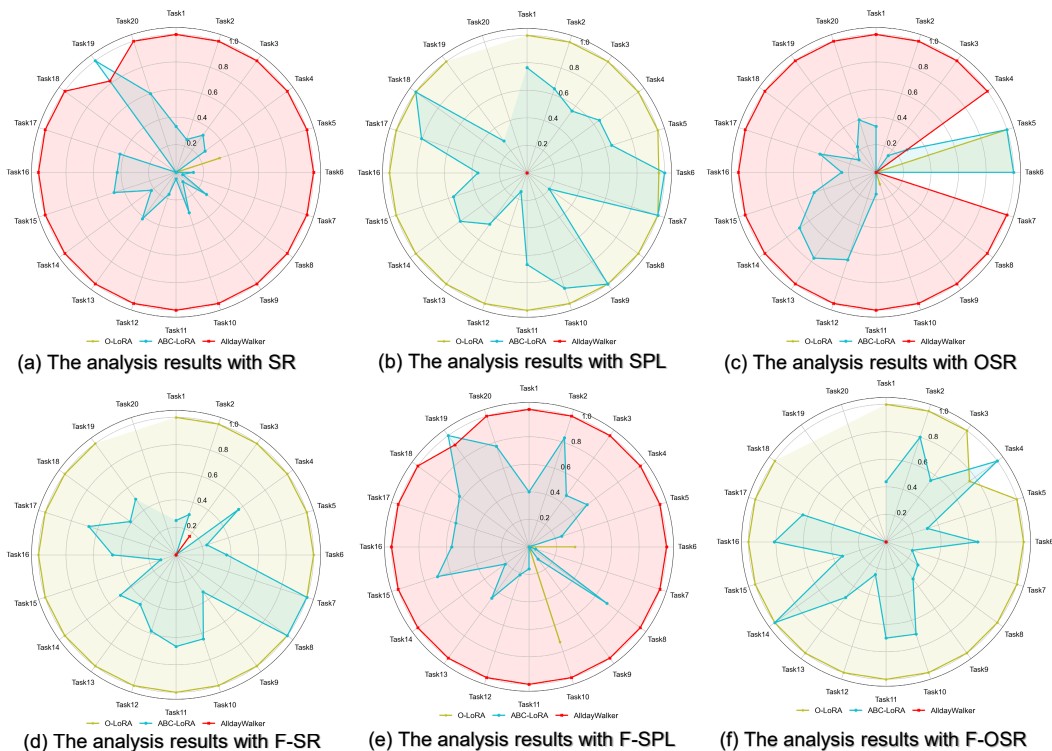

Figure 11: The comparison results for ABC-LoRA with hierarchical architecture.

Table 15: The Comparison Results (SR/F-SR/SPL/F-SPL/OSR/F-OSR in %) of ABC-LoRA for Representation Multi-hierarchical Knowledge.

| Comparisons | T1 | T2 | T3 | T4 | T5 | T6 | T7 | T8 | T9 | T10 | T11 | T12 | T13 | T14 | T15 | T16 | T17 | T18 | T19 | T20 | Avg. |
|---|---|---|---|---|---|---|---|---|---|---|---|---|---|---|---|---|---|---|---|---|---|
| **O-LoRA SR** ↑ | 67 | 19 | 47 | 58 | 31 | 42 | 67 | 27 | 67 | 62 | 58 | 49 | 38 | 68 | 52 | 62 | 46 | 52 | 34 | 71 | 51 |
| **ABC-LoRA SR** ↑ | 71 | 20 | 55 | 64 | 30 | 43 | 68 | 30 | 68 | 66 | 59 | 52 | 43 | 72 | 61 | 68 | 55 | 52 | 35 | 77 | 55 |
| **AlldayWalker SR** ↑ | 79 | 23 | 71 | 81 | 33 | 50 | 87 | 38 | 79 | 75 | 79 | 67 | 50 | 86 | 71 | 76 | 67 | 63 | 43 | 81 | 65 |
| **O-LoRA F-SR** ↓ | 19 | 41 | 41 | 28 | 38 | 24 | 4 | 38 | 15 | 17 | 11 | 13 | 43 | 2 | 40 | 24 | 32 | 5 | 35 | 0 | 24 |
| **ABC-LoRA F-SR** ↓ | 15 | 36 | 33 | 21 | 32 | 25 | 4 | 30 | 15 | 15 | 8 | 7 | 36 | 0 | 33 | 15 | 26 | 5 | 25 | 0 | 19 |
| **AlldayWalker F-SR** ↓ | 2 | 27 | 23 | 8 | 21 | 1 | 0 | 28 | 9 | 0 | 2 | 6 | 30 | -3 | 24 | 10 | 1 | 4 | 21 | 0 | 11 |
| **O-LoRA SPL** ↑ | 64 | 17 | 47 | 55 | 28 | 41 | 66 | 27 | 58 | 61 | 49 | 38 | 35 | 59 | 46 | 58 | 41 | 47 | 28 | 57 | 46 |
| **ABC-LoRA SPL** ↑ | 66 | 17 | 50 | 60 | 28 | 41 | 66 | 27 | 58 | 60 | 52 | 50 | 45 | 70 | 55 | 61 | 50 | 49 | 31 | 61 | 50 |
| **AlldayWalker SPL** ↑ | 70 | 20 | 67 | 73 | 25 | 32 | 80 | 34 | 73 | 71 | 68 | 56 | 48 | 75 | 65 | 70 | 62 | 60 | 41 | 67 | 58 |
| **O-LoRA F-SPL** ↓ | 25 | 44 | 43 | 28 | 42 | 27 | 15 | 38 | 33 | 20 | 21 | 26 | 47 | 17 | 46 | 29 | 41 | 25 | 35 | 0 | 30 |
| **ABC-LoRA F-SPL** ↓ | 16 | 35 | 25 | 21 | 29 | 15 | 15 | 38 | 21 | 15 | 18 | 21 | 42 | 11 | 31 | 22 | 32 | 15 | 31 | 0 | 23 |
| **AlldayWalker F-SPL** ↓ | 13 | 31 | 28 | 12 | 25 | 8 | 12 | 34 | 15 | 6 | 12 | 14 | 38 | 5 | 29 | 16 | 14 | 8 | 27 | 0 | 17 |
| **O-LoRA OSR** ↑ | 71 | 20 | 47 | 58 | 33 | 49 | 68 | 28 | 74 | 76 | 58 | 57 | 39 | 73 | 54 | 62 | 46 | 55 | 37 | 71 | 54 |
| **ABC-LoRA OSR** ↑ | 75 | 25 | 59 | 70 | 35 | 45 | 69 | 35 | 75 | 68 | 62 | 60 | 45 | 76 | 68 | 71 | 60 | 60 | 49 | 81 | 59 |
| **AlldayWalker OSR** ↑ | 81 | 26 | 73 | 81 | 41 | 57 | 88 | 38 | 83 | 79 | 83 | 71 | 52 | 87 | 74 | 78 | 71 | 63 | 48 | 84 | 68 |
| **O-LoRA F-OSR** ↓ | 17 | 39 | 48 | 17 | 37 | 22 | 4 | 35 | 14 | 15 | 11 | 12 | 41 | 0 | 37 | 24 | 32 | 4 | 20 | 0 | 21 |
| **ABC-LoRA F-OSR** ↓ | 8 | 36 | 35 | 20 | 24 | 15 | 0 | 30 | 10 | 10 | 8 | 6 | 34 | 0 | 25 | 21 | 20 | 3 | 20 | 0 | 16 |
| **AlldayWalker F-OSR** ↓ | 1 | 24 | 19 | 8 | 18 | 1 | -1 | 28 | 8 | -2 | 1 | 4 | 27 | -5 | 19 | 8 | -1 | 3 | 20 | 0 | 9 |

# J HOW DOES SCALABILITY FARE WHEN KNOWLEDGE EXHIBITS A MORE MULTI-HIERARCHICAL STRUCTURE?

Our proposed AlldayWalker employs a fourth-order tensor Tucker decomposition to adapt to new navigation scenarios. Under our settings, each navigation scenario comprises two hierarchies of

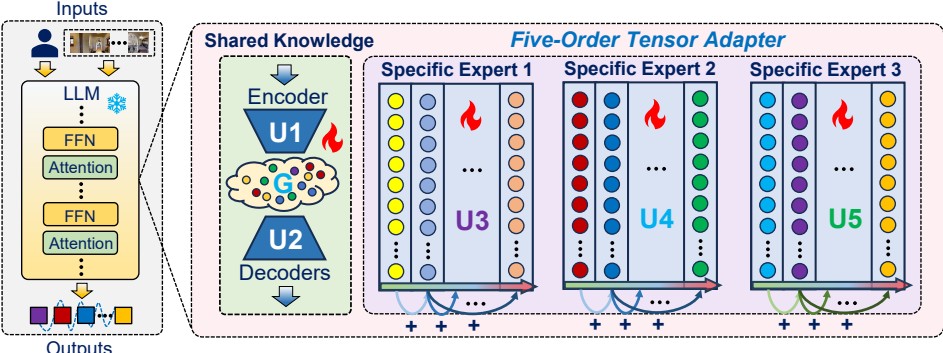

Figure 12: Illustration of the fifth-order tensor TuKA architecture.

knowledge: the scene and the environment. We explore how incorporating more hierarchical knowledge enables rapid adaptation using higher-order TuKA. Specifically, we add an additional hierarchies of knowledge, i.e., language instructions, following Dialogue Understanding Navigation (DUN) (Thomason et al. (2020)). We add one more hierarchies on top of the fourth-order tensor to form a fifth-order tensor $\boldsymbol{\mathcal{X}} \in \mathbb{R}^{a \times b \times M \times N \times P}$, and it can be decomposed into:

$$\boldsymbol{\mathcal{X}} = \boldsymbol{\mathcal{G}} \times_1 \boldsymbol{U}^1 \times_2 \boldsymbol{U}^2 \times_3 \boldsymbol{U}^3 \times_4 \boldsymbol{U}^4 \times_5 \boldsymbol{U}^5, \tag{24}$$

where $\times_n, n = 1, 2, 3, 4, 5$ denotes the $n$-th modal product of the tensor and matrix. $\boldsymbol{\mathcal{G}} \in \mathbb{R}^{r_1 \times r_2 \times r_3 \times r_4 \times r_5}$ is core tensor, which contains interaction information between all patterns, and is used to learn the navigation-shared knowledge. The factor matrix $\boldsymbol{U}^1 \in \mathbb{R}^{a \times r_1}$ represents the transformation pattern of the feature from $r_1$ dimension to $a$, which can be regarded as a dimension-up matrix; $\boldsymbol{U}^2 \in \mathbb{R}^{b \times r_2}$ represents the transformation pattern of the feature from $b$ dimension to $r_2$, which can be regarded as a dimension-reducing matrix. The factor matrix $\boldsymbol{U}^3 \in \mathbb{R}^{M \times r_3}$ represents the $M$ group of scene experts, with each scene expert $\boldsymbol{U}^3[i, :]$ is used to learn the $i$-th specific scene knowledge; $\boldsymbol{U}^4 \in \mathbb{R}^{N \times r_4}$ represents $N$ group of environment experts, with each expert $\boldsymbol{U}^4[j, :]$ is used to learn the $j$-th specific environment knowledge. $\boldsymbol{U}^5 \in \mathbb{R}^{N \times r_5}$ represents $P$ group of instruction experts, with each expert $\boldsymbol{U}^5[q, :]$ is used to learn the $q$-th specific instruction knowledge. The fifth-order tensor TuKA is shown in Figure 12. Thus, for the $t$-th scenario with $s$-th scene, $e$-th environment, and $q$-th instruction adaptation, we extract the task-specific matrices $\boldsymbol{U}^3[s, :]$, $\boldsymbol{U}^4[e, :]$, and $\boldsymbol{U}^5[q, :]$ from the high-order tensor $\boldsymbol{\mathcal{X}}$ to constitute weight $\Delta \boldsymbol{W}_t$:

$$\Delta \boldsymbol{W}_t = \boldsymbol{U}^1 \cdot (\boldsymbol{\mathcal{G}} \times_3 \boldsymbol{U}^3[s, :] \times_4 \boldsymbol{U}^4[e, :] \times_5 \boldsymbol{U}^5[p, :]) \cdot (\boldsymbol{U}^2)^T. \tag{25}$$

For verification of the proposed fifth-order TuKA, we construct a three hierarchies knowledge benchmark containing ten tasks, as shown in Table 16, and train our TuKA on it. The experimental results are summarized in Table 16. Based on experimental results, our TuKA remains effective when extended to a fifth-order tensor, demonstrating superior performance.

## K DETAILED COMPARISON RESULTS

For ease of presentation, the data in Figure 7 is presented after normalization. The original comparison results (SPL, F-SPL, OSR, F-OSR) for Figure 7 are summarized in Table 18, Table 19 Table 20, and Table 21. Due to the complexity of trajectory computation in real-world scenes and we focus more on task success rates on real-world, we only report results from simulated scenes ($T_1$-$T_{20}$). In addition, we summarize the performance and parameter numbers of all comparison methods, as shown in Table 17. Based on the results, our AlldayWalker achieves consistent superiority across various metrics.

## L DOES LEARNING FOR MORE TASKS LEAD TO FORGETTING?

Our 24 tasks already cover diverse scenes and four distinct imaging conditions across both simulation and real-world environments, forming a sufficiently challenging lifelong learning benchmark. To further validate the continual learning performance of AlldayWalker when dealing with more

Table 16: Statistics of the 10 Sequential Tasks for Verifying Fifth-order Tensor TuKA, and Some Comparison Results (SR/F-SR in %).

| Task Statistics | | | | HydraLoRA | | SD-LoRA | | AlldayWalker | |
|---|---|---|---|---|---|---|---|---|---|
| Task ID | Scene ID | Environment | Instruction | SR↑ | F-SR↓ | SR↑ | F-SR↓ | SR↑ | F-SR↓ |
| 1 | ac26ZMwG7aT | Normal | VLN | 46 | 44 | 69 | 13 | 79 | 2 |
| 2 | 5LpN3gDmAk7 | Low-light | DUN | 29 | 60 | 41 | 35 | 51 | 29 |
| 3 | ac26ZMwG7aT | Overexposure | VLN | 30 | 68 | 29 | 34 | 39 | 29 |
| 4 | S9hNv5qa7GM | Scattering | DUN | 51 | 32 | 60 | 21 | 69 | 15 |
| 5 | 5LpN3gDmAk7 | Normal | VLN | 33 | 28 | 69 | 8 | 87 | 0 |
| 6 | ac26ZMwG7aT | Low-light | DUN | 25 | 51 | 41 | 19 | 55 | 5 |
| 7 | S9hNv5qa7GM | Overexposure | VLN | 22 | 48 | 30 | 38 | 33 | 21 |
| 8 | ac26ZMwG7aT | Scattering | DUN | 31 | 52 | 65 | 10 | 70 | 5 |
| 9 | 5LpN3gDmAk7 | Overexposure | VLN | 15 | 79 | 31 | 40 | 40 | 26 |
| 10 | 5LpN3gDmAk7 | Scattering | DUN | 10 | 65 | 19 | 42 | 21 | 30 |
| Avg | – | – | – | 29 | 53 | 45 | 26 | 55 | 16 |

Table 17: Comparison of Baselines: Trainable Parameters and Averaged Metrics on the 24-task AML-VLN Benchmark Learning. Metrics (SR/F-SR in %) Are Averages across 24 Tasks, and Metrics (SPL/F-SPL/OSR/F-OSR in %) are Averages across First 20 Tasks Reported in the Paper.

| Method | Avg SR ↑ | Avg F-SR ↓ | Avg SPL ↑ | Avg F-SPL ↓ | Avg OSR ↑ | Avg F-OSR ↓ | Parameter |
|---|---|---|---|---|---|---|---|
| Seq-FT | 11 | 87 | 4 | 95 | 6 | 91 | 14.68 M |
| LwF-LoRA | 12 | 84 | 6 | 92 | 9 | 86 | 14.68 M |
| EWC-LoRA | 15 | 79 | 9 | 87 | 13 | 81 | 14.68 M |
| Dense MoLE | 20 | 72 | 14 | 80 | 18 | 74 | 16.52 M |
| Sparse MoLE | 28 | 62 | 21 | 70 | 26 | 64 | 16.52 M |
| MoLA | 33 | 55 | 26 | 63 | 31 | 57 | 16.52 M |
| HydraLoRA | 38 | 46 | 33 | 54 | 37 | 47 | 17.20 M |
| BranchLoRA | 44 | 36 | 39 | 43 | 45 | 36 | 17.20 M |
| O-LoRA | 52 | 23 | 46 | 31 | 54 | 22 | 16.52 M |
| SD-LoRA | 56 | 18 | 50 | 25 | 58 | 17 | 16.52 M |
| AlldayWalker | 65 | 11 | 58 | 18 | 68 | 9 | 15.64 M |

Table 18: Test Results (SPL ↑ in %) of Comparison Experiments under the AML-VLN Settings.

| Comparisons | T1 | T2 | T3 | T4 | T5 | T6 | T7 | T8 | T9 | T10 | T11 | T12 | T13 | T14 | T15 | T16 | T17 | T18 | T19 | T20 | Avg. |
|---|---|---|---|---|---|---|---|---|---|---|---|---|---|---|---|---|---|---|---|---|---|
| Seq-FT | 0 | 0 | 0 | 0 | 0 | 0 | 4 | 0 | 9 | 0 | 2 | 8 | 5 | 3 | 4 | 12 | 3 | 6 | 2 | 18 | 4 |
| LwF-LoRA | 5 | 0 | 5 | 0 | 0 | 0 | 4 | 3 | 9 | 5 | 2 | 12 | 8 | 2 | 6 | 14 | 7 | 8 | 3 | 21 | 6 |
| EWC-LoRA | 5 | 6 | 6 | 8 | 2 | 4 | 7 | 4 | 10 | 12 | 5 | 14 | 8 | 4 | 10 | 15 | 14 | 11 | 5 | 23 | 9 |
| Dense MoLE | 9 | 5 | 14 | 16 | 11 | 8 | 16 | 6 | 16 | 14 | 15 | 15 | 10 | 7 | 17 | 18 | 22 | 16 | 9 | 27 | 14 |
| Sparse MoLE | 24 | 7 | 24 | 15 | 17 | 14 | 27 | 9 | 31 | 22 | 22 | 21 | 17 | 21 | 21 | 29 | 28 | 10 | 12 | 33 | 21 |
| MoLA | 29 | 8 | 32 | 32 | 18 | 21 | 36 | 13 | 45 | 27 | 25 | 26 | 22 | 29 | 24 | 36 | 31 | 17 | 14 | 42 | 26 |
| HydraLoRA | 45 | 11 | 39 | 39 | 20 | 31 | 51 | 18 | 52 | 35 | 29 | 29 | 27 | 34 | 29 | 41 | 33 | 24 | 15 | 48 | 33 |
| BranchLoRA | 48 | 13 | 43 | 49 | 24 | 36 | 54 | 20 | 63 | 39 | 41 | 41 | 30 | 48 | 33 | 52 | 42 | 32 | 24 | 54 | 39 |
| O-LoRA | 64 | 17 | 47 | 55 | 28 | 41 | 66 | 27 | 58 | 61 | 49 | 38 | 35 | 59 | 46 | 58 | 41 | 47 | 28 | 57 | 46 |
| SD-LoRA | 62 | 19 | 52 | 61 | 30 | 45 | 71 | 28 | 62 | 68 | 52 | 52 | 39 | 64 | 50 | 65 | 49 | 45 | 36 | 53 | 50 |
| AlldayWalker | 70 | 20 | 67 | 73 | 25 | 32 | 80 | 34 | 73 | 71 | 68 | 56 | 48 | 75 | 65 | 70 | 62 | 60 | 41 | 67 | 58 |

tasks. We also conduct additional continual learning experiments by adding two new real-world tasks and four simulation tasks. For additional task scenes and environments, refer to Table 22. The results are summarized in Table 4. The results show that incorporating more tasks does not lead to noticeable performance degradation, demonstrating that our AlldayWalker remains stable under even larger-scale lifelong learning settings. The visualization of these navigation tasks is shown in Figures 13-14.

Table 19: Test Results (**F-SPL** ↓ in %) of Comparison Experiments under the AML-VLN Settings.

| Comparisons | T1 | T2 | T3 | T4 | T5 | T6 | T7 | T8 | T9 | T10 | T11 | T12 | T13 | T14 | T15 | T16 | T17 | T18 | T19 | T20 | Avg. |
|---|---|---|---|---|---|---|---|---|---|---|---|---|---|---|---|---|---|---|---|---|---|
| Seq-FT | 100 | 100 | 100 | 100 | 100 | 100 | 93 | 100 | 91 | 93 | 95 | 89 | 93 | 94 | 94 | 89 | 90 | 89 | 100 | 82 | 95 |
| LwF-LoRA | 94 | 100 | 94 | 100 | 100 | 100 | 92 | 93 | 90 | 94 | 95 | 86 | 88 | 85 | 92 | 87 | 88 | 85 | 100 | 79 | 92 |
| EWC-LoRA | 91 | 93 | 89 | 90 | 94 | 92 | 88 | 93 | 87 | 85 | 92 | 83 | 93 | 94 | 89 | 80 | 79 | 84 | 94 | 75 | 87 |
| Dense MoLE | 85 | 91 | 86 | 77 | 78 | 78 | 85 | 89 | 80 | 83 | 82 | 82 | 79 | 69 | 83 | 73 | 67 | 72 | 86 | 68 | 80 |
| Sparse MoLE | 74 | 90 | 74 | 68 | 76 | 64 | 63 | 85 | 58 | 71 | 67 | 57 | 75 | 53 | 75 | 67 | 58 | 84 | 85 | 56 | 70 |
| MoLA | 68 | 81 | 63 | 62 | 64 | 65 | 48 | 80 | 48 | 67 | 59 | 49 | 69 | 49 | 72 | 62 | 54 | 69 | 83 | 47 | 63 |
| HydraLoRA | 51 | 67 | 55 | 49 | 64 | 46 | 31 | 68 | 46 | 50 | 52 | 44 | 66 | 46 | 69 | 56 | 49 | 53 | 75 | 33 | 54 |
| BranchLoRA | 39 | 60 | 49 | 41 | 57 | 32 | 22 | 57 | 37 | 52 | 32 | 32 | 63 | 27 | 58 | 37 | 45 | 45 | 58 | 24 | 43 |
| O-LoRA | 25 | 44 | 43 | 28 | 42 | 27 | 15 | 38 | 33 | 20 | 21 | 26 | 47 | 17 | 46 | 29 | 41 | 28 | 44 | 14 | 31 |
| SD-LoRA | 17 | 36 | 35 | 19 | 37 | 16 | 16 | 43 | 25 | 11 | 16 | 21 | 42 | 10 | 38 | 23 | 36 | 15 | 39 | 10 | 25 |
| **AlldayWalker** | 13 | 31 | 28 | 12 | 25 | 8 | 12 | 34 | 15 | 6 | 12 | 14 | 38 | 5 | 29 | 16 | 15 | 9 | 28 | 6 | 18 |

Table 20: Test Results (**OSR** ↑ in %) of Comparison Experiments under the AML-VLN Settings.

| Comparisons | T1 | T2 | T3 | T4 | T5 | T6 | T7 | T8 | T9 | T10 | T11 | T12 | T13 | T14 | T15 | T16 | T17 | T18 | T19 | T20 | Avg. |
|---|---|---|---|---|---|---|---|---|---|---|---|---|---|---|---|---|---|---|---|---|---|
| Seq-FT | 0 | 5 | 0 | 0 | 0 | 3 | 5 | 0 | 10 | 5 | 4 | 10 | 8 | 12 | 7 | 14 | 7 | 8 | 6 | 24 | 6 |
| LwF-LoRA | 5 | 8 | 7 | 4 | 0 | 7 | 4 | 3 | 17 | 8 | 8 | 15 | 8 | 8 | 10 | 15 | 12 | 11 | 7 | 27 | 9 |
| EWC-LoRA | 14 | 10 | 6 | 13 | 3 | 12 | 12 | 4 | 23 | 14 | 13 | 19 | 11 | 13 | 14 | 19 | 16 | 17 | 9 | 24 | 13 |
| Dense MoLE | 19 | 11 | 14 | 21 | 15 | 16 | 18 | 8 | 34 | 17 | 17 | 19 | 13 | 14 | 18 | 20 | 25 | 17 | 10 | 33 | 18 |
| Sparse MoLE | 29 | 13 | 27 | 33 | 18 | 28 | 31 | 11 | 59 | 24 | 23 | 30 | 20 | 29 | 25 | 36 | 31 | 10 | 14 | 38 | 26 |
| MoLA | 34 | 12 | 33 | 38 | 22 | 24 | 37 | 13 | 52 | 32 | 29 | 34 | 26 | 38 | 31 | 39 | 39 | 22 | 14 | 43 | 31 |
| HydraLoRA | 48 | 16 | 39 | 46 | 17 | 37 | 57 | 18 | 58 | 39 | 33 | 38 | 29 | 43 | 33 | 42 | 38 | 28 | 20 | 52 | 37 |
| BranchLoRA | 52 | 16 | 43 | 51 | 26 | 44 | 59 | 25 | 66 | 54 | 50 | 48 | 33 | 59 | 42 | 58 | 44 | 37 | 24 | 64 | 45 |
| O-LoRA | 71 | 20 | 47 | 58 | 33 | 49 | 68 | 28 | 74 | 76 | 58 | 57 | 39 | 73 | 54 | 62 | 46 | 55 | 37 | 71 | 54 |
| SD-LoRA | 72 | 23 | 57 | 69 | 37 | 52 | 75 | 28 | 75 | 74 | 67 | 62 | 43 | 81 | 56 | 72 | 52 | 54 | 43 | 73 | 58 |
| **AlldayWalker** | 81 | 26 | 73 | 81 | 41 | 57 | 88 | 38 | 83 | 79 | 83 | 71 | 52 | 87 | 74 | 78 | 71 | 63 | 48 | 84 | 68 |

Table 21: Test Results (**F-OSR** ↓ in %) of Comparison Experiments under the AML-VLN Settings.

| Comparisons | T1 | T2 | T3 | T4 | T5 | T6 | T7 | T8 | T9 | T10 | T11 | T12 | T13 | T14 | T15 | T16 | T17 | T18 | T19 | T20 | Avg. |
|---|---|---|---|---|---|---|---|---|---|---|---|---|---|---|---|---|---|---|---|---|---|
| Seq-FT | 100 | 91 | 100 | 100 | 100 | 94 | 93 | 100 | 88 | 93 | 94 | 83 | 87 | 89 | 92 | 83 | 87 | 89 | 93 | 73 | 91 |
| LwF-LoRA | 92 | 92 | 88 | 89 | 100 | 90 | 92 | 89 | 77 | 91 | 91 | 77 | 88 | 78 | 87 | 79 | 82 | 80 | 94 | 69 | 86 |
| EWC-LoRA | 81 | 89 | 89 | 77 | 94 | 85 | 84 | 87 | 73 | 82 | 88 | 74 | 81 | 73 | 81 | 77 | 75 | 71 | 91 | 72 | 81 |
| Dense MoLE | 73 | 90 | 84 | 70 | 71 | 77 | 81 | 84 | 69 | 76 | 78 | 72 | 73 | 61 | 79 | 72 | 60 | 69 | 89 | 61 | 74 |
| Sparse MoLE | 68 | 84 | 67 | 50 | 72 | 49 | 53 | 82 | 55 | 68 | 64 | 56 | 67 | 46 | 70 | 57 | 51 | 82 | 82 | 54 | 64 |
| MoLA | 62 | 79 | 63 | 45 | 63 | 58 | 46 | 78 | 42 | 61 | 55 | 41 | 74 | 39 | 66 | 54 | 42 | 60 | 84 | 44 | 57 |
| HydraLoRA | 39 | 62 | 57 | 32 | 58 | 37 | 25 | 67 | 30 | 45 | 49 | 34 | 61 | 40 | 61 | 50 | 44 | 49 | 71 | 29 | 47 |
| BranchLoRA | 34 | 55 | 52 | 27 | 53 | 28 | 12 | 49 | 27 | 48 | 26 | 22 | 58 | 15 | 52 | 28 | 35 | 37 | 56 | 11 | 36 |
| O-LoRA | 17 | 39 | 48 | 17 | 37 | 22 | 4 | 35 | 14 | 15 | 11 | 12 | 41 | 0 | 37 | 24 | 32 | 4 | 36 | 2 | 22 |
| SD-LoRA | 10 | 36 | 33 | 12 | 33 | 10 | 3 | 36 | 11 | 1 | 2 | 8 | 35 | -4 | 36 | 12 | 23 | 11 | 33 | 1 | 17 |
| **AlldayWalker** | 1 | 24 | 19 | 8 | 18 | 1 | -1 | 28 | 8 | -2 | 1 | 4 | 27 | -5 | 19 | 8 | -1 | 4 | 21 | -7 | 9 |

Table 22: Descriptions and dataset statistics of added Tasks T25–T30.

| Task | Scene Description | Environment | Train Number | Test Number |
|---|---|---|---|---|
| T25 | 8WUmhLawc2A | Normal Environment | 460 | 120 |
| T26 | 8WUmhLawc2A | Low-light Environment | 460 | 120 |
| T27 | 8WUmhLawc2A | Scattering Environment | 460 | 120 |
| T28 | 8WUmhLawc2A | Overexposed Environment | 460 | 120 |
| T29 | Real-World Scene 3 | Normal Environment | 400 | 100 |
| T30 | Real-World Scene 3 | Low-light Environment | 400 | 100 |

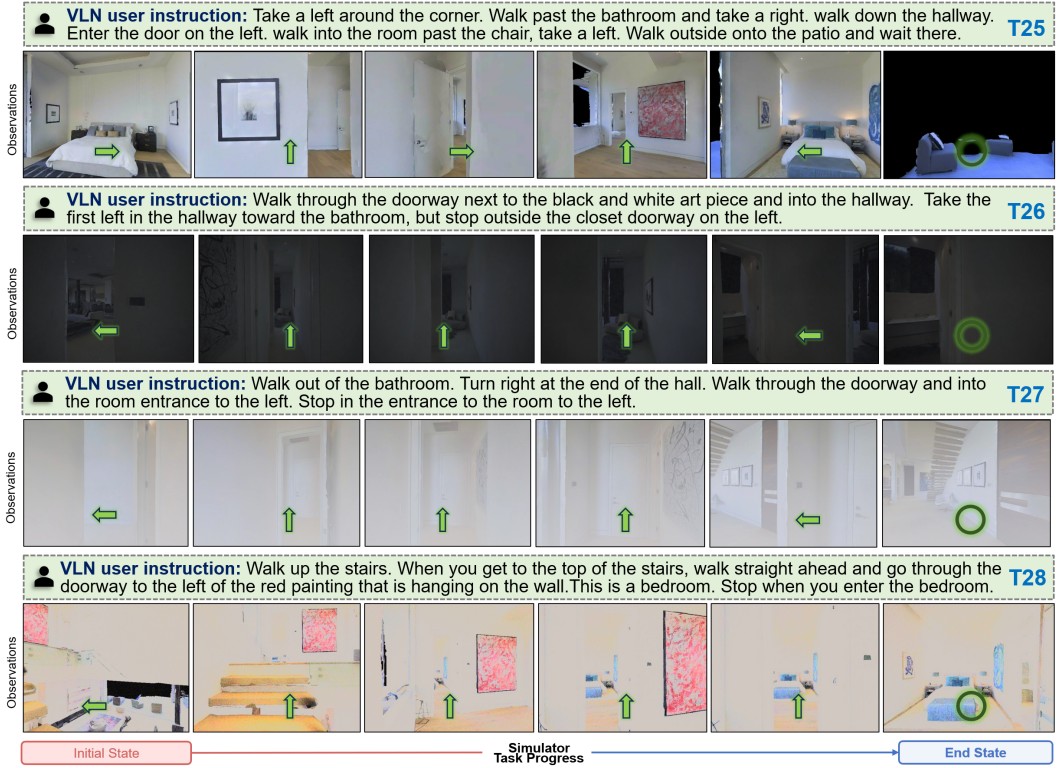

Figure 13: Illustration of the visualization examples for T25-T28.

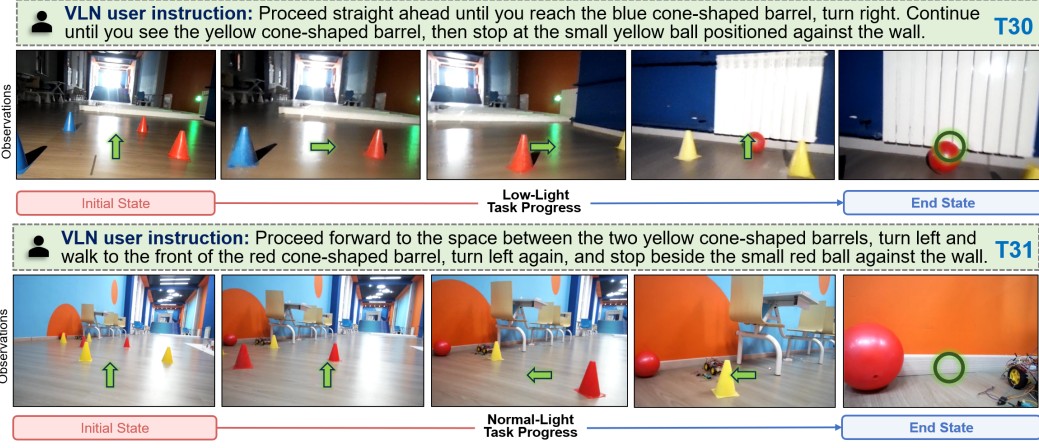

Figure 14: Illustration of the visualization examples for T29-T30.

# M    DISCUSSIONS AND FUTURE WORKS

**Future Works:** Our work introduces Tucker Adaptation and decoupled knowledge incremental learning for lifelong VLN. While the results are encouraging, several interesting directions remain. An interesting direction is to combine TuKA with memory-based or retrieval-augmented architectures, which may further enhance long-term retention and reduce catastrophic forgetting. Furthermore, beyond VLN, the idea of high-order adaptation could generalize to other embodied tasks such as manipulation, dialogue-based navigation, or multimodal planning.

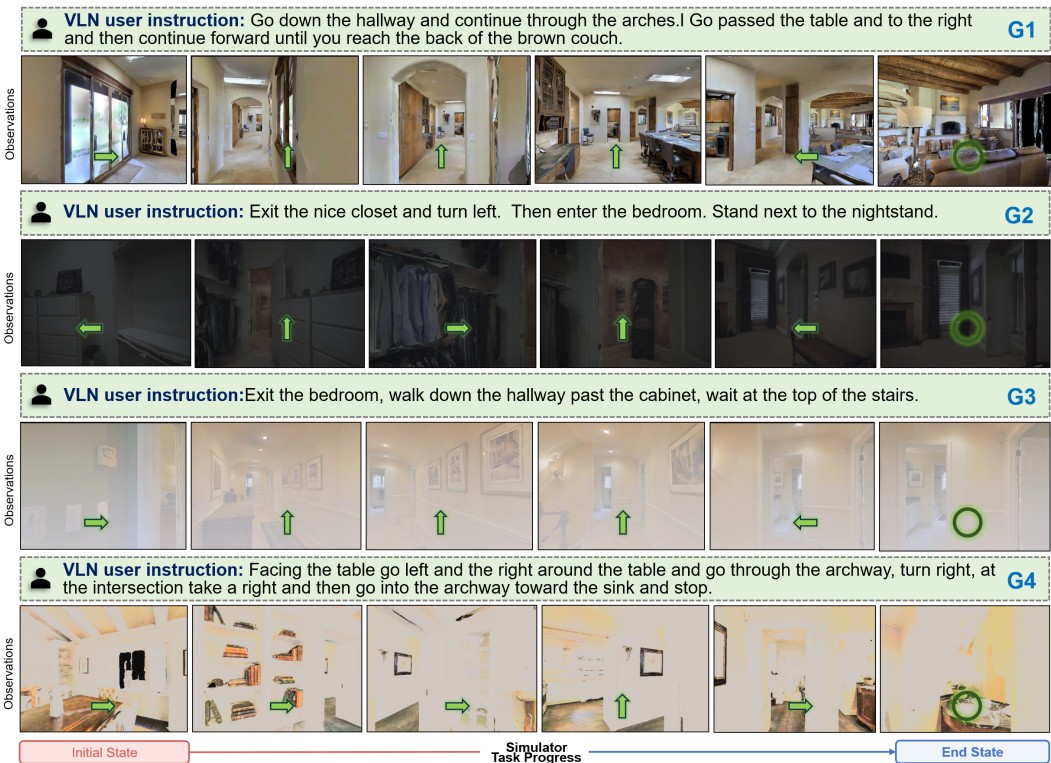

Figure 15: Illustration of the visualization examples for G1-G4.

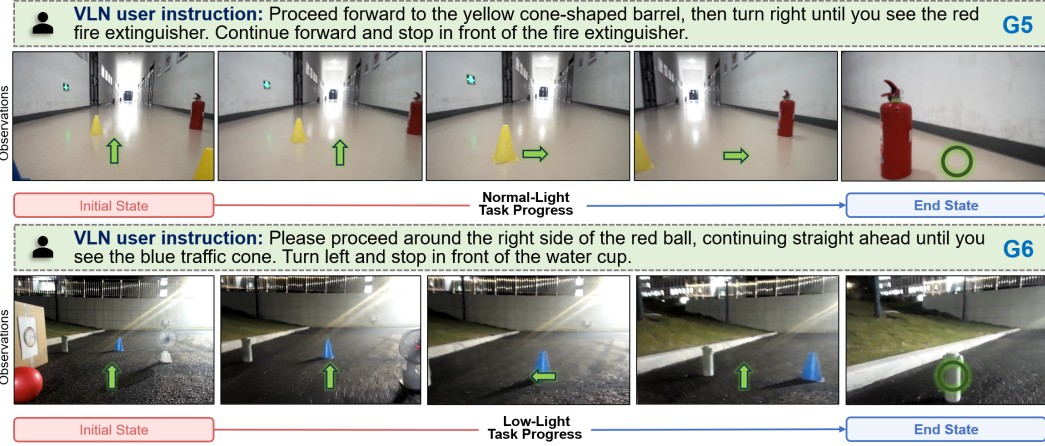

Figure 16: Illustration of the visualization examples for G5-G6.

**Societal Impacts:** This work aims to advance embodied agents that can operate in diverse conditions with less retraining, which may support real-world applications in service robotics, accessibility, and disaster response. However, navigation agents also raise concerns. Privacy and security issues may arise when deploying agents in homes or sensitive environments. Moreover, biases in simulated data or task design could lead to unfair performance across different cultural or environmental contexts. We encourage future work to evaluate not only technical performance but also ethical and societal implications before deployment.

