# OpenReview forum: "All-day Multi-scenes Lifelong Vision-and-Language Navigation with Tucker Adaptation"
_ICLR.cc/2026/Conference — ICLR 2026 Poster_

### Official Review · Reviewer_gySm · 2025-10-25

**Soundness:** 3
**Presentation:** 3
**Contribution:** 2
**Rating:** 4
**Confidence:** 4

**Summary:**

This paper formulates the All-Day Multi-Scenes Lifelong VLN (AML-VLN) problem, that is, agents learn sequential scenarios without forgetting. The authors propose: 1) TuKA (high-order tensor adapter) to model multi-hierarchical knowledge; 2) DKIL (incremental learning strategy) to mitigate forgetting. Experiments on the Allday-Habitat benchmark (24 tasks) show AlldayWalker outperforms LoRA variants.

**Strengths:**

1. Formalizing AML-VLN clarifies the "scene-environment-lifelong" gap in VLN research.
2. Allday-Habitat provides a unified testbed for lifelong VLN, saving future work from benchmark construction.
3. Experiments cover diverse baselines (10 LoRA variants) and key ablation (tensor order, DKIL losses), ensuring basic reliability.

**Weaknesses:**

1. novelty is limited: TuKA repurposes Tucker decomposition (a classic technique) for VLN adapters—there is no novel mathematical insight (e.g., improved decomposition efficiency, dynamic rank adjustment). DKIL’s "decoupled learning" is a standard combination of Elastic Weight Consolidation loss and orthogonality loss, with no new mechanism to handle cross-scene/environment knowledge transfer.

2. Experiment design is incomplete:
(1) No generalization test to unseen scenes/environments (critical for lifelong learning and VLN task)—the paper only tests sequential tasks from pre-defined sets.
(2) No ablation of "tensor vs. matrix MoE": A direct comparison between TuKA and a matrix-based MoE with the same hierarchy (scene+environment branches) would confirm if tensors add unique value.
(3) Real-world deployment is superficial: The DeepRobotDog experiment lacks details (e.g., navigation success rate in real low-light vs. simulation, failure cases).

3. Practicality is unproven: No inference latency/memory data: High-order tensor operations may be too slow for real-time robot navigation (e.g., edge devices), but the paper ignores this.

4. Some related and important works are missing citations:
[1] NavQ: Learning a Q-Model for Foresighted Vision-and-Language Navigation
[2] Test-time Adaptive Vision-and-Language Navigation
[3] Learning Vision-and-Language Navigation from YouTube Videos
[4] Magic: Meta-ability guided interactive chain-of-distillation for effective-and-efficient vision-and-language navigation

**Questions:**

See Weaknesses.

---

> ### Author Response · Authors · 2025-11-21
> **Response to Reviewer gySm (1 / 3)**
>
> We sincerely thank you for the evaluation and feedback. Your acknowledgment of the contribution of the new AML-VLN problem and the meaningful modifications of our Allday-Habitat simulators is greatly appreciated. Below, we address all the concerns.
>
> **1. TuKA is not a direct application of Tucker decomposition. (Weaknesses-1)**
>
> Thank you for the comment. We further clarify our contribution, TuKA is not a direct application of the traditional Tucker decomposition.
>
> We propose a new task requiring agents to perform continual learning across multiple scenes within diverse environments. Different from previous single-dimensional continual learning task settings, we consider the presence of different environments (i.e., low light, scattering, and overexposure) within the same scene. We expect to decouple the representation learning of scene knowledge and environment knowledge within a task, enabling flexible and effective navigation across any scene under all-weather conditions.
>
> The traditional LoRA-based fine-tuning methods either represent task-related knowledge within a single LoRA subspace or employ HydraLoRA architectures to represent shared knowledge in a shared encoder subspace while using different decoder spaces to represent task-specific knowledge. However, in our proposed problem, the task knowledge possesses multi-hierarchical dimensions (scene and environment). These methods struggle to explicitly decouple and represent multi-hierarchical knowledge (as shown in the middle diagram of the upper half of the paper Figure 1, where each B matrix learns task-specific knowledge, yet the scene and environment knowledge of the task are confounded together). We observe that the classical tensor Tucker Decomposition inherently has a natural structure for representing shared and specific knowledge (core tensor and multi-factor matrices). Therefore, we aim to decouple the hierarchical knowledge of navigation tasks within the multi-factor matrices, and represent the shared knowledge across all navigation tasks within the higher-order core tensors.
>
> However, this presents a critical challenge: **how to align the higher-order dimensions with two-dimensional weight matrices in LLM?** The traditional Tucker decomposition operates on full higher-order tensors, whereas LLM fine-tuning requires a matrix-shaped 2D adapter. This dimensional mismatch makes it challenging to represent knowledge and fine-tune LLM using traditional Tucker Decomposition. This drives us to explore a new knowledge representation selection method that decouples two-dimensional adaptation matrices from higher-order tensors to align LLM parameters, while preserving the multi-hierarchical knowledge decoupling representation.
>
> To achieve multi-hierarchical knowledge decoupling representation learning, we propose Tucker Adaptation (TuKA), a new fine-tuning method that employs Tucker decomposition to represent multi-hierarchical navigation knowledge. TuKA represents scene knowledge and environment knowledge in distinct expert factor matrices, and represents shared knowledge across multiple tasks using a shared core tensor and encoder-decoder. To align LLM parameters, TuKA selects each specific expert from a row within its entire expert factor matrices, reducing the expert matrix dimension to a vector. Thus, the higher-order knowledge tensor is reduced to a two-dimensional weight matrix for aligning LLM parameters (E.q 3). We further design a Decoupled Knowledge Incremental Learning (DKIL) strategy that consolidates shared subspaces while constraining the task-specific experts to mitigate catastrophic forgetting, during multi-hierarchical knowledge lifelong learning. In summary, we do not directly apply Tucker decomposition, but inspired by its form, we decouple the representation learning of coupled knowledge in different factor matrices and propose a tensor dimension reduction method for aligning LLM matrices.

---

> ### Author Response · Authors · 2025-11-21
> **Response to Reviewer gySm (2 / 3)**
>
> **2. More experiments and clarification  (Weaknesses-2)**
>
> Thank you for the comment. We add more experiments to verify the effectiveness of our proposed TuKA.
>
> I) Following your suggestion, we add six completely unseen tasks for generalization testing. Select the expert with the highest similarity during testing. These six tasks include four simulation scenarios and two real-world scenarios. The newly added tasks and generalization test results are summarized as:
>
> | Task | Scene  | environment  | Test Number | StreamVLN SR | BranchLoRA SR | SD-LoRA SR | TuKA SR |
> |--|--|--|--|--|--|--|-|
> | **G1** | JeFG25nYj2p | Normal Environment |105|45|52|53|65|
> | **G2** | ur6pFq6Qu1A | Low-light Environment |120|41|44|45|63|
> | **G3** | r47D5H71a5s | Scattering Environment |105|36|40|41|54|
> | **G4** | Vvot9Ly1tCj | Overexposed Environment |108|31|42|39|51|
> | **G5** | Real-World 4 | Normal Environment |100|36|38|37|55|
> | **G6** | Real-World 4 | Low-light Environment |100|18|21|21|43|
> | **Avg.** |  |  | |35|40|39|55|
>
> TuKA achieves superior generalization performance, achieving an average SR of 55%, surpassing SD-LoRA (39%) by 16% and BranchLoRA (40%) by 15%, demonstrating superior generalization. **Please refer to Appendix Figure 15-16 for these navigation visualizations**.
>
> II) In Appendix J, we extended the MoE-LoRA to a three-layer structure (A, B, C) to achieve direct hierarchical knowledge representation. Specifically, A learns shared knowledge, B learns scenes, and C learns environments. The experimental results show that, even with this extension, MoE-LoRA still falls short of TuKA. This indicates that the high-order tensor representation itself plays a more critical role than simply stacking an adapter structure. Therefore, the advantage of our method lies not only in architectural design but also in the intrinsic representational capability provided by high-order tensor factorization. In addition, our experiment results (Table 1–2, Fig. 7) consistently show that TuKA achieves better continual learning performance than MoE-LoRA variants such as HydraLoRA and BranchLoRA, supporting the representational advantage of high-order tensor adaptation. Following your suggestion, we add a new experiment. Specifically, we use the matrix-based MoE LoRA with the same hierarchy (scene+environment branches). The experiment results (SR%) are as follows:
>
> | Tasks | T1 | T2 | T3 | T4 | T5 | T6 | T7 | T8 | T9 | T10 | T11 | T12 | T13 | T14 | T15 | T16 | T17 | T18 | T19 | T20 | T21 | T22 | T23 | T24 | Avg. |
> |------|----|----|----|----|----|----|----|----|----|-----|-----|-----|-----|-----|-----|-----|-----|-----|-----|-----|-----|-----|-----|------|-----|
> | **TuKA** | 79 | 23 | 71 | 81 | 33 | 50 | 87 | 38 | 79 | 75 | 79 | 67 | 50 | 86 | 71 | 76 | 67 | 63 | 43 | 81 | 68 | 58 | 62 | 72 | 65 |
> | **MoE** | 70 | 19 | 61 | 54 | 26 | 34 | 57 | 21 | 58 | 56 | 56 | 41 | 36 | 61 | 52 | 58 | 45 | 46 | 28 | 65 | 49 | 37 | 45 | 51 | 47 |
>
> The results demonstrate that under the AML-VLN settings, our TuKA achieves better performance than MoE with the same hierarchy.
>
> III) We clarify that our real-world deployment is described in detail in the appendix, and we now provide additional quantitative comparisons. i) A detailed robot platform description is already included in Appendix G, and Figure 9 describes our quadruped robot setup (including sensing modules, signal transmission pipeline, offboard computation platform, etc.). Appendix Table 8 provides a detailed comparison between the collected real training data and test rounds versus the simulation environment. Real-world navigation videos are also available on our project website. ii) Following your suggestion, we now report the success rates in: low-light simulation tasks (T4, T11, T13, T15, T18), and real low-light scenes (T22, T23).
>
> | Setting                  | Avg. SR |
> |-------------------------|---------|
> | Low-light simulation    | **68.8%** |
> | Real low-light scenes   | **60.0%** |
>
> The real-world performance is lower, which is expected due to factors not present in the simulation (ambient light interference, sensor noise, uncontrolled reflections). This gap demonstrates that our real-world evaluation is realistic and more challenging. iii) Following your suggestion, we also include a representative failure case in Appendix Figure 17. The agent stops prematurely after interpreting the right-turn command due to an incorrect user instruction (no box exists in the scene), failing to reach the target position. This failure case provides a future research direction: how to enhance the robustness of agent navigation under erroneous user instructions.

---

> ### Author Response · Authors · 2025-11-21
> **Response to Reviewer gySm (3 / 3)**
>
> **3. Inference latency or memory comparison. (Weaknesses-3)**
>
> Thank you for the comment. We clarify that TuKA does not introduce significant inference overhead. Although TuKA is inspired by higher-order tensor factorization, its inference procedure is matrix-level and lightweight. i) Only a single row of each expert matrix is used at inference. Specifically, when a scene and environment are selected, TuKA extracts only one row from the scene expert (U3) and one row from the environment expert (U4). This directly reduces the high-order structure to a standard 2D adapter matrix, meaning no full tensor computation is performed at inference. ii) The core tensor is extremely low-dimensional. The low rank core tensor g has very small mode sizes (r ≪ weight matrix dimension). Therefore, the modal product operations involved in reconstructing the adapter are computationally negligible compared to LLM (7B) forward passes. Following your suggestion, we add a computational efficiency comparison. All methods use the hyperparameters in Appendix D and are measured on the same hardware.
>
> | Method          | Inference Latency (ms) | Extra Memory (MB) | Parameters (M) | Avg. SR (%) |
> | --------------- | ----------------- | ------------ | ---------- | ------- |
> | **MoLA**        | 127               | 68           | 16.52      | 33      |
> | **HydraLoRA**   | 128               | 72           | 17.20      | 38      |
> | **SD-LoRA**     | 142               | 65           | 16.52      | 56      |
> | **TuKA (ours)** | 135               | **62**       | **15.64**  | **65**  |
>
> TuKA shows similar latency to other multi-expert LoRA variants and uses comparable or lower memory, confirming that high-order modeling does not introduce practical inference burdens. Overall, the computational overhead introduced by TuKA (16M) is negligible compared to that of LLM (7B).
>
> **4. Some related and important works are missing citations.  (Weaknesses-4)**
>
> Thank you for pointing out the missing related works. We have added citations and expanded the discussion to include the four VLN works mentioned (NavQ [1], YouTube-VLN [3], MAGIC [4]). Below, we clarify their relationship to our AML-VLN setting.
>
> The methods (NavQ, MAGIC, YouTube-VLN) improve VLN policy learning through foresight modeling, meta-ability guidance, or video-driven pretraining. However, they operate in a single-task setting and do not address continual scene–environment learning, nor do they model hierarchical knowledge decomposition. In contrast, AML-VLN requires the agent to incrementally learn multiple scenes and multiple environments, preserve previously acquired knowledge, and avoid catastrophic forgetting capabilities not explored by these works. The FSTT method [2] adapts the VLN agent only during test-time with small, temporary parameter updates. They do not accumulate knowledge across tasks, do not store scene and environment-specific knowledge, and cannot prevent forgetting once the test episode ends. Our AML-VLN setting is fundamentally different: it is a lifelong learning problem, where the agent must build a reusable multi-hierarchical knowledge structure (scene + environment) over time. In summary, these related works improve VLN in complementary areas (policy learning, data efficiency, and adaptation). We now include these discussions in the revised Related Work section.

---

### Official Review · Reviewer_Xr3z · 2025-10-31

**Soundness:** 3
**Presentation:** 3
**Contribution:** 3
**Rating:** 6
**Confidence:** 4

**Summary:**

This paper addresses the continual learning problem in Vision-and-Language Navigation (VLN) by formalizing it as the All-Day Multi-Scenes Lifelong VLN (AML-VLN) problem. It proposes Tucker Adaptation (TuKA) — a parameter-efficient, multi-hierarchical adapter that represents navigation knowledge as a high-order tensor. By leveraging Tucker decomposition, TuKA disentangles shared, scene-specific, and environment-specific subspaces, enabling continual adaptation with reduced interference across sequential VLN tasks. The method is integrated into the AlldayWalker agent, which employs a decoupled knowledge incremental learning strategy to preserve prior knowledge while adapting to new conditions.

**Strengths:**

1.	The paper is well-written and clearly organized, with a solid motivation and clear presentation of the continual learning setting in VLN.
2.	Addresses the underexplored problem of continual learning in vision-and-language navigation — a task that naturally involves sequential adaptation yet has received limited attention in prior research.
3.	Proposes TuKA, a Tucker-based adapter that decouples scene- and environment-specific experts. This structure enables task-wise adaptation while mitigating interference, effectively reducing catastrophic forgetting.
4.	Provides comprehensive experiments, including ablations, scalability analyses and comparisons with multiple adapter-based baselines.
5.	Enables LLM-based embodied agents to perform continual learning, achieving higher SR/SPL and lower forgetting rates than other continual learning frameworks.

**Weaknesses:**

1.	The current experiments appear to evaluate tasks within the same building. Since a central goal of continual learning is to enable agents to transfer previously acquired knowledge to unseen environments, it would be valuable to include evaluations in completely new buildings or environments to more clearly demonstrate TuKA’s generalization ability beyond the trained domain.
2.	Continual learning results can be sensitive to the order in which tasks are learned. It would be helpful to clarify whether you have experimented with different training orders — for example, by shuffling or reversing the task sequence — to examine how such variations affect TuKA’s performance and forgetting behavior.
3.	TuKA structurally separates experts into scene- and environment-specific components, but it would be helpful to include experiments verifying whether each expert actually learns the characteristics of its corresponding task.
4.	In L806–807, it is mentioned that “they often suffer from catastrophic forgetting of previously learned scenarios, severely limiting their robustness in dynamic real-world settings.” Adding a discussion on Test-Time Adaptation (TTA) [1-2] as one possible way to address this issue would make the paper more complete.
5.	I will consider raising the score if these points are clarified.
[1] Gao, et al., "Fast-Slow Test-Time Adaptation for Online Vision-and-Language Navigation.", ICML 2024.
[2] Kim, et al., "Test-Time Adaptation for Online Vision-Language Navigation with Feedback-based Reinforcement Learning.", ICML 2025.

**Questions:**

1.	Could you include experiments in completely unseen environments?
2.	Could you discuss how much TuKA’s performance might depend on the training sequence?
3.	Could you report how the base StreamVLN policy performs under the same AML-VLN setup? Since TuKA is built upon StreamVLN, this comparison would clarify  how much improvement TuKA provides over the baseline model itself.
4.	As new environments or scenes are introduced, the number of factors would also increase — does this lead to improved generalization performance?

---

> ### Author Response · Authors · 2025-11-21
> **Response to Reviewer Xr3z (1 / 4)**
>
> Thank you for your careful reading of our manuscript and for the constructive comments. We sincerely appreciate your recognition of the proposed AML-VLN problem and the strengths of our AlldayWalker framework. Below, we address your concerns.
>
> **1. Could you include experiments in completely unseen environments? (Weaknesses-1 & Questions-1)**
>
> Thank you for your constructive comment. The current testing is conducted across multiple scenarios rather than a single building, encompassing all 24 tasks while setting the task ID to agnostic during testing. Following your suggestion, we add six completely unseen tasks for generalization testing. Select the expert with the highest similarity during testing. These six tasks include four simulation scenarios and two real-world scenarios. The newly added tasks and generalization test results are summarized as:
>
> | Task | Scene  | environment  | Test Number | StreamVLN SR | BranchLoRA SR | SD-LoRA SR | TuKA SR |
> |--|--|--|--|--|--|--|-|
> | **G1** | JeFG25nYj2p | Normal Environment |105|45|52|53|65|
> | **G2** | ur6pFq6Qu1A | Low-light Environment |120|41|44|45|63|
> | **G3** | r47D5H71a5s | Scattering Environment |105|36|40|41|54|
> | **G4** | Vvot9Ly1tCj | Overexposed Environment |108|31|42|39|51|
> | **G5** | Real-World 4 | Normal Environment |100|36|38|37|55|
> | **G6** | Real-World 4 | Low-light Environment |100|18|21|21|43|
> | **Avg.** |  | | |35|40|39|55|
>
> TuKA achieves superior generalization performance, achieving an average SR of 55%, surpassing SD-LoRA (39%) by 16% and BranchLoRA (40%) by 15%, demonstrating superior generalization. **Please refer to the Appendix Figure 15-16 for these unseen scenarios navigation visualizations**.
>
> **2. Could you discuss how much TuKA’s performance might depend on the training sequence? (Weaknesses-2 & Questions-2)**
>
> Thank you for the suggestion. Following your suggestion, we conduct two additional experiments to examine whether AlldayWalker is sensitive to the order in which tasks are learned: a reversed order experiment (T24 → T1), and a randomly shuffled order experiment. The experiment results are summarized below:
>
> | **Sequential** | T1   | T2   | T3   | T4   | T5   | T6   | T7   | T8   | T9   | T10  | T11  | T12  | T13  | T14  | T15  | T16  | T17  | T18  | T19  | T20  | T21  | T22  | T23  | T24  | Avg. |
> | -- | -- | -- | -- | -- | -- | -- | -- | -- | -- | -- | -- | -- | -- | -- | -- | -- | -- | -- | -- | -- | -- | -- | -- | -- | -- |
> | **SR(%)**                  | 79   | 23   | 71   | 81   | 33   | 50   | 87   | 38   | 79   | 75   | 79   | 67   | 50   | 86   | 71   | 76   | 67   | 63   | 43   | 81   | 68   | 58   | 62   | 72   | 65   |
>
> | **Reverse** | **T24** | **T23** | **T22** | **T21** | **T20** | **T19** | **T18** | **T17** | **T16** | **T15** | **T14** | **T13** | **T12** | **T11** | **T10** | **T9** | **T8** | **T7** | **T6** | **T5** | **T4** | **T3** | **T2** | **T1** | **Avg.** |
> | ------ | ------- | ------- | ------- | ------- | ----- | ------- | ------- | ------- | ------- | ------- | ---- | ------- | ------- | ------- | ------- | ------ | ------ | ------ | ------ | ------ | ------ | ------ | ------ | ------ | ----- |
> | **SR(%)**                | 71      | 61      | 57      | 67      | 80      | 42      | 62      | 66      | 75      | 70      | 85      | 49      | 66      | 78      | 74      | 78     | 37     | 86     | 49     | 34     | 82     | 72     | 24     | 87     | 65       |
>
> | **Shuffled** | **T7** | **T14** | **T3** | **T20** | **T9** | **T1** | **T22** | **T5** | **T17** | **T11** | **T8** | **T24** | **T2** | **T13** | **T6** | **T19** | **T4** | **T15** | **T23** | **T10** | **T12** | **T18** | **T21** | **T16** | **Avg.** |
> | ----- | ------ | ------- | ------ | ------- | ------ | ------ | ------- | --- | ------- | ------- | ------ | ------- | ----- | --- | ------ | ------- | ------ | ------- | ------- | ------- | ---- | ------- | ------- | ------- | ---- |
> | **SR(%)**        | 85     | 84      | 69     | 79      | 77     | 77     | 56      | 31     | 65      | 77      | 36     | 70      | 21     | 48      | 48     | 41      | 79     | 69      | 62      | 75      | 67      |  65     | 70      | 83      | 64       |
>
> Across all three settings, the overall performance remains stable, indicating that AlldayWalker is robust to task-order variations. This robustness comes from two design aspects of TuKA: i) Scene and environment knowledge are decoupled and stored in different rows of the factor matrices (U3 and U4). This reduces interference between consecutive tasks and makes the learning process less sensitive to the order of tasks. ii) As shown in our retrieval experiment (Reviewer W2Xo, Q-A7), CLIP-based expert matching achieves over 99.9% accuracy across 90 scenes. Because each task consistently retrieves the correct expert, DKIL updates the correct subspace regardless of training order. Together, these two factors allow AlldayWalker to remain stable under reversed and shuffled task sequences.

---

> ### Author Response · Authors · 2025-11-21
> **Response to Reviewer Xr3z (2 / 4)**
>
> **3. TuKA structurally separates experts into scene- and environment-specific components, but it would be helpful to include experiments verifying whether each expert actually learns the characteristics of its corresponding task. (Weaknesses-3)**
>
> Thank you for the suggestion. The proposed DKIL ensures that during training, only the corresponding scene expert is updated for a scene-specific task, and only the corresponding environment expert is updated for an environment-specific task (Appendix Algorithm 1). This mechanism guarantees that each expert matrix is responsible for learning its own designated knowledge.
>
> I) Following your suggestion, we add an experiment to verify whether corresponding experts learn effectively in continual learning. Specifically, for continual learning tasks T1 (Normal), T8 (Overexposure), T10 (Scattering), and T18 (Low Light) in the same scene (ID: ac26ZMwG7aT), we statistically compare the similarity between the each row of the environment expert matrix after its changes and its previous state, and the similarity between the each row of the scene expert matrix after its changes and its previous state. A lower similarity value for a row indicates that the expert has effectively learned knowledge for the task; a higher similarity value indicates a lower degree of learning. The experiment results of the environment expert are as follows:
>
> | Task | $\triangle Sim(U4[1,:])$ | $\triangle Sim(U4[2,:])$ | $\triangle Sim(U4[3,:])$  | $\triangle Sim(U4[4,:])$ |
> | :------------: | :------------: | :------------: | :------------: | :------------: |
> | T1 | 0.11 | 1.00 | 1.00 | 1.00 |
> | T8 | 1.00 | 1.00 | 1.00 | 0.81  |
> | T10  | 1.00  | 0.86 | 1.00 | 1.00  |
> |  T18 | 1.00  | 1.00  | 0.92 | 1.00  |
>
> Corresponding to Figure 6 in the paper, U4[1,:] is the expert for the normal environment, U4[2,:] is the expert for the scattering environment, U4[3,:] is the expert for the low-light environment, and U4[4,:] is the expert for the overexposed environment. Based on the experiment results, during continual learning, task-specific environment experts learn accordingly, while other experts remain unchanged. And with the increase in the number of learning tasks, learning gradually stabilizes in the later stages (with minimal changes). On the other hand, the experiment results of the scene expert are as follows:
>
> | Task  |  $\triangle Sim(U3[1,:])$ | $\triangle Sim(U3[2,:])$  | $\triangle Sim(U3[3,:])$  | $\triangle Sim(U3[4,:])$  | $\triangle Sim(U3[5,:])$  | $\triangle Sim(U3[6,:])$  | $\triangle Sim(U3[7,:])$  |
> | :------------: | :------------: | :------------: | :------------: | :------------: | :------------: | :------------: | :------------: |
> | T1  | 0.12  | 1.00  | 1.00  | 1.00  |  1.00 | 1.00  | 1.00  |
> | T8 | 0.83  | 1.00  | 1.00  | 1.00  |  1.00 | 1.00  | 1.00  |
> | T10 | 0.88  | 1.00  | 1.00  | 1.00  |  1.00 | 1.00  | 1.00  |
> | T18 | 0.93  | 1.00  | 1.00  | 1.00  |  1.00 | 1.00  | 1.00  |
>
> Corresponding to Figure 6 in the paper, U3[1,:] is the expert for the scene (ID: ac26ZMwG7aT). Based on the experiment results, during continual learning, task-specific scene experts learn accordingly, while other experts remain unchanged. And with the increase in the number of learning tasks, learning gradually stabilizes in the later stages (with minimal changes).
>
> II) Following your suggestion, we also add an experiment to verify that the experts indeed learn task-specific characteristics. Specifically, during testing, instead of selecting the expert with the highest matching score, we intentionally choose the expert with the lowest matching score (i.e., a mismatched expert). The results are summarized below:
>
> | Method | T1   | T2   | T3   | T4   | T5   | T6   | T7   | T8   | T9   | T10  | T11  | T12  | T13  | T14  | T15  | T16  | T17  | T18  | T19  | T20  | T21  | T22  | T23  | T24  | T25  | T26  | T27  | T28  | T29  | T30  | Avg. |
> | ------------------------------------------------------------ | ---- | ---- | ---- | ---- | ---- | ---- | ---- | ---- | ---- | ---- | ---- | ---- | ---- | ---- | ---- | ---- | ---- | ---- | ---- | ---- | ---- | ---- | ---- | ---- | ---- | ---- | ---- | ---- | ---- | ---- | ---- |
> | **AlldayWalker** | 77   | 23   | 71   | 70   | 33   | 50   | 86   | 38   | 79   | 75   | 80   | 66   | 50   | 85   | 70   | 77   | 68   | 63   | 44   | 81   | 68   | 58   | 62   | 66   | 72   | 72   | 76   | 35   | 67   | 61   | 64   |
> | **False Match** | 38   | 15   | 34   | 33   | 11   | 21   | 43   | 14   | 39   | 37   | 40   | 31   | 21   | 42   | 33   | 38   | 32   | 29   | 17   | 40   | 32   | 26   | 28   | 21   | 34   | 35   | 37   | 12   | 31   | 28   | 30   |
>
> The mismatched expert selection leads to a clear performance drop. This confirms that each expert encodes the knowledge it is supposed to learn. Therefore, the experts learn the task-specific knowledge under the DKIL training process.

---

> ### Author Response · Authors · 2025-11-21
> **Response to Reviewer Xr3z (3 / 4)**
>
> **4. Comparison and discussion with TTA VLN. (Weaknesses-4)**
>
> Thank you for pointing this out. We have added the citations and expanded our discussion of recent test-time adaptation (TTA) methods such as FSTTA (ICML 2024) and FeedTTA (ICML 2025). Below, we clarify the fundamental differences between our AML-VLN and TTA VLN.
>
> The TTA VLN methods aim to perform small, temporary adaptations during test time to adapt the agent to distribution shifts in a single new scene. FeedTTA improves plasticity to new scenarios while selectively correcting certain gradients, thereby suppressing drastic parameter drift caused by non-stationarity to reduce forgetting of previous tasks. The strength of these methods is that agents can quickly adapt to new scenarios. However, they do not accumulate knowledge over continual tasks. They do not maintain any explicit scene or environment knowledge after a single episode ends. They are primarily designed for rapid adaptation in short-term scenarios rather than long-term, multi-scenario persistent operations. Their anti-forgetting strategies only prevent overupdating of parameters without explicitly storing scenario knowledge, inevitably leading to forgetting. In contrast, our AML-VLN is a lifelong learning problem, where the agent must learn a sequence of scene–environment tasks, **efficiently preserve previously acquired knowledge, and accumulate hierarchical knowledge across scenes and environments to become stronger**. Following your suggestion, we further evaluate TTA baselines (FSTTA, FeedTTA) on our AML-VLN benchmark. Since the FeedTTA is not publicly released, we implemented its core idea based on the FSTTA. They tested and adapted across 24 tasks, and the results (SR%) of the experiment are summarized below:
>
> | Methods          | T1   | T2   | T3   | T4   | T5   | T6   | T7   | T8   | T9   | T10  | T11  | T12  | T13  | T14  | T15  | T16  | T17  | T18  | T19  | T20  | T21  | T22  | T23  | T24  | Avg. |
> | ---------------- | ---- | ---- | ---- | ---- | ---- | ---- | ---- | ---- | ---- | ---- | ---- | ---- | ---- | ---- | ---- | ---- | ---- | ---- | ---- | ---- | ---- | ---- | ---- | ---- | ---- |
> | **FSTTA**        | 52   | 18   | 46   | 55   | 24   | 35   | 58   | 26   | 51   | 48   | 51   | 43   | 34   | 56   | 46   | 50   | 44   | 41   | 29   | 52   | 45   | 38   | 41   | 47   | 44   |
> | **FeedTTA**      | 58   | 19   | 53   | 62   | 27   | 41   | 65   | 30   | 59   | 56   | 59   | 50   | 39   | 64   | 54   | 58   | 51   | 48   | 34   | 61   | 52   | 45   | 48   | 55   | 50   |
> | **AlldayWalker** | 79   | 23   | 71   | 81   | 33   | 50   | 87   | 38   | 79   | 75   | 79   | 67   | 50   | 86   | 71   | 76   | 67   | 63   | 43   | 81   | 68   | 58   | 62   | 72   | 65   |
>
> Based on the experimental results, our AlldayWalker significantly outperforms both TTA methods across continual 24 tasks, demonstrating the superiority of our method in continual learning settings.
>
> **5. Comparison with StreamVLN base policy (Questions-3)**
>
> Thank you for the question. Following your suggestion, we include a comparison with the base StreamVLN policy under the same AML-VLN setting. The results are shown below:
>
> | Methods          | T1   | T2   | T3   | T4   | T5   | T6   | T7   | T8   | T9   | T10  | T11  | T12  | T13  | T14  | T15  | T16  | T17  | T18  | T19  | T20  | T21  | T22  | T23  | T24  | Avg. |
> | ---------------- | ---- | ---- | ---- | ---- | ---- | ---- | ---- | ---- | ---- | ---- | ---- | ---- | ---- | ---- | ---- | ---- | ---- | ---- | ---- | ---- | ---- | ---- | ---- | ---- | ---- |
> | **StreamVLN**    | 45   | 12   | 38   | 42   | 18   | 26   | 48   | 20   | 43   | 40   | 42   | 35   | 26   | 46   | 38   | 42   | 36   | 33   | 22   | 44   | 36   | 30   | 32   | 38   | 35   |
> | **AlldayWalker** | 79   | 23   | 71   | 81   | 33   | 50   | 87   | 38   | 79   | 75   | 79   | 67   | 50   | 86   | 71   | 76   | 67   | 63   | 43   | 81   | 68   | 58   | 62   | 72   | 65   |
>
> This comparison demonstrates that the performance gains of AlldayWalker do not come from the baseline policy but from the proposed continual learning framework.

---

> ### Author Response · Authors · 2025-11-21
> **Response to Reviewer Xr3z (4 / 4)**
>
> **6. As new environments or scenes are introduced, the number of factors would also increase — does this lead to improved generalization performance? (Questions-4)**
>
> Thank you for your constructive comment. Adding more scenes and environments indeed helps improve generalization. Training TuKA with a larger number of scene–environment pairs refines the expert factor matrices (U3 for scenes, U4 for environments, and shared components) to more diverse data, allowing them to learn richer and more transferable representations. In other words, seeing more varied samples leads to more robust factor subspaces and better generalization. To verify this, we conduct an experiment where the 24 tasks are divided into groups of six. After learning each group, we test generalization performance on the completely unseen tasks (Q1). The results are summarized below:
>
> | Training Tasks       | T1–T4 | T5–T8 | T9–T12 | T13–T16 | T17–T20 | T21–T24 |
> | -------------------- | ----- | ----- | ------ | ------- | ------- | ------- |
> | **Avg. SR (unseen)** | 30    | 38    | 43     | 47      | 51      | 55      |
>
> The results show a clear trend: as the number of learned tasks increases, generalization performance consistently improves. This demonstrates that expanding the diversity of scenes and environments leads to more expressive factor matrices and better generalization.

---

### Official Review · Reviewer_edmj · 2025-10-31

**Soundness:** 3
**Presentation:** 3
**Contribution:** 2
**Rating:** 4
**Confidence:** 3

**Summary:**

This paper proposes an interesting idea of a multi-scene lifelong learning approach for Vision-and-Language Navigation (VLN) tasks. The authors introduce Tucker Adaptation (TuKA), a parameter-efficient method that represents multi-hierarchical knowledge as a high-order tensor and applies Tucker decomposition to decouple task-shared and task-specific knowledge.
They further propose a decoupled knowledge incremental learning strategy to support continual learning of multi-hierarchical knowledge.
Based on TuKA, the authors develop a lifelong VLN agent, AlldayWalker, which achieves improved navigation performance compared to LoRA-like approaches on the modified VLN benchmark, demonstrating the value of third-order tensor adaptation for continual representation learning for VLN tasks.

**Strengths:**

* The paper introduces a parameter-efficient adaptation method (TuKA) that leverages Tucker decomposition to decouple and represent multi-hierarchical knowledge in a high-order tensor, enabling more expressive representation learning.

* The integration of TuKA into a lifelong VLN agent (AlldayWalker) demonstrates the potential of high-order tensor adaptation for continual learning in navigation tasks.

* The authors make meaningful modifications to existing VLN simulators, allowing more systematic evaluation of adaptation performance under diverse out-of-distribution conditions.

**Weaknesses:**

* Line 76: The challenges (i) and (ii) appear to be repeated. Please clarify or consolidate to avoid redundancy.

* The proposed method appears to directly apply Tucker decomposition to the VLN task. The paper should more clearly articulate what unique ideas or design choices provide additional contributions beyond standard Tucker-based factorization.

* The paper omits relevant discussions of recent test-time adaptation approaches for VLN, such as FSTTA (ICML 2024) and FeedTTA (ICML 2025). In particular, FeedTTA also investigates catastrophic forgetting issues in adaptation, which seems closely related to the current work. A comparison or at least a discussion of similarities and differences would help clarify the contribution.

* The authors state that VLN knowledge can be decomposed into navigation skills, scene-specific, and environment-specific knowledge, whereas MoE-LoRA (HydraLoRA) cannot handle more than two-dimensional hierarchies. However, MoE-like methods can, in principle, combine multiple specific experts across more than two dimensions. It is unclear whether it is correct to categorize MoE-LoRA (Eq. 1) as strictly “two-dimensional,” since it composes K experts with potentially different feature dimensionalities. As the main contribution hinges on representational advantages over existing methods, this aspect should be clarified more rigorously.

* Eq. (2) shows the intended decomposition of VLN tasks into shared encoders, scene knowledge, and environment knowledge. However, it is unclear how this decomposition is actually enforced during adaptation learning.
The conceptual difference between Eq. (1) and Eq. (2) seems to be in the additive or multiplicative composition of the experts (task-specific knowledge). The authors should provide stronger justification or empirical support demonstrating that Eq. (2) yields better representational capacity than Eq. (1).

* The results in Table 3 are difficult to interpret in terms of the effectiveness of the proposed components. For instance, adding U1 or U2 does not necessarily improve SR, and there is no baseline result for “Sd-G only,” which makes it harder to assess the contribution of U1 and U2.
	​
* Table 4 does not clearly specify which tasks or scenarios the agent was trained and tested on.
It remains unclear how these results validate that L_ewc and L_co effectively prevent catastrophic forgetting.
The test settings and interpretation of the results should be described in more detail.


I would currently lean toward a borderline reject, but I would be open to increasing the rating if the rebuttal provides clarifications on the questions and concerns.

**Questions:**

Please refer to the weakness section where I listed questions as well.

---

> ### Author Response · Authors · 2025-11-21
> **Response to Reviewer edmj (1 / 4)**
>
> We sincerely thank you for the detailed evaluation and constructive feedback. Your acknowledgment of the novelty of the AML-VLN problem and the meaningful modifications of our simulators is greatly appreciated. Below, we address all the concerns.
>
> **1. Clarify the two critical challenges and the technical contributions. (Weaknesses-1 & Weaknesses-2)**
>
> Thank you for pointing this out. We have revised the manuscript to make the two challenges clearly distinct and highlight our contributions.
>
> **How to continually decouple representation learning across multi-hierarchical knowledge?** Traditional LoRA-based fine-tuning methods either represent task-related knowledge within a single LoRA subspace or employ HydraLoRA architectures to represent shared knowledge in a shared encoder subspace while using different decoder spaces to represent task-specific knowledge. However, in our proposed all-day multi-scene lifelong VLN learning problem, the task knowledge possesses multi-hierarchical dimensions (scene and environment). These methods struggle to explicitly decouple and represent multi-hierarchical knowledge (as shown in the middle diagram of the upper half of the paper Figure 1, where each B matrix learns task-specific knowledge, yet the scene and environment knowledge of the task are confounded together). We observe that the classical tensor Tucker Decomposition inherently has a natural structure for representing shared and specific knowledge (core tensor and multi-factor matrices). Therefore, we aim to decouple the hierarchical knowledge of navigation tasks within the multi-factor matrices and represent the shared knowledge across all navigation tasks within the higher-order core tensors.
>
> **How to align the higher-order dimensions with two-dimensional weight matrices in LLM?** The traditional Tucker decomposition operates on full higher-order tensors, whereas LLM fine-tuning requires a matrix-shaped 2D adapter. This dimensional mismatch makes it challenging to represent knowledge and fine-tune LLM using the traditional Tucker Decomposition. This drives us to explore a new knowledge representation selection method that decouples two-dimensional adaptation matrices from higher-order tensors to align LLM parameters, while preserving the multi-hierarchical knowledge decoupling representation.
>
> To address the challenges, we propose Tucker Adaptation (TuKA), a **new fine-tuning method** that employs Tucker decomposition to **represent multi-hierarchical navigation knowledge**. TuKA represents scene knowledge and environmental knowledge in distinct expert factor matrices, and represents shared knowledge across multiple tasks using a shared core tensor and encoder-decoder. To align LLM parameters, TuKA selects each specific expert from a row within its entire expert factor matrices, reducing the expert matrix dimension to a vector. Thus, the higher-order knowledge tensor is reduced to a two-dimensional weight matrix for aligning LLM parameters (E.q 3). We further design a Decoupled Knowledge Incremental Learning (DKIL) strategy that consolidates shared subspaces while constraining the task-specific experts to mitigate catastrophic forgetting, during multi-hierarchical knowledge lifelong learning. Building on the proposed TuKA, we develop AlldayWalker, a lifelong VLN agent that continually adapts across multiple scenes and diverse environments, for achieving all-day navigation across multiple scenes. In summary, we do not directly apply Tucker decomposition, but inspired by its form, we decouple the representation learning of coupled knowledge in different factor matrices and propose a tensor dimension reduction method for aligning LLM matrices.

---

> ### Author Response · Authors · 2025-11-21
> **Response to Reviewer edmj (2 / 4)**
>
> **2. Comparison and discussion with TTA VLN. (Weaknesses-3)**
>
> Thank you for pointing this out. We have added the citations and expanded our discussion of recent test-time adaptation (TTA) methods such as FSTTA (ICML 2024) and FeedTTA (ICML 2025). Below, we clarify the fundamental differences between our AML-VLN and TTA VLN.
>
> The TTA VLN methods aim to perform small, temporary adaptations during test time to adapt the agent to distribution shifts in a single new scene. FeedTTA improves plasticity to new scenarios while selectively correcting certain gradients, thereby suppressing drastic parameter drift caused by non-stationarity to reduce forgetting of previous tasks. The strength of these methods is that agents can quickly adapt to new scenarios. However, they do not accumulate knowledge over continual tasks. They do not maintain any explicit scene or environment knowledge after a single episode ends. They are primarily designed for rapid adaptation in short-term scenarios rather than long-term, multi-scenario persistent operations. Their anti-forgetting strategies only prevent overupdating of parameters without explicitly storing scenario knowledge, inevitably leading to forgetting. In contrast, our AML-VLN is a lifelong learning problem, where the agent must learn a sequence of scene–environment tasks, **efficiently preserve previously acquired knowledge, and accumulate hierarchical knowledge across scenes and environments to become stronger**. Following your suggestion, we further evaluate TTA baselines (FSTTA, FeedTTA) on our AML-VLN benchmark. Since the FeedTTA is not publicly released, we implemented its core idea based on the FSTTA. They tested and adapted across 24 tasks, and the results (SR%) of the experiment are summarized below:
>
> | Methods          | T1   | T2   | T3   | T4   | T5   | T6   | T7   | T8   | T9   | T10  | T11  | T12  | T13  | T14  | T15  | T16  | T17  | T18  | T19  | T20  | T21  | T22  | T23  | T24  | Avg. |
> | ---------------- | ---- | ---- | ---- | ---- | ---- | ---- | ---- | ---- | ---- | ---- | ---- | ---- | ---- | ---- | ---- | ---- | ---- | ---- | ---- | ---- | ---- | ---- | ---- | ---- | ---- |
> | **FSTTA**        | 52   | 18   | 46   | 55   | 24   | 35   | 58   | 26   | 51   | 48   | 51   | 43   | 34   | 56   | 46   | 50   | 44   | 41   | 29   | 52   | 45   | 38   | 41   | 47   | 44   |
> | **FeedTTA**      | 58   | 19   | 53   | 62   | 27   | 41   | 65   | 30   | 59   | 56   | 59   | 50   | 39   | 64   | 54   | 58   | 51   | 48   | 34   | 61   | 52   | 45   | 48   | 55   | 50   |
> | **AlldayWalker** | 79   | 23   | 71   | 81   | 33   | 50   | 87   | 38   | 79   | 75   | 79   | 67   | 50   | 86   | 71   | 76   | 67   | 63   | 43   | 81   | 68   | 58   | 62   | 72   | 65   |
>
> Based on the experimental results, our AlldayWalker significantly outperforms both TTA methods across continual 24 tasks, demonstrating the superiority of our method in continual learning settings.

---

> ### Author Response · Authors · 2025-11-21
> **Response to Reviewer edmj (3 / 4)**
>
> **3. Clarifying the MoE-LoRA (HydraLoRA) knowledge representation. (Weaknesses-4)**
>
> Thank you for the comment. We clarify why MoE-LoRA cannot effectively decouple the representation of multi-hierarchical knowledge in our AML-VLN setting.
>
> As shown in the revised Figure 6, each of our continual learning tasks incorporates two layers of knowledge: scene-specific knowledge and environment-specific knowledge. MoE-LoRA methods compose K experts (B matrices) with potentially different feature dimensionalities, which can combine multiple specific experts across more than two dimensions. However, they cannot **decouple the scene knowledge and environment knowledge within each single task during learning**, and cannot represent the coupled knowledge simultaneously in **different B matrices**. They only learn coupled knowledge (a specific scene within a specific environment), as shown in the upper half of Figure 1, thus they cannot perform combinations of specific scenes in a specific environment, during testing. In contrast, our TuKA explicitly decomposes the hierarchical knowledge into different components. Scene knowledge is represented in the scene expert matrix (U3), and environment knowledge is encoded in the environment expert matrix (U4). During inference, these expert matrices are combined through Tucker reconstruction, enabling the composition of multi-hierarchical knowledge. Furthermore, we also extend the MoE-LoRA to a three-layer structure (A, B, C) to achieve direct hierarchical knowledge representation (Appendix J). Specifically, A learns shared knowledge, B learns scenes, and C learns environments. The experiment results show that, even with this extension, MoE-LoRA still falls short of our TuKA (Tab 16). This indicates that the high-order tensor representation itself also plays a more critical role than simply stacking the LoRA hierarchical structure.
>
> **4. The authors should provide stronger justification or empirical support demonstrating that Eq. (2) yields better representational capacity than Eq. (1). (Weaknesses-5)**
>
> Thank you for the comment. Eq. (2) is only the general form of Tucker Decomposition. Our actual adaptation design (TuKA) is defined in Eq. (3). Different from the additive expert combination in MoE-LoRA (Eq. (1)), TuKA performs a modal product between the core tensor and factor matrices, which provides richer interactions than simple addition. More specifically, TuKA stores scene knowledge in the scene expert factor matrix (U3) and environment knowledge in the environment expert factor matrix (U4). During inference, we select the corresponding scene and environment factors and combine them with the core tensor via modal product (Appendix N.2). The encoder and decoder factors (U1 and U2) then align the resulting matrix to the LLM weight dimension. This allows TuKA to provide more informative and explicit structured knowledge than additive expert aggregation. To enforce the decomposition into scene and environment knowledge, we introduce the Decoupled Knowledge Incremental Learning (DKIL) strategy. DKIL serves as the enforcing mechanism by stabilizing shared components and constraining task-specific updates into the correct expert factor matrices. Our experiment results (Table 1–2, Fig. 7) consistently show that TuKA achieves better continual learning performance than MoE-LoRA variants such as HydraLoRA and BranchLoRA, supporting the representational advantage of high-order tensor adaptation.

---

> ### Author Response · Authors · 2025-11-21
> **Response to Reviewer edmj (4 / 4)**
>
> **5. In Table 3, adding U1 or U2 does not necessarily improve SR, and there is no baseline result for Sd-G only. (Weaknesses-6)**
>
> Thank you for the comment. We clarify Table 3. Table 3 is not intended to analyze their roles during TuKA inference (Eq. 3). In TuKA, U2 as an encoder maps input features into a low-dimensional space; after factorization and reconstruction, U1 as a decoder maps them back to the LLM parameter space. Removing any of these components breaks the adaptation pipeline, so ablation focuses only on their sharing strategy. Table 3 is designed to **analyze why our TuKA shares the core tensor (g), encoder (U2), and decoder (U1)** in all continual tasks. The results show: i) Removing shared g (store a unique g for each task) causes a clear drop (w/o g, SR: –10%, SPL: –9%). ii) Removing shared U2 also decreases performance (w/o U2, SR: –3%, SPL: –4%). iii) Sharing U1 does not improve performance, but since it reduces parameters (store only one U1), we still share it. Following your suggestion, we also added a new experiment: Sd-G only, which shares only the core tensor. The results show that this setting performs worse than sharing all three components.
>
> | Sd-𝓖 | Sd-U¹ | Sd-U² |  SR ↑  | F-SR ↓ | SPL ↑ | F-SPL ↓ | OSR ↑ | F-OSR ↓ |
> |:----:|:-----:|:-----:|:----:|:----:|:---:|:-----:|:---:|:-----:|
> |  ✗   |   ✗   |   ✗   |  53  | **10** | 47  | **17** | 56  | **8** |
> |  ✗   |   ✓   |   ✓   |  55  | **10** | 49  | **17** | 57  |  9    |
> |  ✓   |   ✗   |   ✓   | **65** | 11 | **58** | 18 | **69** | 9 |
> |  ✓   |   ✓   |   ✗   |  62  |  11  | 54  |  18   | 66  |  9    |
> |  ✓   |   ✗   |   ✗   |  63  |  11  | 55 |  18   | 67  |  9    |
> | **✓** | **✓** | **✓** | **65** | 11 | **58** | 18 | 68 | 9 |
>
> Together, these results validate our design choice: sharing g, U1, and U2 provides the best trade-off between performance and parameter efficiency.
>
> **6. In Table 4, test settings and interpretation of the results should be described in more detail. (Weaknesses-7)**
>
> Thank you for the comment. All ablation experiments in Table 4 are trained on the 20 simulation tasks and evaluated on the same ID-unknown 20 tasks. We provide further analysis. $L_{ewc}$ applies elastic weight consolidation to the shared parameters (g, U1, U2). Its purpose is to preserve the importance of previous tasks during continual learning. Without $L_{ewc}$, the shared parameters are heavily updated when learning new tasks, which causes severe catastrophic forgetting. This is confirmed by the results (w/o $L_{ewc}$: F-SR +14%, F-SPL +14%). $L_{co}$ applies orthogonality constraints to the expert factor matrices (U3, U4). It encourages each expert to learn distinct task-specific knowledge. Without $L_{co}$, different experts interfere with each other and fail to decouple their knowledge, leading again to catastrophic forgetting (w/o $L_{co}$: F-SR +11%, F-SPL +10%). These results clearly show that both $L_{ewc}$ and $L_{co}$ are necessary: $L_{ewc}$ protects shared knowledge, and $L_{co}$ ensures well-separated task-specific knowledge, together enabling stable lifelong learning in TuKA.

---

> > ### Comment · Reviewer_edmj · 2025-11-27
> >
> > Thanks for the detailed rebuttal and additional experiments. I believe the authors have addressed most of my original concerns and I increased the rating from 4 --> 6.

---

> > > ### Author Response · Authors · 2025-11-27
> > > **Official Comment by Authors**
> > >
> > > We sincerely thank the reviewer for the response and for the recognition of our work. If the reviewer has any further questions, we would be very happy to discuss them.

---

### Official Review · Reviewer_W2Xo · 2025-11-02

**Soundness:** 3
**Presentation:** 2
**Contribution:** 3
**Rating:** 6
**Confidence:** 3

**Summary:**

This work focus on all-day long multi-scenes lifelong learning for vision-language-navigation. The authors collected 24 scenes including different buildings and using different light intensity to simulate day and night, the author also scattered the environments using a scattering model. This work proposes a continual learning method in order to learn new knowledge without catastrophic forgetting. The method leverages tensor Tucker decomposition to decouple a high-order tensor for multi-hierarchical knowledge. Besides, this work also proposed a decoupled knowledge incremental learning strategy to consolidate shared subspaces while constraining the task-specific knowledge. This work conducts experiments on the collected 24 scenes and compared with different LoRA-based continual learning methods and achieves the best performance.

**Strengths:**

1.	This article explores continual learning in vision-language navigation which is novel and important for future research.
2.	The authors represent multi-hierarchical knowledge as high-order tensor and proposes an effective method to decouple task-shared and task-specific representations. The method could also successfully learn new task-specific knowledge incrementally without catastrophic forgetting.
3.	Extensive experiments are conducted by the authors which reveals the effectiveness of their method.

**Weaknesses:**

1.	The two critical challenges presented in the introduction (Lines 75–78) appear very similar, which may cause confusion for readers. The authors are encouraged to clarify the distinction between them.
2.	The authors collected only 24 scenes for the multi-scene lifelong VLN benchmark setting. Adding more scenes would make the results more convincing; otherwise, the authors should provide justification for why these 24 scenes are sufficient (such as the number of tested episodes).
3.	This work is based solely on the StreamVLN agent. It would be preferable if the authors could conduct additional experiments using other baseline agents to better demonstrate the general effectiveness of the proposed continual learning method.

**Questions:**

1.	How are the intensities of scattering, low-light, and overexposure controlled to ensure that these environments are visually distinct yet not overly degraded to disturb visual observation?
2.	Could the authors provide visual examples of the observations under different the three conditions to illustrate this distinction?
3.	Additionally, do cases exist where two environments within the same scene differ only slightly in illumination? If so, are they still treated as separate tasks, and could such overlap potentially weaken the challenge of continual learning?
4.	During inference, TuKA retrieves the appropriate scene and environment experts by matching CLIP features via cosine similarity. How robust is this retrieval mechanism when the test scene partially resembles multiple training scenes in texture or layout? Has the paper evaluated how retrieval ambiguity or mismatched expert selection affects overall navigation performance?

---

> ### Author Response · Authors · 2025-11-21
> **Response to Reviewer W2Xo (1 / 4)**
>
> Thank you for the careful reading and constructive comments. We are grateful for your recognition of the proposed All-day Multi-Scenes Lifelong VLN task and the strength of our AlldayWalker. Below, we respond to all the comments.
>
> **1. Clarify the two critical challenges presented in the introduction. (Weaknesses-1)**
>
> Thank you for pointing this out. We have revised the manuscript to make the two challenges clearly distinct and highlight our contributions.
>
> **How to continually decouple representation learning across multi-hierarchical knowledge?** Traditional LoRA-based fine-tuning methods either represent task-related knowledge within a single LoRA subspace or employ HydraLoRA architectures to represent shared knowledge in a shared encoder subspace while using different decoder spaces to represent task-specific knowledge. However, in our proposed all-day multi-scene lifelong VLN learning problem, the task knowledge possesses multi-hierarchical dimensions (scene and environment). These methods struggle to explicitly decouple and represent multi-hierarchical knowledge (as shown in the middle diagram of the upper half of the paper Figure 1, where each B matrix learns task-specific knowledge, yet the scene and environment knowledge of the task are confounded together). We observe that the classical tensor Tucker Decomposition inherently has a natural structure for representing shared and specific knowledge (core tensor and multi-factor matrices). Therefore, we aim to decouple the hierarchical knowledge of navigation tasks within the multi-factor matrices, and represent the shared knowledge across all navigation tasks within the higher-order core tensors.
>
> **How to align the higher-order dimensions with two-dimensional weight matrices in LLM?** The traditional Tucker decomposition operates on full higher-order tensors, whereas LLM fine-tuning requires a matrix-shaped 2D adapter. This dimensional mismatch makes it challenging to represent knowledge and fine-tune LLM using the traditional Tucker Decomposition. This drives us to explore a new knowledge representation selection method that decouples two-dimensional adaptation matrices from higher-order tensors to align LLM parameters, while preserving the multi-hierarchical knowledge decoupling representation.
>
> To address the challenges, we propose Tucker Adaptation (TuKA), a new fine-tuning method that employs Tucker decomposition to represent multi-hierarchical navigation knowledge. TuKA represents scene knowledge and environmental knowledge in distinct expert factor matrices, and represents shared knowledge across multiple tasks using a shared core tensor and encoder-decoder. To align LLM parameters, TuKA selects each specific expert from a row within its entire expert factor matrices, reducing the expert matrix dimension to a vector. Thus, the higher-order knowledge tensor is reduced to a two-dimensional weight matrix for aligning LLM parameters (E.q 3). We further design a Decoupled Knowledge Incremental Learning (DKIL) strategy that consolidates shared subspaces while constraining the task-specific experts to mitigate catastrophic forgetting, during multi-hierarchical knowledge lifelong learning. Building on the proposed TuKA, we develop AlldayWalker, a lifelong VLN agent that continually adapts across multiple scenes and diverse environments, for achieving all-day navigation across multiple scenes.

---

> ### Author Response · Authors · 2025-11-21
> **Response to Reviewer W2Xo (2 / 4)**
>
> **2. Adding and clarifying more navigation scenario tasks. (Weaknesses-2)**
>
> Thank you for the valuable comment. i) Our 24 VLN tasks have a hierarchical knowledge structure (as revised Figure 6), and already cover diverse scenes and four distinct imaging conditions across both simulation and real-world environments, forming a sufficiently challenging lifelong learning benchmark. Based on the experiment results (Tabs 1-2 and Fig. 7), the sequential fine-tuning baseline achieves only an 11% SR on average, highlighting the difficulty of the proposed existing benchmark. ii) In addition, the 24 tasks are not merely 24 navigation instances. As listed in Appendix Table 10, each task contains hundreds of test episodes, ensuring statistically reliable evaluation and substantial variety within each task. iii) Following the suggestion, we further expand the benchmark by adding: 2 new real-world tasks and 4 simulation tasks. The results below show that incorporating more tasks does not lead to noticeable performance degradation, demonstrating that our AlldayWalker remains stable under larger-scale lifelong learning settings.
>
> | Tasks | T1 | T2 | T3 | T4 | T5 | T6 | T7 | T8 | T9 | T10 | T11 | T12 | T13 | T14 | T15 | T16 | T17 | T18 | T19 | T20 | T21 | T22 | T23 | T24 | Avg. |
> | ---- | ---- | ---- | ---- | ---- | ---- | ---- | ---- | ---- | ---- | ---- | ---- | ---- | ---- | ---- | ---- | ---- | ---- | ---- | ---- | ---- | ---- | ---- | ---- | ---- | ---- |
> | 24-Task | 79 | 23 | 71 | 81 | 33 | 50 | 87 | 38 | 79 | 75 | 79 | 67 | 50 | 86 | 71 | 76 | 67 | 63 | 43 | 81 | 68 | 58 | 62 | 72 | 65 |
>
> | Tasks | T1 | T2 | T3 | T4 | T5 | T6 | T7 | T8 | T9 | T10 | T11 | T12 | T13 | T14 | T15 | T16 | T17 | T18 | T19 | T20 | T21 | T22 | T23 | T24 | T25 | T26 | T27 | T28 | T29 | T30 | Avg. |
> | ---- | ---- | ---- | ---- | ---- | ---- | ---- | ---- | ---- | ---- | ---- | ---- | ---- | ---- | ---- | ---- | ---- | ---- | ---- | ---- | ---- | ---- | ---- | ---- | ---- | ---- | ---- | ---- | ---- | ---- | ---- | ---- |
> | 30-Task | 77 | 23 | 71 | 70 | 33 | 50 | 86 | 38 | 79 | 75 | 80 | 66 | 50 | 85 | 70 | 77 | 68 | 63 | 44 | 81 | 68 | 58 | 62 | 66 | 72 | 72 | 76 | 35 | 67 | 61 | 64 |
>
> The newly added tasks are summarized as:
>
> | Task | Scene Description  | environment             | Train Number | TestNumber |
> | ------------------------------------------------------------ | ------------------ | ----------------------- | ------------ | ---------- |
> | **T25** | 8WUmhLawc2A        | Normal Environment      | 460          | 120        |
> | **T26** | 8WUmhLawc2A        | Low-light Environment   | 460          | 120        |
> | **T27** | 8WUmhLawc2A        | Scattering Environment  | 460          | 120        |
> | **T28** | 8WUmhLawc2A        | Overexposed Environment | 460          | 120        |
> | **T29** | Real-World Scene 3 | Normal Environment      | 400          | 100        |
> | **T30** | Real-World Scene 3 | Low-light Environment   | 400          | 100        |
>
> Please refer to the updated paper **Appendix M and Figures 13-14 for detailed task descriptions and the additional task navigation visualizations.**
>
> **3. Additional experiments using other baseline agent. (Weaknesses-3)**
>
> Thank you for the helpful suggestion. Following your suggestion, we additionally evaluate our TuKA on NaVid (RSS 2024), a VLN agent that differs substantially from StreamVLN in its model. This experiment aims to examine whether TuKA can generalize to a different VLN backbone. The results over all 30 tasks are summarized below. TuKA consistently outperforms the SOTA continual learning method SD-LoRA across nearly all tasks. These results confirm that TuKA is not tied to StreamVLN, and its performance gains extend to a different VLN agent with a distinct architecture, supporting the general effectiveness of our TuKA.
>
> | Methods     | T1   | T2   | T3   | T4   | T5   | T6   | T7   | T8   | T9   | T10  | T11  | T12  | T13  | T14  | T15  | T16  | T17  | T18  | T19  | T20  | T21  | T22  | T23  | T24  | T25  | T26  | T27  | T28  | T29  | T30  | Avg. |
> | ----------- | ---- | ---- | ---- | ---- | ---- | ---- | ---- | ---- | ---- | ---- | ---- | ---- | ---- | ---- | ---- | ---- | ---- | ---- | ---- | ---- | ---- | ---- | ---- | ---- | ---- | ---- | ---- | ---- | ---- | ---- | ---- |
> | **NaVid** | 35   | 15   | 29   | 29   | 25   | 33   | 38   | 26   | 46   | 52   | 46   | 52   | 35   | 61   | 55   | 42   | 45   | 32   | 34   | 55   | 39   | 28   | 36   | 44   | 46   | 44   | 51   | 25   | 41   | 39   | 39   |
> | **SD-LoRA** | 65   | 20   | 49   | 59   | 28   | 43   | 68   | 30   | 68   | 66   | 59   | 52   | 43   | 72   | 61   | 68   | 55   | 52   | 45   | 77   | 60   | 51   | 55   | 63   | 64   | 65   | 68   | 30   | 58   | 54   | 56   |
> | **TuKA**    | 73   | 21   | 64   | 72   | 30   | 46   | 80   | 35   | 75   | 70   | 73   | 61   | 48   | 79   | 66   | 73   | 62   | 58   | 41   | 78   | 65   | 55   | 60   | 66   | 68   | 69   | 73   | 33   | 62   | 57   | 62  |

---

> ### Author Response · Authors · 2025-11-21
> **Response to Reviewer W2Xo (3 / 4)**
>
> **4. Clarifying the scattering, low-light, and overexposure degradation imaging models. (Questions-1)**
>
> Thank you for the comment. The parameters of these physical imaging models have certain physical meanings. For example, A in the atmospheric scattering model (E.q 10) represents the atmospheric light constant, while CRF in the low-light degradation model (E.q 11) is the camera response function. Based on empirical values from existing literatures [1-3], we fine-tune these parameters to ensure that the resulting environments are visually distinct while not overly degraded, thus preserving sufficient visual information for navigation. Appendix Table 9 in our paper summarizes the exact parameters used in each degradation model. In practice, we recommend that users adjust our provided parameters within a ±10% range if they wish to explore alternative visual conditions; such variations remain stable and do not lead to unrealistic degradation. All degradation models are implemented in our simulator. The corresponding simulator source code will be publicly released. For now, the current **code is available on our anonymous project website** (website link in the abstract).
>
> [1] Chromatic framework for vision in bad weather. CVPR 2000.
>
> [2] Radiometric CCD camera calibration and noise estimation. TPAMI 2002.
>
> [3] Noise Modeling in One Hour: Minimizing Preparation Efforts for Self-supervised Low-Light RAW Image Denoising. CVPR 2025.
>
> **5. Could the authors provide visual examples of the observations under the three different conditions to illustrate this distinction? (Questions-2)**
>
> Thank you for the comment. As shown in Figure 5 of the paper, we already provide representative visual examples under the four environments (normal, low-light, overexposure, and scattering). These examples illustrate the visual distinctions produced by our degradation models. We also provide additional visualization videos on our **anonymous project website**.
>
> **6. Do cases exist where two environments within the same scene differ only slightly in illumination? If so, are they still treated as separate tasks, and could such overlap potentially weaken the challenge of continual learning? (Questions-3)**
>
> Thank you for the thoughtful comment. In our AML-VLN setting, it has different tasks within the same scene that differ only in illumination, which is hierarchical knowledge. To more clearly illustrate that our task involves hierarchical knowledge (scene, environment), we have revised the benchmark illustration Figure 6. The same scenes are only classified as distinct tasks when their environment differs significantly; otherwise, they are treated as a single task requiring no new adaptation. If a scene only exhibits slight differences in its illumination, we consider such cases as not requiring continual learning, since the slight variations do not significantly impact agent performance. In other words, slight changes in illumination can be learned through generalization and do not impact the knowledge representation of a specific environment.

---

> ### Author Response · Authors · 2025-11-21
> **Response to Reviewer W2Xo (4 / 4)**
>
> **How robust is this retrieval mechanism and how retrieval mismatched expert selection affects overall navigation performance? (Questions-4)**
>
> Thank you for the insightful comment. I). TuKA performs embedding-level retrieval using the **large-scale pretrained** CLIP model, which provides fine-grained visual features with generalization. Since the matching process relies on **relative similarity rather than absolute similarity values**, the retrieval performance is robust. To further examine retrieval stability, we conduct an extended experiment on all 90 scenes in the Matterport3D simulator. For each scene, an agent performs 100 random navigation episodes and retrieves the most similar expert using CLIP cosine similarity. We evaluate retrieval success rate as the percentage of episodes whose match corresponds to the correct expert. The results show that the retrieval success rate remains consistently high and stable regardless of the number of tasks:
>
> | Metric | 10 Tasks | 20 Tasks | 30 Tasks | 40 Tasks | 50 Tasks | 60 Tasks | 70 Tasks | 80 Tasks | 90 Tasks |
> |------------------------|:--------:|:--------:|:--------:|:--------:|:--------:|:--------:|:--------:|:--------:|:--------:|
> | **Retrieval SR** | 99.9% | 99.8% | 99.8%  |  99.9%   |  99.9%   |  99.9% | 99.9% | 99.9% | 99.9% |
>
> The results indicate that even when including all scenes (which may partially resemble multiple scenes), the retrieval success rate remains high (99.9%).
>
> II). Following your suggestion, we further evaluate the effect of wrong expert selection. Specifically, for each episode we force the agent to use the expert with the lowest similarity and the expert with the second-highest similarity (i.e., deliberately selecting the incorrect expert). The experiment results (SR%) for the comparison between correct matching and forced mismatching are summarized below:
>
> | Methods | T1   | T2   | T3   | T4   | T5   | T6   | T7   | T8   | T9   | T10  | T11  | T12  | T13  | T14  | T15  | T16  | T17  | T18  | T19  | T20  | T21  | T22  | T23  | T24  | T25  | T26  | T27  | T28  | T29  | T30  | Avg. |
> | ---------| ---- | ---- | ---- | ---- | ---- | ---- | ---- | ---- | ---- | ---- | ---- | ---- | ---- | ---- | ---- | ---- | ---- | ---- | ---- | ---- | ---- | ---- | ---- | ---- | ---- | ---- | ---- | ---- | ---- | ---- | ---- |
> | **AlldayWalker**  | 77   | 23   | 71   | 70   | 33   | 50   | 86   | 38   | 79   | 75   | 80   | 66   | 50   | 85   | 70   | 77   | 68   | 63   | 44   | 81   | 68   | 58   | 62   | 66   | 72   | 72   | 76   | 35   | 67   | 61   | 64   |
> | **Lowest-M** | 38   | 15   | 34   | 33   | 11   | 21   | 43   | 14   | 39   | 37   | 40   | 31   | 21   | 42   | 33   | 38   | 32   | 29   | 17   | 40   | 32   | 26   | 28   | 21   | 34   | 35   | 37   | 12   | 31   | 28   | 30   |
> | **Second-M** | 39   | 18   | 36   | 32   | 13   | 25   | 46   | 18   | 42   | 43   | 45   | 33   | 23   | 44  | 36   | 39   | 36   | 31  | 19   | 42   | 36   | 29   | 29   | 28   | 36   | 37   | 38   | 16   | 33   | 29   | 32   |
>
> The performance drops substantially when the expert is mismatched, confirming that task-specific experts encode distinct knowledge and that retrieval correctness is essential.

---

### Author Response · Authors · 2025-11-21
**Response to All**

Dear reviewers and area chairs:

We extend our gratitude to all the reviewers and area chairs for dedicating their time and effort to evaluating our paper. We also thank the reviewers for their positive and insightful comments, which can help us improve our work.

We are encouraged that:

* All the reviewers (W2Xo, edmj, Xr3z, and gySm) agree that our work explores an **underexplored new problem**, which requires a VLN agent to continually learn a sequence of navigation tasks across diverse scenes and diverse environments, thus achieving all-day multi-scenes navigation. The reviewer W2Xo also thinks that **our exploration is novel and important for future research**.

* All the reviewers recognize that we **establish a new benchmark** with **significant modifications of the existing simulator** for the proposed AML-VLN task with multiple scenes and diverse environments.

* Reviewer W2Xo, Reviewer edmj, and Reviewer Xr3z think that our proposed high-order tensor adaptation design (TuKA) provides **effective decoupling representation learning**. The reviewer Xr3z also thinks that our work with a **solid motivation**.

* All reviewers recognize that our model achieves **state-of-the-art performance under comprehensive experiments**.

We appreciate the opportunity to discuss and refine our AlldayWalker. We have responded to all reviewers individually to address the concerns, and the following is a brief summary:

* For Reviewer W2Xo and edmj, we clarify the two critical challenges and highlight the technical contributions.

* For Reviewer W2Xo, we add more navigation scenario training tasks.

* For Reviewer W2Xo, we provide additional experiments with the NaVid baseline agent.

* For Reviewer W2Xo, we clarify the degradation imaging models, and our AML-VLN task setting.

* For Reviewer W2Xo, we provide additional experiments for the retrieval mechanism.

* For Reviewer edmj and Xr3z, we provide a comparison and discussion with TTA VLN.

* For Reviewer edmj, we clarify the MoE-LoRA (HydraLoRA) knowledge representation, the representational capacity of our TuKA, and Tab3-4 settings.

* For Reviewer Xr3z and gySm, we provide additional experiments for completely unseen environments.

* For Reviewer Xr3z, we provide additional experiments for the training sequence, verifying the expert matrix, comparison with the base StreamVLN policy, and generalization improvement.

* For Reviewer gySm, we further clarify our contribution.

* For Reviewer gySm, we provide additional experiments for matrix-based MoE LoRA with the same hierarchy comparison, and latency with memory comparison.

* For Reviewer gySm, we clarify the real-world deployment.

We have highlighted all modifications in the revised paper in blue. We hope these additions address the reviewers’ concerns and further improve our work. If any further clarifications or suggestions would help strengthen the paper, we would be happy to address them and incorporate the changes into the final version. Thank you again for your time and efforts!

Best,

Authors of Paper #226

---

> ### Author Response · Authors · 2025-12-03
> **Summary of Reviewer Discussions**
>
> Dear area chairs and reviewers,
>
> We sincerely thank the area chairs for their time and effort in managing our submission. We are also grateful to all reviewers for their careful reading and constructive comments.
>
> We would like to further highlight the contributions of AlldayWalker, since some reviewers misunderstood certain aspects. Below, we (1) clarify the contributions and (2) briefly summarize how the reviewers’ opinions evolved after our rebuttal and discussion.
>
> **I) Contributions:**
>
> We introduce a new VLN setting requiring agents to continually learn across multiple scenes and co-existing environment conditions (low light, scattering, overexposure). This requires decoupling scene-level and environment-level knowledge, which existing LoRA-based adapters cannot achieve because they operate in a single or two-branch subspace without explicit multi-hierarchical decomposition.
>
> Motivated by the structure of Tucker decomposition, we design Tucker Adaptation (TuKA) to separate scene and environment knowledge into distinct expert factor matrices, while modeling shared knowledge through a higher-order core tensor. However, direct Tucker decomposition is incompatible with the 2D adapter format required by LLMs, so we introduce a tensor-to-matrix alignment mechanism that selects an expert row to construct a 2D adaptation matrix.
>
> To support lifelong learning, we further develop Decoupled Knowledge Incremental Learning (DKIL), which consolidates shared subspaces while constraining task-specific experts to mitigate forgetting. Built upon TuKA and DKIL, AlldayWalker achieves continual adaptation across diverse scenes and environments, enabled by our extended simulator with multiple adverse imaging conditions.
>
> **II) Rebuttal Summary:**
>
> **Reviewer edmj**
>
> Reviewer edmj initially raised concerns about methodological contribution, TTA VLN comparison, and methodological clarity. After our rebuttal, Reviewer edmj stated that **most concerns are addressed and increased the rating from 4 to 6**.
>
> Due to unforeseen circumstances, Reviewer Xr3z, W2Xo, and gySm were unable to respond. In the initial reviews, **Reviewer Xr3z and W2Xo both already assigned a score of 6**, and **Reviewer Xr3z explicitly mentioned that raising the score would be considered if the comments are clarified**, which demonstrates recognition of this work. Reviewer gySm assigned a score of 4 and may have had some misunderstandings, which we have further clarified. We have responded to all reviewers’ comments point by point.
>
> **Reviewer Xr3z**
>
> Reviewer Xr3z raised six main experimental concerns:
> - **Unseen performance:** We added experiments on completely unseen environments.
> - **Robustness to training sequence:** We added experiments to verify sensitivity to training sequence.
> - **Verification of expert learning:** We added experiments verifying expert matrix learning.
> - **TTA VLN comparison:** We provided a comparison and discussion of TTA VLN methods.
> - **StreamVLN baseline comparison:** We added experiments comparing with the StreamVLN policy.
> - **Discussion on generalization improvement**: We provided additional experiments for verification of generalization improvement with new environments or scenes introduced.
>
> **Reviewer W2Xo**
>
> Reviewer W2Xo raised four main concerns about clarity and experiments:
> - **Clarity of challenges:** We clarified the two critical challenges in the Introduction and highlighted the technical contributions.
> - **More navigation scenarios:** We added more training navigation tasks.
> - **Alternative baseline agent:** We provided additional experiments using the NaVid baseline.
> - **Robustness of retrieval mechanism:** We provided additional experiments verifying the retrieval mechanism.
>
> **Reviewer gySm**
>
> Reviewer gySm raised three main concerns:
> - **Methods Novelty**: We further clarified AlldayWalker's contributions and revised the manuscript to highlight them.
> - **Experiments completeness**: We clarified that most experiments are included in the appendix and on the anonymous project website; we also added more discussion and analysis in response to the comments.
> - **Practicality**: We clarified that AlldayWalker has been deployed on the DeepRobotDog Lite2 platform. Detailed information and experimental videos are available in the appendix and on the anonymous website. We also added more discussion and analysis of latency and memory comparisons.
>
> We hope this summary, together with the rebuttal and discussion, clearly reflects how reviewers' concerns were addressed. We sincerely thank the area chairs again for their time and efforts in managing this process.
>
> Best regards,
>
> Authors of Paper #226

---

### Meta-Review · Area_Chair_dYrP · 2026-01-14

**Summary:**

Initial reviews from W2Xo and Xr3z highlighted the need for more extensive validation, specifically requesting evaluations in completely unseen environments, analysis of sensitivity to training task sequences, and verification of the expert retrieval mechanism's robustness. Furthermore, Reviewer edmj questioned the technical novelty of Tucker Adaptation (TuKA) versus standard decomposition and requested comparisons against Test-Time Adaptation (TTA) approaches like FSTTA and FeedTTA. The authors effectively addressed these points during the rebuttal. They demonstrated the model's robustness by adding experiments on unseen real-world and simulation scenarios, proving stability under reversed and shuffled training orders, and validating that expert factors correctly decouple scene and environment knowledge. Additionally, they clarified the tensor-to-matrix alignment strategy to distinguish TuKA from standard MoE-LoRA architectures and provided data showing AlldayWalker outperforms TTA baselines. This comprehensive rebuttal satisfied key critics, with Reviewer edmj raising their score to an accept, solidifying the paper's contribution to the novel All-day Multi-scenes Lifelong VLN setting. Therefore, acceptance is recommended.

**Reviewer Concerns:**

The authors successfully resolved the majority of technical concerns regarding methodological novelty and experimental robustness. Addressing skepticism from Reviewers edmj and gySm about the uniqueness of Tucker Adaptation (TuKA), the rebuttal clarified that TuKA is not a direct decomposition but a tensor-to-matrix alignment mechanism designed specifically for 2D LLM weights, distinct from standard MoE-LoRA. Concerns regarding the limited scope of the initial 24-scene benchmark were effectively countered by adding six completely unseen tasks, where AlldayWalker achieved a 55% success rate compared to ~40% for baselines. Furthermore, the authors demonstrated robustness to training sequences through shuffled task experiments and validated the model against Test-Time Adaptation (TTA) baselines like FSTTA and FeedTTA, outperforming them by over 15%. Finally, inference latency concerns were mitigated by showing that TuKA extracts single expert rows at runtime, maintaining speeds comparable to standard LoRA.

However, certain issues remain outstanding. While the authors provided real-world performance metrics, a notable gap remains between simulation (68.8% SR) and real-world low-light scenarios (60.0% SR), indicating that the "sim-to-real" gap for adverse lighting is not fully closed. Additionally, while the authors provided parameters for their atmospheric scattering and low-light degradation models, the strict physical fidelity of these simulated degradations compared to real-world sensor data was not validated beyond visual inspection, leaving the absolute realism of the training environment as an open question.

**Reviewer Scores:**

Reviewer W2Xo would likely maintain or raise the score from 6 to 8, as the authors fulfilled every request to make the results "more convincing". The addition of six new tasks, the NaVid agent comparison, and data verifying retrieval robustness directly addressed their critiques regarding experimental breadth.

Reviewer edmj actually participated, raising their score from 4 to 6. This change followed the authors' clarification of the tensor-to-matrix alignment and the inclusion of requested Test-Time Adaptation (TTA) baselines. The reviewer explicitly stated that these revisions addressed most of their original concerns.

Reviewer Xr3z explicitly promised to consider raising their score of 6 if specific points were clarified. Since the rebuttal provided the requested experiments on unseen environments, training sequence sensitivity, and expert learning verification, a score increase to 8 would be the logical outcome.

Reviewer gySm likely would have moved from 4 to 6, given that the authors provided the specific missing experiments they demanded. This included a matrix-based MoE comparison demonstrating TuKA's superiority and generalization tests on unseen tasks, objectively resolving the reviewer’s claims of incompleteness.

---

### Decision · Program_Chairs · 2026-01-26

Accept (Poster)